# Convergence and Implicit Bias of Gradient Descent on Continual Linear Classification

**Hyunji Jung**[*]
Graduate School of Artificial Intelligence
POSTECH
jeonghj2001@postech.ac.kr

**Hanseul Cho**[*]**, Chulhee Yun**
Kim Jaechul Graduate School of AI
KAIST
{jhs4015, chulhee.yun}@kaist.ac.kr

## Abstract

We study continual learning on multiple linear classification tasks by sequentially running gradient descent (GD) for a fixed budget of iterations per task. When all tasks are jointly linearly separable and are presented in a cyclic/random order, we show the directional convergence of the trained linear classifier to the *joint (offline) max-margin* solution. This is surprising because GD training on a single task is implicitly biased towards the individual max-margin solution for the task, and the direction of the joint max-margin solution can be largely different from these individual solutions. Additionally, when tasks are given in a cyclic order, we present a non-asymptotic analysis on *cycle-averaged forgetting*, revealing that (1) alignment between tasks is indeed closely tied to catastrophic forgetting and backward knowledge transfer and (2) the amount of forgetting vanishes to zero as the cycle repeats. Lastly, we analyze the case where the tasks are no longer jointly separable and show that the model trained in a cyclic order converges to the unique minimum of the joint loss function.

## 1 Introduction

Continual learning (CL) aims to sequentially learn a model from a stream of tasks or datasets, to extend its knowledge continuously. The main challenge in CL is *catastrophic forgetting*, meaning that their performance on previous tasks degrades after learning new ones (McCloskey & Cohen, 1989; Goodfellow et al., 2013). It has led to a growing body of works focusing on heuristic methods of mitigating forgetting, including regularization-based methods (Kirkpatrick et al., 2017; Aljundi et al., 2018; Li & Hoiem, 2017), replay-based methods (Chaudhry et al., 2019; Lopez-Paz & Ranzato, 2017; Shin et al., 2017), and optimization-based methods (Farajtabar et al., 2020; Javed & White, 2019; Mirzadeh et al., 2020).

As CL is receiving significant attention in practice, it is also important to theoretically understand the mechanism of continual learning. A vast amount of the theoretical works on CL so far has focused on regression problems (Bennani et al., 2020; Doan et al., 2021; Asanuma et al., 2021; Lee et al., 2021; Evron et al., 2022; Goldfarb & Hand, 2023; Li et al., 2023), whereas most of the practical application of deep learning is based on classification. Thus, theoretical analysis of continual classification methods and their learning dynamics is of significant interest and importance. Indeed, a few results study continual classification (Raghavan & Balaprakash, 2021; Kim et al., 2022; 2023; Shi & Wang, 2023), albeit focusing on theoretical perspectives that are different from ours; we review these related works in Appendix A.

This paper is mainly motivated by a recent result studying continual linear classification on a collection of jointly separable datasets (Evron et al., 2023). The authors consider continual training of a linear classifier under weak regularization, where the linear classifier is trained until convergence at every given task. By taking the limit of the regularization coefficient $\lambda \to 0$, this training procedure is shown to be equivalent (in terms of the parameter *direction* as $\lambda \to 0$) to a projection-based scheme called Sequential Max-Margin (SMM): every time we encounter a new binary classification task, we project the current model parameter vector to a convex set defined by the margin conditions of the given dataset. Then, under this framework of projection onto convex sets, the authors show linear convergence of the iterates of SMM to an *offline solution* (i.e., a classifier that solves all tasks at once) under cyclic/random ordering of the tasks. More details can be found in Appendix B.

---

[*]Authors contributed equally to this paper.

In light of the insightful analyses by Evron et al. (2023), we now highlight some aspects of their work that motivate the setup of our interest. First of all, Evron et al. (2023) consider minimizing the regularized training loss of each task *until convergence*; however, it is far more common to spend a finite budget of iterations per task in practice (i.e., online one-pass setting, or fixed-epoch setting). Training until convergence, combined with sending the regularization coefficient $\lambda \to 0$, also raises an issue on the claimed equivalence of weakly regularized training and the projection-based scheme. As $\lambda \to 0$, the solution of the training objective diverges to infinity, which does not match the fact that the iterate of the SMM travels only for a finite distance at every stage.[1] Another noteworthy characteristic of the considered SMM scheme is that it does not always converge to the *offline max-margin solution*, i.e., the hard-margin support vector machine solution that solves *all* tasks jointly, which is known to be beneficial in terms of generalization (Vapnik, 2013). Lastly, in their concluding section, Evron et al. (2023) also suggest studying *unregularized* continual training with *early stopping* and highlight that the behavior may be different. These observations triggered our investigation into a gradient-based algorithm for continual linear classification and its convergence and algorithmic bias.

In this work, we theoretically study continual linear classification via sequentially running gradient descent (GD) on the *unregularized* logistic loss for a fixed budget of iterations at every stage.[2] When all tasks are jointly separable and revealed in the cyclic order (as studied by Evron et al. (2023)), we show that sequential GD converges in the direction of the offline max-margin solution, unlike SMM. We highlight that this is an interesting result for at least two reasons:

- It reveals a clear difference between sequential GD and the projection-based SMM algorithm in terms of algorithmic bias.

- It is well-known that GD applied to an individual task has its implicit bias towards the task's own max-margin direction (Soudry et al., 2018). However, the direction of the offline max-margin solution can largely differ from the max-margin directions of individual tasks, not even lying on the subspace spanned by the individual directions (see Figure 1 and Appendix C.1).

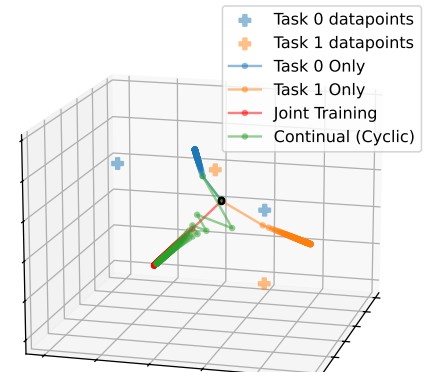

Figure 1: Trajectory of sequential GD on a two-task toy example (Appendix C.1) in which the offline max-margin direction is not on the subspace spanned by individual task max-margin solutions. Sequential GD iterates initially oscillate but quickly start to evolve along the same direction as the offline max-margin direction.

Therefore, the convergence of sequential GD to the *offline max-margin solution* highlights that repeated continual training eventually drives the model to learn all tasks well, overcoming the biases towards individual tasks. In addition to the implicit bias result, we also characterize the convergence rate in terms of total loss and the vanishing rate of the per-cycle forgetting. Our analysis reveals a surprising but intuitive link between positive/negative task alignments and forgetting. Furthermore, we broaden the scope of our analysis to the random task ordering case and a jointly non-separable case. We summarize our main contributions below.

## 1.1 SUMMARY OF CONTRIBUTIONS

We study continual linear classification using *sequential GD*, where the model is updated by $K$ iterations of GD on the unregularized training loss of each given task.

- In Section 3, we study the scenario where the tasks are jointly separable and are given in a cyclic order. We prove that the joint (full) training loss asymptotically converges to zero (Theorem 3.1) and the sequential GD iterates in fact align with the *joint (offline) max-margin* solution (Theorem 3.2). We also provide a non-asymptotic analysis of the *cycle-averaged forgetting*, diminishing at the rate of $\mathcal{O}\left(\frac{\ln^4 J}{J^2}\right)$ (Theorem 3.4), based on a non-asymptotic loss

---

[1]Recall that Evron et al. (2023) show their equivalence in terms of parameter *direction*.

[2]We focus on this setup instead of early stopping because it is closer to common practice in deep learning.

convergence bound of rate $\mathcal{O}(\frac{\ln^2 J}{J})$ which holds for all number of cycles $J > 1$. Our analysis delivers a clear intuition on how positive or negative task alignment has an impact on forgetting.

- In Section 4, we again consider the jointly separable setup but with the tasks given in a uniformly random order. We show that the asymptotic loss convergence and directional convergence to the joint max-margin solution still happen, albeit almost surely (Theorems 4.1 and 4.2).

- Lastly, in Section 5 we consider the case where the tasks are no longer jointly separable, in which the global minimum of the joint training loss uniquely exists. We derive a fast non-asymptotic convergence rate of $\mathcal{O}(\frac{\ln^2 J}{J^2})$ towards the global minimum when the tasks are presented cyclically.

## 2 PROBLEM SETUP

In this section, we outline the problem setup considered throughout the paper.

**Notation.** We denote the joint data matrix as $\boldsymbol{X} \in \mathbb{R}^{d \times N}$, whose columns are the $d$-dimensional data points $\boldsymbol{x}_i$'s. For a square matrix $\boldsymbol{A}$, we denote the maximum/minimum eigenvalue of it by $\lambda_{\max}(\boldsymbol{A})$ and $\lambda_{\min}(\boldsymbol{A})$, respectively. In particular, we write $\sigma_{\max} = \sqrt{\lambda_{\max}(\boldsymbol{X}\boldsymbol{X}^\top)}$ as the maximum singular value of $\boldsymbol{X}$. The Euclidean ($\ell_2$) norm of a vector $\boldsymbol{v}$ is denoted as $\|\boldsymbol{v}\|$. Let $\mathbb{R}_{\geq 0}^N$ be the set of $N$-dimensional vectors whose elements are greater than or equal to zero. Also, for a pair of integers $K_1 \leq K_2$, we write $[K_1 : K_2]$ to denote a set of consecutive integers $\{K_1, K_1 + 1, \ldots, K_2\}$.

### 2.1 CONTINUAL LINEAR BINARY CLASSIFICATION

We consider binary classification, where each input data $\boldsymbol{x} \in \mathbb{R}^d$ has its own label $y \in \{-1, +1\}$. We assume that our learning algorithm encounters $M$ different binary classification **tasks** in a sequential manner, and our goal is to find an **offline solution** that jointly solves all the tasks. Each task consists of a set of data pairs $\{(\boldsymbol{x}_i, y_i)\}_{i \in I_m}$, where $I_m \subset [0 : N - 1]$ is the set of data indices in task $m \in [0 : M - 1]$. Let us use the notation that the total index set $I := [0 : N - 1]$ is partitioned into $I = \biguplus_{m=0}^{M-1} I_m$; thus, we use disjoint index sets for distinct tasks.

We consider a simple linear model $f(\boldsymbol{x}; \boldsymbol{w}) = \boldsymbol{x}^\top \boldsymbol{w}$ parameterized by a weight vector $\boldsymbol{w} \in \mathbb{R}^d$. With a differentiable loss function $\ell(u)$, we aim to minimize **offline (joint) training loss** defined as

$$\mathcal{L}(\boldsymbol{w}) := \sum_{i \in I} \ell\left(y_i f(\boldsymbol{x}_i; \boldsymbol{w})\right) = \sum_{i \in I} \ell\left(y_i \boldsymbol{x}_i^\top \boldsymbol{w}\right).$$

Likewise, the training loss of task $m \in [0 : M - 1]$ is defined as

$$\mathcal{L}_m(\boldsymbol{w}) := \sum_{i \in I_m} \ell\left(y_i \boldsymbol{x}_i^\top \boldsymbol{w}\right).$$

Consider a CL setup run in multiple stages. At each stage $t \geq 0$, the only available dataset for training is taken from a single task $m_t \in [0 : M - 1]$: we denote the corresponding index set at stage $t$ as $I^{(t)} = I_{m_t}$. Note that the learning algorithm does *not* have the freedom to choose the next task; we assume that the task is presented to the algorithm by the "environment." We study two common scenarios that dictate the order of the tasks to be learned:

**(1) Cyclic task ordering.** The tasks are revealed in a predefined cyclic order, i.e., $m_t = t \bmod M$.

**(2) Random task ordering.** Every task is independently sampled uniformly at random, i.e., $m_t \overset{\text{iid}}{\sim} \mathrm{Unif}([0 : M - 1])$.

Both ordering schemes have been studied theoretically and empirically (Evron et al., 2022; 2023; Cossu et al., 2022; Houyon et al., 2023). Indeed, such schemes naturally occur in real-world scenarios. For instance, cyclic task ordering covers search engines influenced by periodic events[3] and seasonal financial data (Gultekin & Gultekin, 1983; Yang et al., 2022). Random task ordering bears a resemblance to autonomous driving in randomly recurring environments (Verwimp et al., 2023).

---

[3]A trend of interest about a keyword, e.g., `trends.google.com/trends/`

## 2.2 ALGORITHM: SEQUENTIAL GRADIENT DESCENT

At every stage $t$, we run GD with a fixed learning rate $\eta > 0$ on the corresponding training loss

$$\mathcal{L}^{(t)}(\boldsymbol{w}) := \mathcal{L}_{m_t}(\boldsymbol{w}) = \sum_{i \in I^{(t)}} \ell\left(y_i \boldsymbol{x}_i^\top \boldsymbol{w}\right) \quad (\because I^{(t)} = I_{m_t}) \tag{1}$$

for a fixed budget ($K \geq 1$). We call this algorithm **sequential GD** and its update rule is written as

$$\boldsymbol{w}_{k+1}^{(t)} = \boldsymbol{w}_k^{(t)} - \eta \nabla \mathcal{L}^{(t)}(\boldsymbol{w}_k^{(t)}) \quad \text{for } k \in [0 : K-1]; \qquad \boldsymbol{w}_0^{(t+1)} = \boldsymbol{w}_K^{(t)}. \tag{2}$$

That is, for the task $m_t$ given at stage $t$, we run $K$ steps of GD updates and move on to the next task by setting the initial iterate of the next stage $\boldsymbol{w}_0^{(t+1)}$ as the last iterate of the current stage $\boldsymbol{w}_K^{(t)}$.

## 3 CYCLIC LEARNING OF JOINTLY SEPARABLE TASKS

In this section, we focus on the *jointly linearly separable* datasets (Evron et al., 2023). We dive deep into the case of cyclic task ordering and prove that sequential GD on separable linear classification tasks converges in a direction to the offline max-margin solution of the total dataset. Additionally, through a non-asymptotic analysis of the loss convergence, we also characterize the average forgetting within cycles and show that the forgetting vanishes to zero faster than the loss convergence.

### 3.1 DEFINITIONS AND ASSUMPTIONS

The first assumption is that the total dataset is linearly separable.

**Assumption 3.1** (Joint Separability). There exists $\boldsymbol{w} \in \mathbb{R}^d$ such that $y_i \boldsymbol{x}_i^\top \boldsymbol{w} > 0$ for $\forall i \in I$.

Under Assumption 3.1, we can state an important definition central to our analysis. We define the **joint (offline) $\ell_2$ max-margin solution** (where we usually omit "$\ell_2$" for convenience)

$$\hat{\boldsymbol{w}} := \underset{\boldsymbol{w} \in \mathbb{R}^d}{\arg\min} \|\boldsymbol{w}\|^2 \quad \text{subject to } y_i \boldsymbol{x}_i^\top \boldsymbol{w} \geq 1, \; \forall i \in I. \tag{3}$$

It can be shown that the optimization problem in Equation (3) has a unique solution $\hat{\boldsymbol{w}}$ (Mohri et al., 2018). The max-margin solution is of key interest in the study of linear classification, because it is well known that it has a good generalization guarantee (Vapnik, 2013) and GD applied to a single separable binary classification problem has an implicit bias towards its $\ell_2$ max-margin solution (Soudry et al., 2018). To be more specific, it is shown in Soudry et al. (2018) that the norm of GD iterates diverges to infinity, but their direction converges to $\frac{\hat{\boldsymbol{w}}}{\|\hat{\boldsymbol{w}}\|}$. In our CL setting, we consider running multiple steps of GD on a single task before switching to a different task, and still aim to find the joint max-margin solution that solves all tasks at once.

Given the definition of joint max-margin solution, we now define several key quantities. The **maximum margin** of (normalized) $\hat{\boldsymbol{w}}$ is defined as

$$\phi := \min_{i \in I} \frac{y_i \boldsymbol{x}_i^\top \hat{\boldsymbol{w}}}{\|\hat{\boldsymbol{w}}\|}. \tag{4}$$

In fact, it can be shown that $\phi = 1/\|\hat{\boldsymbol{w}}\|$. A **support vector** is a data point $\boldsymbol{x}_i$ that attains this minimum $\phi$; we define the index set of support vectors as $S := \{i \in I : y_i \boldsymbol{x}_i^\top \frac{\hat{\boldsymbol{w}}}{\|\hat{\boldsymbol{w}}\|} = \phi\}$, and define the index sets of support vectors of each task $S_m := S \cap I_m$ for all $m \in [0 : M-1]$. Let the support vector matrix be $\boldsymbol{X}_S \in \mathbb{R}^{d \times |S|}$, a submatrix of the data matrix $\boldsymbol{X}$ that only contains columns corresponding to support vectors. Lastly, we define the **second margin** $\theta := \min_{i \in I \setminus S} y_i \boldsymbol{x}_i^\top \hat{\boldsymbol{w}} > 1$, which will appear in our techincal analyses.

To show directional convergence to the joint max-margin solution (Theorem 3.2), we additionally pose a mild assumption on the support vectors.

**Assumption 3.2** (Non-Degeneracy of Support Vectors, e.g., Soudry et al., 2018). For each $i \in S$, there exists a unique $\alpha_i > 0$ satisfying $\hat{\boldsymbol{w}} = \sum_{i \in S} \alpha_i \cdot y_i \boldsymbol{x}_i$: i.e., no support vectors are degenerate.

It is worth mentioning that Assumption 3.2 is valid for almost all linearly separable datasets sampled from a continuous distribution where no more than $d$ support vectors can be on the same hyperplane.

In the upcoming sections, we present several results on convergence, implicit bias, and forgetting of sequential GD. They rely on different assumptions on $\ell(u)$; we collect them here. It is noteworthy that the renowned *logistic loss* $\ell(u) = \ln(1 + e^{-u})$ satisfies all the assumptions listed below.

**Assumption 3.3.** The loss function $\ell(u)$ is positive, $\beta$-smooth, monotonically decreasing to zero (i.e., $\lim_{u \to +\infty} \ell(u) = 0$), and satisfies $\limsup_{u \to -\infty} \ell'(u) < 0$.

**Assumption 3.4** (Tight Exponential Tail, Soudry et al., 2018). The negative loss derivative $-\ell'(u)$ has a tight exponential tail, i.e., there exist positive constants $\mu_+, \mu_-$, and $\bar{u}$ such that for all $u > \bar{u}$:

$$(1 - \exp(-\mu_- u))e^{-u} \leq -\ell'(u) \leq (1 + \exp(-\mu_+ u))e^{-u}.$$

**Assumption 3.5** (Convexity). The loss $\ell(u)$ is convex, i.e., $\ell(u) \geq \ell(v) + \ell'(v) \cdot (u - v)$ $(u, v \in \mathbb{R})$.

### 3.2 ASYMPTOTIC RESULTS: LOSS CONVERGENCE & IMPLICIT BIAS TO JOINT MAX-MARGIN

Now, we analyze the asymptotic convergence of offline training loss and characterize the directional convergence of sequential GD (2) on jointly separable cyclic tasks. We start by understanding the asymptotic behavior of the joint task loss $\mathcal{L}(\boldsymbol{w})$.

**Theorem 3.1.** Let $\{\boldsymbol{w}_k^{(t)}\}_{k \in [0:K-1], t \geq 0}$ be the sequence of GD iterates (2) from any starting point $\boldsymbol{w}_0^{(0)}$, where tasks are given cyclically. Under Assumptions 3.1 and 3.3, if the learning rate satisfies $\eta < \frac{\phi^2}{2K\beta\sigma_{\max}^3(M\phi + \sigma_{\max})}$, then

1. Loss converges to zero: $\lim_{t \to \infty} \mathcal{L}(\boldsymbol{w}_k^{(t)}) = 0, \forall k \in [0 : K - 1]$.

2. All data points are eventually classified correctly: $\lim_{t \to \infty} y_i \boldsymbol{x}_i^\top \boldsymbol{w}_k^{(t)} = \infty, \forall k \in [0 : K - 1], i \in I$.

3. Square sum of the change of weight is finite: $\sum_{t=0}^\infty \sum_{k=0}^{K-1} \|\boldsymbol{w}_{k+1}^{(t)} - \boldsymbol{w}_k^{(t)}\|^2 < \infty$.

The theorem above shows that cyclic continual learning on the jointly separable data will eventually learn all tasks, or equivalently, find an offline solution without any additional techniques such as regularization. This result matches the recent empirical findings that DNN can mitigate catastrophic forgetting when tasks are given repetitively (Lesort et al., 2023). The last part on the sum of squared changes is used for proving the upcoming Theorem 3.2. We note that Theorem 3.1 does not require convexity of $\ell$. The proof can be found in Appendix E.1.

Although Theorem 3.1 shows that the joint training loss converges to zero, due to the joint separability (Assumption 3.1), there are multiple directions in which $\boldsymbol{w}_k^{(t)}$ could evolve to make the offline training loss decay to zero. That is, the loss convergence only guarantees finding *an* offline solution, but does not characterize *which*. Under additional assumptions of non-degeneracy and tight exponential tails, we characterize *which* direction $\boldsymbol{w}_k^{(t)}$ diverges to, and show that the model parameter in fact aligns with the joint $\ell_2$ max-margin solution $\hat{\boldsymbol{w}}$ (3).

**Theorem 3.2.** Let $\{\boldsymbol{w}_k^{(t)}\}_{k \in [0:K-1], t \geq 0}$ be the sequence of GD iterates (2) from any initial point $\boldsymbol{w}_0^{(0)}$, where tasks are given cyclically. Suppose that Assumptions 3.1, 3.2, 3.3, and 3.4 hold. Then, under the same learning rate condition as in Theorem 3.1, $\boldsymbol{w}_k^{(t)}$ will behave as

$$\boldsymbol{w}_k^{(t)} = \ln(t)\hat{\boldsymbol{w}} + \boldsymbol{\rho}_k^{(t)},$$

for all sufficiently large $t$, where $\|\boldsymbol{\rho}_k^{(t)}\|$ stays bounded as $t \to \infty$.

The proof is in Appendix E.2. A key implication of Theorem 3.2 is that the weight vector converges in the direction of the joint max-margin solution while diverging in magnitude at a rate $\mathcal{O}(\ln t)$:

$$\lim_{t \to \infty} \frac{\boldsymbol{w}_k^{(t)}}{\|\boldsymbol{w}_k^{(t)}\|} = \frac{\hat{\boldsymbol{w}}}{\|\hat{\boldsymbol{w}}\|}, \quad \forall k \in [0 : K - 1]. \tag{5}$$

It implies that sequential GD without any regularization not only learns every task, but also *implicitly biased* towards the joint max-margin direction whose corresponding linear classifier separates the total dataset with the largest margin, even without full access to it at all update steps of the classifier weight. This implication is visually verified in Figure 2.

**Remark 3.6** (Asymptotic Convergence Rate of Joint Training Loss). Recall that Theorem 3.2 relies on Assumption 3.2 as well as the assumptions about the shape of the loss function $\ell(\cdot)$, namely,

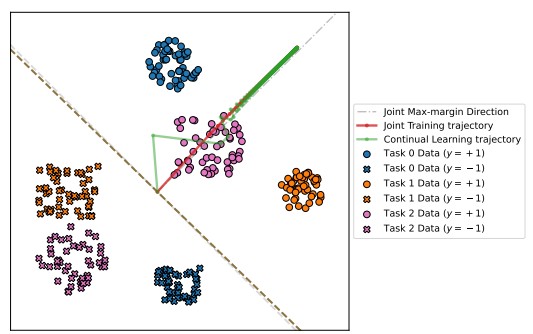 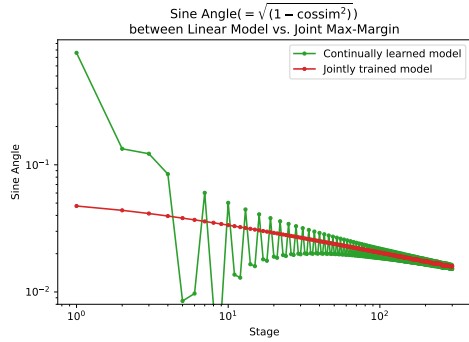

(a) Data points, training trajectory (beginning-of-stage iterates), and the decision boundaries (dashed).

(b) Sine angles, implying the implicit bias toward joint max-margin direction.

Figure 2: **Comparison between continually learned and jointly trained linear classifier.** We generate three jointly separable binary classification tasks (with 2D inputs) and run (1) sequential GD in a cyclic task ordering and (2) full-batch GD. It is well-known that the offline full-batch GD converges to the offline $\ell_2$ max-margin solution (Soudry et al., 2018). We verify a similar implicit bias of sequential GD iterates (which we proved in Theorem 3.2) by observing the decrease in angle between the model weight and the joint max-margin direction (set as $(1, 1)$). We also observe similar phenomena for a more general experimental setup (e.g., random task ordering): see Appendix C.2.

Assumptions 3.3 and 3.4. Combining these assumptions again with the theorem, we can show that the $\mathcal{L}(\boldsymbol{w}_k^{(t)})$ will eventually converge to zero at a rate $\mathcal{O}(1/t)$, given that $t$ is *sufficiently* large. We prove this claim in Appendix E.3. Moreover, we will compare it with a forthcoming non-asymptotic loss convergence rate (Theorem 3.3) later in Section 3.3.

**Remark on Assumption 3.2.** As noted earlier, the non-degeneracy assumption (Assumption 3.2) is borrowed from Soudry et al. (2018); the purpose of adopting this assumption is to facilitate a more complete analysis of the residual $\boldsymbol{\rho}_k^{(t)}$. In fact, in Soudry et al. (2018), the conclusion on the directional convergence (similar to Equation (5), but for single-task GD training) still holds even under the absence of Assumption 3.2. In light of this, we also believe that directional convergence of sequential GD (5) will hold even without Assumption 3.2, but we did not pursue removing the assumption because it does not offer substantial additional insights.

**Additional Experiments.** Although we analyze continual learning in a setting where each task has a fixed dataset, the insight of our analysis extends to general setups. To show this, we conduct experiments in a setting where each task has its own (separable) data distribution and a dataset is freshly sampled at every new stage. We observe the same directional convergence behavior of sequential GD toward the true joint max-margin direction. The detailed results are in Appendix C.2.4. In addition, we provide experiments with shallow ReLU networks, verifying analogous insights on implicit bias and loss convergence of continually learned models: see Appendix C.4.

### 3.3 Non-Asymptotic Results: Loss Convergence and Forgetting Bounds

In Section 3.2, we presented asymptotic results characterizing the convergence of total training loss to zero and the directional convergence of sequential GD iterates to the max-margin solutions. We now supplement these results with an additional *non-asymptotic* convergence analysis on total training loss, which we can use to obtain a non-asymptotic analysis of *cycle-averaged forgetting* as well.

As aforementioned, the main challenge in CL is mitigating catastrophic forgetting. Analyses of continual learning methods aim to show that methods decrease forgetting, theoretically or empirically. In this paper, we are interested in how strong forgetting is in our continual linear classification setup.

We start by stating a common definition of forgetting, which quantifies the amount of loss increase at the end of stage $t$ compared to the end of $K$ steps of GD on $\mathcal{L}^{(s)}$ executed in stage $s \le t$.

**Definition 3.7** (Forgetting). The **forgetting** at stage $t$ of the task learned in stage $s$ ($\le t$) is the change of the task loss $\mathcal{L}^{(s)}$ from the moment the $K$ GD steps were finished in stage $s$. That is,

$$\mathcal{F}^{(s)}(t) := \mathcal{L}^{(s)}(\boldsymbol{w}_K^{(t)}) - \mathcal{L}^{(s)}(\boldsymbol{w}_K^{(s)}).$$

Notice that forgetting is zero by definition when $t = s$. While it is usually expected that forgetting is a positive quantity, it could also be negative by definition. Such a case can happen when the tasks seen in stages between $s$ and $t$ are well-aligned with $\mathcal{L}^{(s)}$, so that the model improves on the task previously seen in stage $s$. This phenomenon is called *backward knowledge transfer*.

When CL tasks do not necessarily repeat, it is common to evaluate the average forgetting that occurred during all past stages, namely $\frac{1}{t} \sum_{s=0}^{t-1} \mathcal{F}^{(s)}(t)$. However, since we consider the case where tasks are given cyclically, it is natural to define our quantity of interest as below:

**Definition 3.8** (Cycle-Averaged Forgetting). The **cycle-averaged forgetting** at cycle $j$ is the average loss change of previous tasks from the stage in which it was learned. That is,

$$\mathcal{F}_{\mathrm{cyc}}(j) := \frac{1}{M} \sum_{m=0}^{M-1} \mathcal{F}^{(Mj+m)}(Mj + M - 1)$$

$$= \frac{1}{M} \sum_{m=0}^{M-1} \mathcal{L}_m(\boldsymbol{w}_0^{(Mj+M)}) - \mathcal{L}_m(\boldsymbol{w}_K^{(Mj+m)}).$$

By studying cycle-averaged forgetting, we would like to understand how much forgetting happens during the cyclic learning process, and how the amount of forgetting changes as we repeat the cycles.

Although the asymptotic convergence to joint max-margin solution (Theorem 3.2) suggests that the model will suffer a diminishing level of forgetting in the long run, characterizing the amount of forgetting for a given cycle count $J$ necessitates a more careful non-asymptotic analysis of the loss convergence. For this purpose, we present an additional theorem characterizing the non-asymptotic convergence of offline training loss $\mathcal{L}$; we then build on this theorem to prove upper and lower bounds on cycle-averaged forgetting. The new convergence theorem requires the same set of assumptions as Theorem 3.1, except for an additional assumption of convex $\ell(u)$.

**Theorem 3.3.** *Suppose that Assumptions 3.1, 3.3, and 3.5 hold under the cyclic task ordering setup. Then, if we choose a learning rate satisfying $\eta < \frac{\phi^2}{4K\beta\sigma_{\max}^3(M\phi+\sigma_{\max})}$, we have for any $m \in [0 : M-1]$ and $k \in [0 : K-1]$ that*

$$\mathcal{L}(\boldsymbol{w}_k^{(MJ+m)}) \leq \left( |S| + \frac{\sum_{i=0}^{m-1} |S_i| + \frac{k}{K} |S_m|}{J} \right) \ell(\ln MJ) + \frac{\|\boldsymbol{w}_0^{(0)} - \hat{\boldsymbol{w}} \ln MJ\|^2}{2\eta KJ} + \frac{D_1}{J}$$

$$+ \left( |I| - |S| + \frac{\sum_{i=0}^{m-1}(|I_i| - |S_i|) + \frac{k}{K}(|I_m| - |S_m|)}{J} \right) \ell(\theta \ln MJ),$$

*where $\theta > 1$ is the second margin defined in Section 3.1, and $D_1$ is a constant independent to $J$.*

The proof, as well as the detailed expression of $D_1$, can be found in Appendix E.4. One can revisit Section 3.1 to recall the definitions of symbols such as $\sigma_{\max}$, $\phi$, and $\beta$. The bound in Theorem 3.3 may be a bit difficult to parse. First of all, notice that whenever $\ell(u) \leq e^{-u}$, which is true for logistic loss $\ell(u) = \ln(1 + e^{-u})$, we have $\ell(\ln MJ) \leq \frac{1}{MJ}$ and $\ell(\theta \ln MJ) \leq \frac{1}{(MJ)^\theta}$. Combined with other terms, this implies an overall $\mathcal{O}(\frac{\ln^2 J}{J})$ upper bound for the offline training loss.

Unlike the asymptotic result, Theorem 3.3 enables capturing the behavior of training for any small $J$. we can notice for any fixed $J$, the upper bound in fact *grows* with $k$ and $m$. This unusual growth of the upper bound reflects the effect of forgetting that can happen within cycles. We demonstrate this mid-cycle increase of joint loss using a toy example in Appendix C.3. Moreover, despite the possible forgetting, Theorem 3.3 indicates that if tasks are given cyclically, then the loss bound is guaranteed to decrease at the end of every cycle.

We can now use Theorem 3.3 to derive bounds on cycle-averaged forgetting we defined in Definition 3.8. We characterize how fast the cycle-averaged forgetting $\mathcal{F}_{\mathrm{cyc}}(J)$ converges to zero as the cycles replay. For this theorem, we specifically consider logistic loss, which satisfies all the assumptions about loss in the paper.

**Theorem 3.4.** *Suppose Assumption 3.1 and let $\ell(u) = \ln(1 + e^{-u})$ be the logistic loss, which satisfies Assumptions 3.3, 3.4, and 3.5. If the learning rate satisfies $\eta < \frac{\phi^2}{4K\beta\sigma_{\max}^3(M\phi+\sigma_{\max})}$, then*

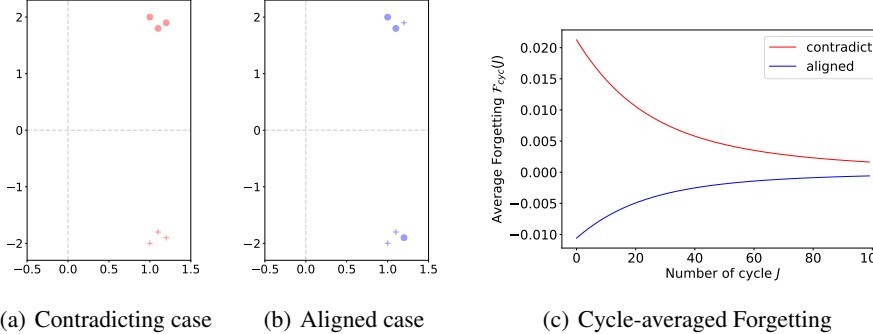

(a) Contradicting case      (b) Aligned case      (c) Cycle-averaged Forgetting

Figure 3: We compare two continual learning scenarios with the same joint dataset $\mathcal{D} = \{(1,2),(1.1,1.8),(1.2,1.9),(1,-2),(1.1,-1.8),(1.2,-1.9)\}$, where labels are all $+1$ and hence omitted. We mark Task 1's data as '$\circ$' and Task 2's data as '$+$'. We used $M = 2$ and $K = 10$. Figure 3(a) displays a data composition that makes large $A_{1,2}^-$, whereas Figure 3(b) displays a data composition that makes relatively small $A_{1,2}^-$ and large $A_{1,2}^+$. Figure 3(c) is a plot of cycle-averaged forgetting ($\mathcal{F}_{\text{cyc}}$), evolving over cycles. For the "contradict" scenario (red), $\mathcal{F}_{\text{cyc}}$ is always positive and diminishing to 0. In contrast, for the "aligned" scenario (blue), $\mathcal{F}_{\text{cyc}}$ is always negative and rising to 0.

*the cycle-averaged forgetting $\mathcal{F}_{\text{cyc}}(J)$ for cycle $J$ satisfies the following upper and lower bounds:*

$$-\eta K \cdot \{L(J)\}^2 \cdot \frac{\sum_{p \neq q} A_{p,q}^+}{M} \leq \mathcal{F}_{\text{cyc}}(J) \leq \eta K \cdot \{L(J)\}^2 \cdot \frac{\sum_{p \neq q} A_{p,q}^-}{M},$$

*where $L(J) = \mathcal{O}\left(\frac{\ln^2 J}{J}\right)$ and $A_{p,q}^+ := \sum_{\substack{(i,j) \in I_p \times I_q \\ \boldsymbol{x}_i^\top \boldsymbol{x}_j > 0}} \boldsymbol{x}_i^\top \boldsymbol{x}_j, \ A_{p,q}^- := \sum_{\substack{(i,j) \in I_p \times I_q \\ \boldsymbol{x}_i^\top \boldsymbol{x}_j < 0}} \left| \boldsymbol{x}_i^\top \boldsymbol{x}_j \right|.$*

The proof and the detailed expressions are in Appendix E.5. Theorem 3.4 shows a nonnegative upper bound and a nonpositive lower bound on the cycle-averaged forgetting at cycle $J$. Note that both upper and lower bounds decay to zero as $J$ grows. Convergence of $\mathcal{F}_{\text{cyc}}(J)$ is of rate $\mathcal{O}(\frac{\ln^4 J}{J^2})$, which is faster than the convergence rate $\mathcal{O}(\frac{\ln^2 J}{J})$ of joint training loss shown in Theorem 3.3.

The bounds in Theorem 3.4 reflect how positive/negative data alignment between different tasks impact forgetting. The quantities $A_{p,q}^+$ and $A_{p,q}^-$ capture show how similar and different (respectively) data points are, for a pair of tasks $(p, q)$. In particular, if $\sum_{p \neq q} A_{p,q}^- = 0$, it is guaranteed that (non-negative) cycle-averaged forgetting will not happen regardless of $J$. Rather, training on a task will decrease the loss for all previously learned tasks, which can be considered an extreme form of *backward knowledge transfer*. On the other hand, if $\sum_{p \neq q} A_{p,q}^+ = 0$, it is guaranteed that the model will suffer forgetting at every cycle; however, even in this case, Theorem 3.4 implies that repeating tasks over cycles mitigates catastrophic forgetting.

Even when the joint dataset $\mathcal{D}$ is the same, forgetting behavior can differ depending on how the data points are distributed over different tasks. This matches the former theoretical explanation of how distribution affects forgetting. For instance, Lin et al. (2023) show that a larger distance between each task's optimal solution leads to larger forgetting. For a straightforward interpretation, consider the following example of two tasks: their cycle-averaged forgetting for two different decompositions of $\mathcal{D}$ is plotted in Figure 3. We can observe that two tasks contradicting each other (i.e., large $A_{1,2}^-$) result in positive forgetting, whereas two tasks aligning better (i.e., large $A_{1,2}^+$) exhibit negative forgetting. Nevertheless, cycle-averaged forgetting converges to zero in both cases.

**Remark 3.9.** When $\ell(u)$ is the logistic loss, a direct application of an asymptotic loss convergence rate mentioned in Remark 3.6 implies the forgetting convergence of rate $\mathcal{O}(\frac{1}{J^2})$ without any additional logarithmic factors, which can be shown by almost an identical proof as Theorem 3.4. However, it is guaranteed for a sufficiently large number of cycles $J$. See Appendix E.3 for discussion about it.

## 4    RANDOM-ORDER LEARNING OF JOINTLY SEPARABLE TASKS

In this section, we consider the scenario where tasks are given in a random order, while still assuming that the tasks are jointly separable. Formally, at the end of the $K$-th GD iteration of stage $t$, the next task is sampled independently and uniformly at random. Even in this case, our analysis reveals that the asymptotic results shown in Section 3.2 continue to hold *almost surely*.

We first show that the offline training loss converges to zero almost surely, which is a random-order counterpart of Theorem 3.1. The proof is in Appendix F.1.

**Theorem 4.1.** *Let $\{w_k^{(t)}\}_{k\in[0:K-1],t\geq0}$ be the sequence of GD iterates* (2) *from any starting point $w_0^{(0)}$, where tasks are given randomly. Under Assumptions 3.1 and 3.3, if the learning rate satisfies $\eta < \frac{2\phi^2}{\beta\sigma_{\max}^4}$, then the following statements hold with probability 1:*

*1. Loss converges to zero:* $\lim\limits_{t\to\infty}\mathcal{L}(w_k^{(t)}) = 0, \forall k \in [0 : K - 1].$

*2. All data points are eventually classified correctly:* $\lim\limits_{t\to\infty}y_i x_i^\top w_k^{(t)} = 0, \forall k \in [0 : K-1], i \in I.$

*3. Square sum of the change of weight is finite:* $\sum_{t=0}^\infty \sum_{k=0}^{K-1} \|w_{k+1}^{(t)} - w_k^{(t)}\|^2 < \infty.$

We derive the same asymptotic loss convergence result, with a minor difference that the learning rate can be chosen independently of the number of tasks $M$ and the iteration count $K$.

We now state the random-order counterpart of Theorem 3.2, which implies that the sequential GD iterates converge to the joint $\ell_2$ max-margin solution almost surely. The proof is in Appendix F.2.

**Theorem 4.2.** *Let $\{w_k^{(t)}\}_{k\in[0:K-1],t\geq0}$ be the sequence of GD iterates* (2) *from any starting point $w_0^{(0)}$, where tasks are given randomly. Suppose that Assumptions 3.1, 3.2, 3.3, and 3.4 hold. Then, under the same learning rate condition as in Theorem 4.1, $w_k^{(t)}$ will behave as:*

$$w_k^{(t)} = \ln(t)\,\hat{w} + \rho_k^{(t)},$$

*where $\|\rho_k^{(t)}\|$ stays bounded as $t$ grows.*

## 5    BEYOND JOINTLY SEPARABLE TASKS

Now we turn our attention to the CL on a strictly *non-separable* set of $M$ tasks, where the tasks are presented in a *cyclic* manner. In this section, we assume that the set of all data points spans the whole space $\mathbb{R}^d$ without loss of generality. This is a mild assumption because every gradient update happens in the span of data points. In this case, if we assume the strict non-separability on the full dataset (see Assumption 5.1), the offline training loss $\mathcal{L}(w) = \sum_{m=0}^{M-1}\mathcal{L}_m(w)$ defined with logistic losses becomes strictly convex and coercive (i.e., $\lim_{\|w\|\to\infty}\mathcal{L}(w) = +\infty$); thus, it has a unique minimum $w_\star \in \mathbb{R}^d$. We show that, under cyclic task ordering, the iterates of sequential GD converge to $w_\star$ with a rate $\mathcal{O}(\frac{\ln^2 J}{J^2})$, which is faster than the loss convergence rate of the separable case.

The core idea of the analysis is to identify the local strong convexity of the offline training loss on a compact set on which every end-of-cycle iterates lie (Freund et al., 2018). To this end, we introduce a strict non-separability of the joint dataset as defined below.

**Assumption 5.1** (Joint Strict Non-Separability Condition (Freund et al., 2018))**.** Assume that the whole collection of data points is of full rank: $\text{span}(\{x_i : i = 0, \dots, N-1\}) = \mathbb{R}^d$. Additionally, assume that there exists $b > 0$ defined as

$$b := \min_{v\in\mathbb{R}^d:\|v\|=1}\sum_{i=0}^{N-1}[y_i x_i^\top v]^-,$$

where $[a]^- := \max\{0, -a\}$.

Note that a large $b$ means that the joint data points are highly non-separable: for any classifier vector $v$, there exist some data points with the incorrect prediction of the label with a large margin. We also remark that individual tasks are not necessarily strictly non-separable. Hence, our analysis covers the case where all individual tasks are separable but the full dataset is not separable.

We additionally assume some mild properties of the loss function $\ell(\cdot)$.

**Assumption 5.2.** The loss function $\ell : \mathbb{R} \to \mathbb{R}_+$ is a strictly convex, $\beta$-smooth function with a positive second derivative such that $\ell(u) \geq G \cdot [u]^-$ for some $G > 0$.

Note that the logistic loss $\ell(u) = \ln(1+e^{-u})$ satisfies the assumption above with $\beta = 1/4$ and $G = 1$. From the assumptions, we have that (1) the training loss of $m$-th task $\mathcal{L}_m(\boldsymbol{w}) = \sum_{i \in I_m} \ell(y_i \boldsymbol{x}_i^\top \boldsymbol{w})$ is convex and $\beta_m$-smooth for $\beta_m := \beta \lambda_{\max} (\boldsymbol{X}_m \boldsymbol{X}_m^\top)$, where $\boldsymbol{X}_m \in \mathbb{R}^{d \times |I_m|}$ is a data matrix of task $m$ consisting of columns $\{\boldsymbol{x}_i : i \in I_m\}$; (2) due to the strict non-separability, the offline training loss $\mathcal{L}(\boldsymbol{w}) = \sum_{m=0}^{M-1} \mathcal{L}_m(\boldsymbol{w})$ has a unique minimum $\boldsymbol{w}_\star$. Furthermore, we can prove that the end-of-cycle iterates of the sequential GD stay bounded in a compact set $\mathcal{W}$ around $\boldsymbol{w}_\star$. Consequently, we obtain local strong convexity of the offline training loss on $\mathcal{W}$. The proof is in Appendix G.1.

**Lemma 5.1.** *Consider learning $M$ linear classification tasks cyclically. Suppose that Assumptions 5.1 and 5.2 hold. Let $B := \sum_{m=0}^{M-1} \beta_m$ and $V_\star := \sum_{m=0}^{M-1} \frac{1}{\beta_m} \|\nabla \mathcal{L}_m(\boldsymbol{w}_\star)\|^2$. Take a step size $\eta \leq \frac{1}{2\sqrt{2}KB}$. Then, there exists a compact set $\mathcal{W} \subset \mathbb{R}^d$ containing $\boldsymbol{w}_\star$ and every $\boldsymbol{w}_0^{(jM)}$ $(j = 0, 1, 2, \ldots)$, whose radius is independent of $J$ (the number of cycles) but depends on other parameters like $b$, $G$, $B$, and $V_\star$. Also, the offline training loss $\mathcal{L}$ is $\mu$-strongly convex on $\mathcal{W}$, where*

$$\mu := \left( \min_{i \in [0:N-1], \boldsymbol{w} \in \mathcal{W}} \ell'' \left( y_i \boldsymbol{x}_i^\top \boldsymbol{w} \right) \right) \cdot \lambda_{\min} \left( \boldsymbol{X} \boldsymbol{X}^\top \right) > 0. \tag{6}$$

We remark that the radius of the set $\mathcal{W}$ largely depends on the non-separability $b$ (Assumption 5.1): loosely speaking, $\mathcal{W}$ can be arbitrarily large if $b$ goes to zero since $\|\boldsymbol{w} - \boldsymbol{w}_\star\| = \mathcal{O}(1/b)$ for any $\boldsymbol{w} \in \mathcal{W}$. In particular, for the logistic loss $\ell$, the local strong convexity coefficient $\mu$ can get small if $b$ is small, because of the (possibly) large radius of $\mathcal{W}$. With the local strong convexity, we finally have a fast non-asymptotic convergence rate of $\tilde{\mathcal{O}}(J^{-2})$ towards the global minimum. The proof can be found in Appendix G.2.

**Theorem 5.2.** *Suppose we learn $M$ tasks cyclically for $J > 1$ cycles. We adopt the notation from Lemma 5.1. Then, with a step size $\eta = \mathcal{O}\left( \frac{1}{K} \cdot \min\left\{ \frac{1}{B}, \frac{\ln(J)}{J} \right\} \right)$, sequential GD satisfies*

$$\left\| \boldsymbol{w}_0^{(MJ)} - \boldsymbol{w}_\star \right\|^2 \leq \mathcal{O}\left( \exp\left( -\frac{\mu J}{(1+2\sqrt{2})B} \right) \cdot \left\| \boldsymbol{w}_0^{(0)} - \boldsymbol{w}_\star \right\|^2 + \frac{B^2 V_\star \ln^2 J}{\mu^3 J^2} \right). \tag{7}$$

**Remark on the Loss Convergence Rate.** Since the $\mathcal{L}(\boldsymbol{w})$ is $B$-smooth, it satisfies that

$$\mathcal{L}(\boldsymbol{w}) - \mathcal{L}(\boldsymbol{w}_\star) \leq \langle \nabla \mathcal{L}(\boldsymbol{w}_\star), \boldsymbol{w} - \boldsymbol{w}_\star \rangle + \frac{B}{2} \|\boldsymbol{w} - \boldsymbol{w}_\star\|^2 = \frac{B}{2} \|\boldsymbol{w} - \boldsymbol{w}_\star\|^2. \tag{8}$$

Thus, our Theorem 5.2 naturally implies the loss convergence at the same rate (in terms of $J$).

**Experiments on a Real-World Dataset.** For those interested, we also provide an experiment on a real-world dataset CIFAR-10 (Krizhevsky, 2009), which is not guaranteed to be linearly separable: see Appendix C.5.

## 6 CONCLUSION

We considered continual linear classification by running gradient descent for a fixed number of iterations per task. When there exist solutions that can solve every task, we found that even without any regularization or CL methods, the classifier eventually converges to the joint max-margin direction. This implicit bias was shown to exist in both cyclic/random task ordering. We further presented a non-asymptotic analysis on cycle-averaged forgetting with respect to positive/negative alignments of tasks and the number of cycles. Lastly, we showed that if no linear classifier solves all tasks simultaneously, the model converges to the unique minimum of the offline training loss.

**Future Works.** We believe the convergence on continual classification can be extended to other model structures, bridging the gap between empirical findings and theoretical understanding of the impact of task repetition. Also, our results are restricted to the "small learning rate" regime, and do not cover larger learning rates or even the "edge of stability" regime (Wu et al., 2024); relaxing this restriction is left for future work.

ACKNOWLEDGMENTS

This work was partly supported by the Institute for Information & communications Technology Planning & Evaluation (IITP) grants funded by the Korean government (MSIT) (No. RS-2019-II190075, Artificial Intelligence Graduate School Program (KAIST); No. RS-2022-II220184, Development and Study of AI Technologies to Inexpensively Conform to Evolving Policy on Ethics; No. RS-2019-II191906, Artificial Intelligence Graduate School Program (POSTECH)). This research was supported in part by the NAVER-Intel Co-Lab. The work was conducted by KAIST and reviewed by both NAVER and Intel.

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

CONTENTS

## A    OTHER RELATED WORKS

**Theoretical Results on Continual Learning.**    Continual regression has been studied under various regimes and algorithms. The work by Lin et al. (2023) analyzes empirical and population risks by drawing Gaussian samples for each task. It also investigates the impact of overparameterization and task similarity on forgetting. Bennani et al. (2020); Doan et al. (2021); Karakida & Akaho (2022) study forgetting in NTK regime. Specifically, Bennani et al. (2020); Doan et al. (2021) analyze forgetting of orthogonal GD (OGD, Farajtabar et al., 2020), while Karakida & Akaho (2022) studies continual transfer learning. Other settings, such as teacher-student setup (Lee et al., 2021), and feature extraction (Peng & Risteski, 2022) have been considered in task-incremental learning.

On the other hand, most of the works on continual classification focus on analyzing the performance of the model without assuming its architecture. Instead, they focus on the specific types of continual learning (Kim et al., 2022; Shi & Wang, 2023). In class-incremental learning (CIL), Kim et al. (2022) prove that attaining strong within-task prediction (WP) and strong task-id prediction (TP) are necessary and sufficient for achieving strong CIL. Furthermore, they relate TP to the out-of-distribution detection problem. On the other hand, Shi & Wang (2023) consider the domain-incremental learning setup. They especially suggest a framework with a memory buffer that unifies earlier methods. The work by Raghavan & Balaprakash (2021) studies the generalization-forgetting trade-off by viewing it as a two-player sequential game. However, to the best of our knowledge, no other work—apart from Evron et al. (2023)—considers specific model architectures and analyzes the behavior of specific algorithms as in continual regression.

**Implicit Bias of GD for Linear Classification.**    As far as we know, the work by Soudry et al. (2018) is the first to prove the implicit bias of GD towards the max-margin direction for a separable linear classification problem. Another work by Ji & Telgarsky (2018) also considers the data that are not necessarily (strictly) separable, yet the GD iterate may diverge to infinity, although the convergence rate is slower due to the absence of the non-degeneracy condition. The work by Nacson et al. (2019) shows a similar implicit bias result for the *stochastic* GD. Ji & Telgarsky (2021) show a faster convergence rate under decreasing learning rate via a primal-dual analysis. Very recently, it has been proved by Wu et al. (2024) that GD on logistic loss eventually converges to the max-margin direction even when the learning rate is large, which contrasts with the existing findings requiring small enough learning rates.

## B   BRIEF OVERVIEW OF EVRON ET AL. (2023) AND COMPARISONS

To highlight how our sequential GD algorithm differs from Evron et al. (2023), we briefly summarize the Sequential Max-Margin (SMM) framework considered in the existing paper and its theoretical results.

Evron et al. (2023) consider minimizing the regularized training loss of each task until convergence, where the loss function is chosen to be the exponential loss $\ell(u) = \exp(-u)$. Let $\left\{ w_\lambda^{(t)} \right\}_t$ be the iterates trained by regularized continual learning with regularization coefficient $\lambda$. The algorithm can be written as follows:

$$w_\lambda^{(t+1)} = \arg \min_{w \in \mathbb{R}^d} \sum_{i \in I^{(t)}} \exp\left( -y_i x_i^\top w \right) + \frac{\lambda}{2} \left\| w - w_\lambda^{(t)} \right\|^2. \tag{9}$$

Also, let $w_{\mathrm{SMM}}^{(t)}$ be the weight trained by the Sequential Max-Margin algorithm. The update rule is

$$w_{\mathrm{SMM}}^{(t+1)} = P^{(t)}(w_{\mathrm{SMM}}^{(t)}) := \arg \min_{w \in \mathbb{R}^d} \left\| w - w_{\mathrm{SMM}}^{(t)} \right\|^2 \quad \text{subject to } y_i x_i^\top w \geq 1, \forall i \in I^{(t)}, \tag{10}$$

where the operator $P^{(t)}$ is the orthogonal projection onto a convex polyhedral set

$$\left\{ w \in \mathbb{R}^d : y_i x_i^\top w \geq 1, \forall i \in I^{(t)} \right\} \tag{11}$$

defined by the margin conditions on the data points in $I^{(t)}$. That is, $w_{\mathrm{SMM}}^{(t)}$ is the same as the sequential projection onto such convex sets. Evron et al. (2023) showed the relation of $w_\lambda^{(t)}$ and $w_{\mathrm{SMM}}^{(t)}$, when the regularization coefficient $\lambda \to 0$:

**Theorem B.1** (Theorem 3.1 of Evron et al. (2023)). *For almost all jointly separable dataset, in the limit of $\lambda \to 0$, it holds that $w_\lambda^{(t)} \to w_{\mathrm{SMM}}^{(t)}$ with a residual of $O(t \log \log \left( \frac{1}{\lambda} \right))$. Hence, at any $t = o\left( \frac{\log(1/\lambda)}{\log \log(1/\lambda)} \right)$, we get*

$$\lim_{\lambda \to 0} \frac{w_\lambda^{(t)}}{\|w_\lambda^{(t)}\|} = \frac{w_{\mathrm{SMM}}^{(t)}}{\|w_{\mathrm{SMM}}^{(t)}\|}.$$

Based on this equivalence in terms of parameter *direction*, Evron et al. (2023) expect that the behavior of $w_\lambda^{(t)}$ can be analyzed through the lens of $w_{\mathrm{SMM}}^{(t)}$ as long as $\lambda$ is close to 0, since Theorem B.1 holds for all $t = o\left( \frac{\log(1/\lambda)}{\log \log(1/\lambda)} \right)$.

Given this background, we now highlight some differences between Evron et al. (2023) and our analysis. First of all, as seen in (9), Evron et al. (2023) study regularized exponential loss trained until convergence, whereas we study unregularized logistic loss trained for a fixed number of iterations. Training the weakly regularized loss until convergence, in conjunction with the limit $\lambda \to 0$, sends each $w_\lambda^{(t)}$ to infinity. Hence, each stage requires a growing number of iterations, and the grounds for the equivalence between (9) and (10) become weaker, since the solutions become vastly different in terms of magnitude.

Second, thanks to the connection between weakly-regularized continual learning and SMM, Evron et al. (2023) could obtain the exact trajectory of every stage via the projection method. On the other hand, in our sequential GD setting, it is very difficult to keep track of the exact location of the iterate after a task is trained, since the iterates are updated multiple times, but training stops before convergence. This makes it challenging to analyze implicit bias and forgetting by tracking the exact trajectory stage by stage. To overcome this challenge, we use different proof techniques from Evron et al. (2023). Rather than pinpointing the exact position of the iterate after each stage, we focus on the direction in which the sequential GD eventually converges.

On top of that, importantly, our analysis of sequential GD reveals that training on unregularized loss using a fixed number of GD iterations results in the joint/offline max-margin solution. In contrast, although the convergence to *some* offline solutions is already shown for SMM (Evron et al., 2023), the converged offline solution can be different from the offline *max-margin* solution. In fact, in the next section (Appendix C.1), we demonstrate by a toy example that SMM can indeed converge to a point other than the joint max-margin solution.

## C EXPERIMENT DETAILS & OMITTED EXPERIMENTAL RESULTS

### C.1 EXPERIMENT DETAILS OF FIGURE 1

In this section, we present a simple toy example that demonstrates interesting facts about max-margin solutions in continual linear classification:

- The joint max-margin direction of the joint dataset can be quite different from the max-margin solutions of individual tasks. Specifically, the joint solution may *not* be on the subspace spanned by the individual solutions.
- The limit of Sequential Max-Margin (SMM) iterations can be different from the joint max-margin solution, whereas the limit direction of sequential GD does align with it.

We consider the case of $M = 2$ tasks, where the input points come from $\mathbb{R}^3$. Without loss of generality, we assume that all the labels are $+1$, and hence omit them. We let $\{(1, 1, 0), (1, -2, 1)\}$ be the dataset of task 1, and $\{(1, 0, 1), (1, 1, -2)\}$ be the data of task 2. One can verify that:

- Their joint max-margin direction is $(1, 0, 0)$.
- The max-margin direction for tasks 1 and 2 are $\left(\frac{10}{11}, \frac{1}{11}, \frac{3}{11}\right)$ and $\left(\frac{10}{11}, \frac{3}{11}, \frac{1}{11}\right)$, respectively.

Therefore, the joint max-margin solution does not belong to the span of individual max-margin solutions.

We ran numerical experiments running the SMM iterations, which are done by solving the constrained minimization problems using `fmincon` in MATLAB Optimization Toolbox. The code is provided in our supplementary material. We find that SMM converges to $\left(\frac{12}{11}, \frac{1}{11}, \frac{1}{11}\right)$; the trajectory for 10 cycles can be seen in Figure 4.

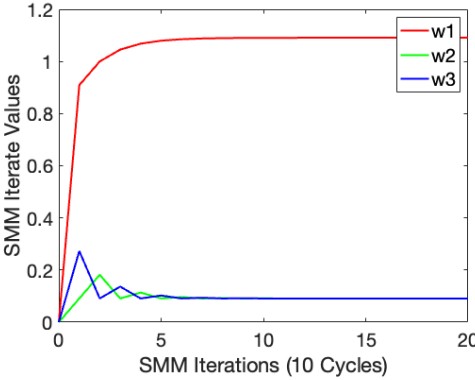

Figure 4: We run SMM iterations on the toy example by solving the projection problems using an optimization solver.

### C.2 EXPERIMENT DETAILS OF FIGURE 2 & MORE RESULTS

Here we present the experimental details of Figure 2. We also provide omitted results related to it. Then, more importantly, we extend our experimental setups beyond the cyclic task ordering and the fixed total offline dataset.

#### C.2.1 EXPERIMENTAL DETAIL

**Data Generation.** We carefully design three 2D synthetic datasets. Each dataset (of size 100) is randomly sampled from a bounded support. Below, we describe the data distribution from which we draw samples. Note that the label $y \in \{\pm 1\}$ is uniformly randomly sampled before sampling the 2D input points.

- Task 0, $\boldsymbol{x}|y = +1$: Uniform distribution on a round disk (i.e., inside of a circle) with radius $0.9$ and centered at $(0.6, 4.5)$.
- Task 0, $\boldsymbol{x}|y = -1$: Uniform distribution on a rectangle $[0, 1.5] \times [-3.9, -2.7]$.
- Task 1, $\boldsymbol{x}|y = +1$: Uniform distribution on a round disk with radius $0.75$ and centered at $(5.1, 0)$.
- Task 1, $\boldsymbol{x}|y = -1$: Uniform distribution on a rectangle $[-4.2, -2.1] \times [-0.9, 0.9]$.
- Task 2, $\boldsymbol{x}|y = +1$: Uniform distribution on a rectangle $[0.6, 3] \times [0.6, 2.7]$.
- Task 2, $\boldsymbol{x}|y = -1$: Uniform distribution on a disk with radius $1.2$ and centered at $(-3, -2.4)$.

Among all 300 data points, we randomly choose 3 points (one for each task) and replace them by $(\boldsymbol{x} = (1.5, -2.7), y = -1)$ (for task 0), $(\boldsymbol{x} = (-2.1, 0.9), y = -1)$ (for task 1), and $(\boldsymbol{x} = (0.6, 0.6), y = +1)$ (for task 2), which are the points included in the support of the data distribution(s). These three points play the role of supporting vectors so that the joint max-margin direction becomes $\frac{\hat{\boldsymbol{w}}}{\|\hat{\boldsymbol{w}}\|} = (\frac{1}{\sqrt{2}}, \frac{1}{\sqrt{2}})$, where the size of maximum margin (Equation (4)) is $\phi = 0.6\sqrt{2} > 0$ (thus, jointly separable).

**Optimization.** We run sequential GD for 300 stages in total. Since there are three tasks, for the cyclic ordering case, it is equivalent to $J = 100$. The step size we used is $\eta = 0.1$. Also, we allow and conduct $K = 1{,}000$ updates per stage. For the joint training case, we run full-batch GD on the union of all datasets for $MJK = 300{,}000$ steps.

### C.2.2 OMITTED LOSS CONVERGENCE RESULT IN FIGURE 2

Although we only displayed the directional convergence in the main text, we also observed the loss convergence to zero, which we proved in Theorems 3.1 and 3.3: see Figure 5. Note that we depict the loss values for a jointly trained model (with full-batch GD) every $K = 1{,}000$ gradient updates, for a fair comparison with a continually learned model (with sequential GD). It is omitted due to space limitations and being relatively more obvious than directional convergence.

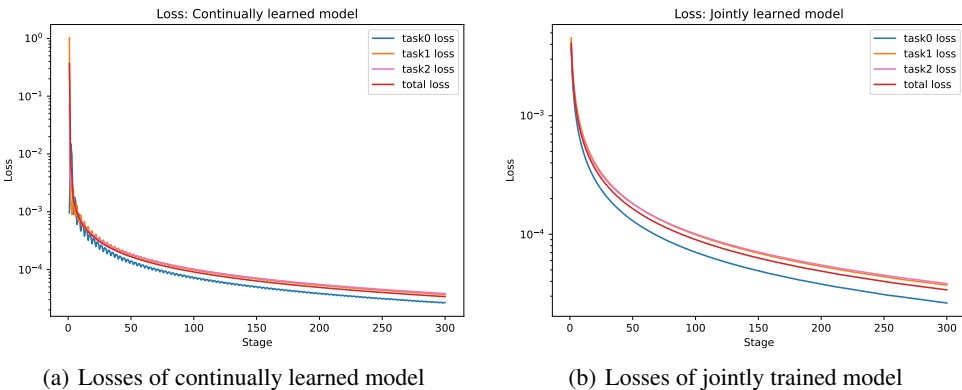

(a) Losses of continually learned model          (b) Losses of jointly trained model

Figure 5: Loss convergence results for cyclic task ordering. The joint training loss is divided by the number of tasks in order to match the scale.

### C.2.3 RANDOM TASK ORDERING

In Section 4, we theoretically showed that loss convergence, as well as the implicit bias result, holds almost surely under the random task ordering. Indeed, we observe a similar tendency of directional convergence and loss decrease even under the random task ordering. The result is shown in Figure 6.

### C.2.4 BEYOND THEORETICAL SETUP: TOWARDS CONTINUAL LEARNING ON ONLINE DATA

Most theoretical analysis in this work exploits a structural assumption on the data points: there is a pre-defined set of offline datasets, which is divided into chunks and accessible one by one at each stage. Thus, exactly the same batch of data is guaranteed to be reused (surely or with high probability). Can we go beyond this repetition and apply our theoretical intuition to more general setups?

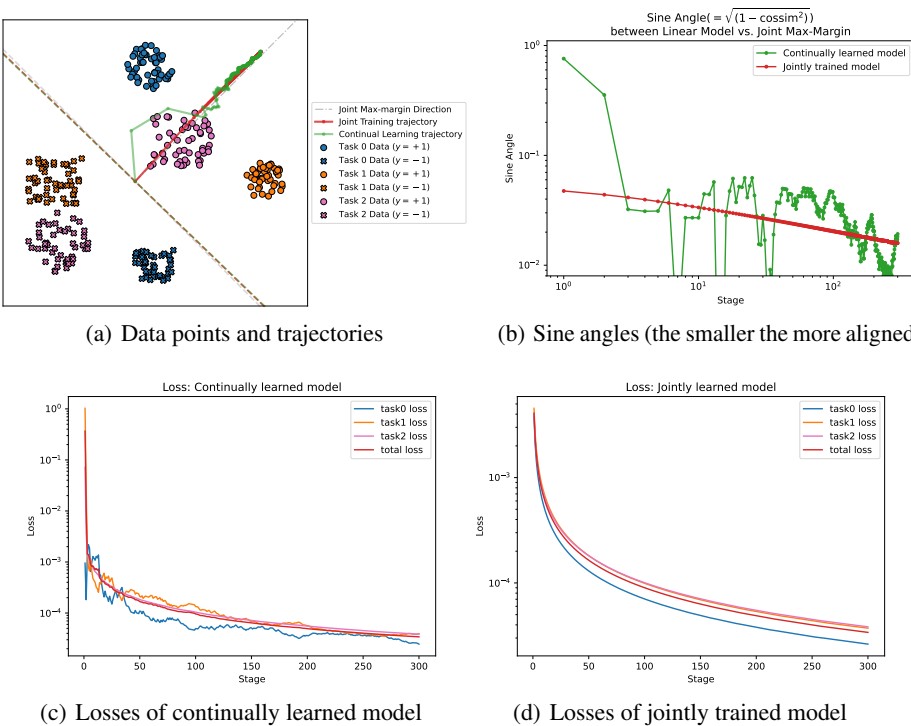

(a) Data points and trajectories

(b) Sine angles (the smaller the more aligned)

(c) Losses of continually learned model

(d) Losses of jointly trained model

Figure 6: Experiments on 2D synthetic data under random task ordering.

Here, we demonstrate that the results of our theoretical findings are not really limited to the task repetition setup. Instead, our insight about jointly separable continual linear classification applies to several general setups. In this section, we showcase an analogous behavior of sequential GD when the total dataset is no longer fixed throughout the continual learning process. We consider the setup where there are $M$ different (jointly separable) data *distributions*, rather than datasets; every time we encounter a task, we have access to a totally new sample of data points drawn from the task's distribution. For simplicity of visualization, we still stick to the bounded support cases.

An implementational difference from the previous sections is that we resample the data points from a predefined data distribution at every stage. Another minor detail is that we no longer fix the three support vectors as mentioned in Appendix C.2.1: thus, at every stage, we never reuse the same data point(s) from the previous stage, almost surely. We test whether a similar trend happens even when we add the resampling process, under the same data distribution described in Appendix C.2.1. The results are shown in Figures 7 and 8 for cyclic task ordering and random task ordering cases, respectively.

## C.3 TOY EXAMPLE FOR INCREASING LOSS IN A CYCLE

Here, we give a toy example that shows temporarily increasing joint training loss during a cycle, even with a small learning rate.

Let the datasets $\mathcal{D}_i$ ($i = 1, ..., 5$) be as follows. We choose all labels as $+1$ without loss of generality, hence we omitted them.

$$D_1 = \{(1, -2)\}, \ D_2 = \{(1, 2)\}, \ D_3 = \{(1.1, 2.1)\},$$
$$D_4 = \{(1.1, 2.2)\}, \ D_5 = \{(1.1, 2.3)\}.$$

In this case, the max-margin direction is $(1, 0)$, while most of the task has their individual max-margin direction around $(1, 2)$. We set $K = 10, \eta = 10^{-6}$ so that $\eta$ satisfies the learning rate condition.

When task 1 is being trained, joint training loss increases while it decreases when other tasks are being trained. This is because most of the tasks have their own max-margin direction around $(1, 2)$, dominating the joint training loss.

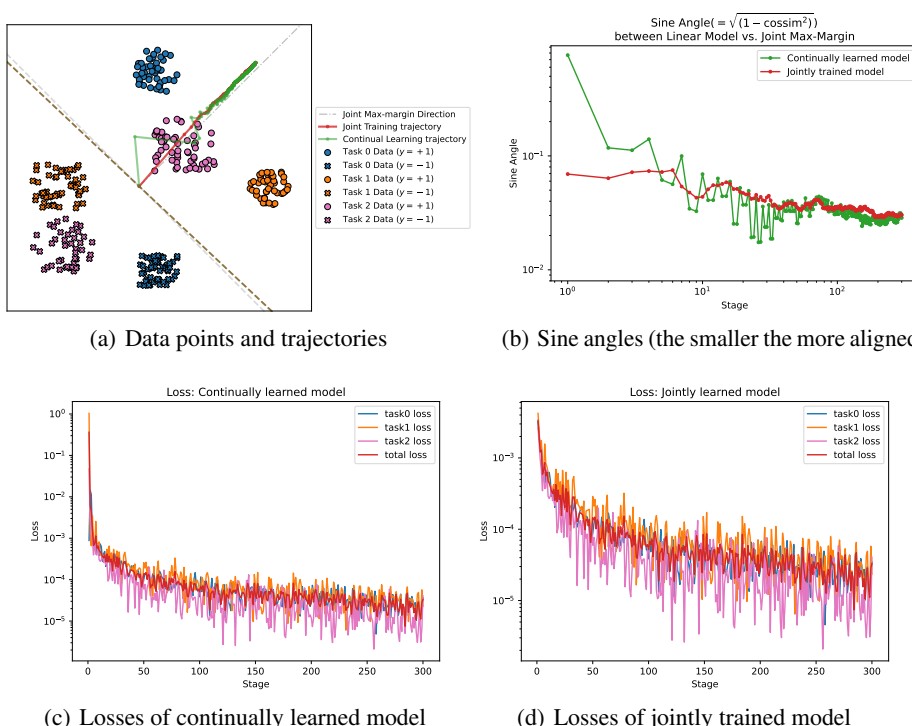

(a) Data points and trajectories

(b) Sine angles (the smaller the more aligned)

(c) Losses of continually learned model

(d) Losses of jointly trained model

Figure 7: 2D synthetic experiments: Cyclic task ordering, jointly separable online dataset (which is drawn in an online/streaming fashion from a task's predefined data distribution).

## C.4    EXPERIMENTS WITH NEURAL NETWORKS: BEYOND LINEAR MODELS

We explore the possibility of extending our theoretical insight to *nonlinear* models, in particular, wide two-layer ReLU networks.

For a *linear* classifier with a single linear layer, recall that we already verified that the sequentially trained model (in cyclic/random task ordering) directionally converges to the max-margin direction. However, it is more difficult to analyze and visualize the dynamics of the multi-layer neural net's parameter values. Moreover, it might be nonsense to discuss the relationship (e.g., alignment, directional convergence) between the max-margin direction and the parameter matrices of a neural net, because the parameter matrices themselves no longer have a semantic meaning in the data space.

Instead of inspecting the parameter values, we move our attention to the *decision boundary* of the model. Observe that the decision boundary of a linear binary classifier is a hyperplane (i.e., $d - 1$ dimensional subspace) of the data space (of $d$ dimensions), whose orthogonal complement is the span of the classifier's weight vector. Thus, the alignment between the weight vector and the max-margin direction (i.e., the implicit bias guarantee) is semantically equivalent to the alignment between the classifier's decision boundary and a hyperplane determined by the max-margin solution as a normal vector; this hyperplane can be approximated well by jointly training a single-layer linear classifier. Thus, we can still verify the similar idea of implicit bias even for a neural network by observing, not only that a continually learned model (with sequential GD, under task repetition) eventually classifies all the data points correctly, but also that the decision boundary of the continually trained model getting similar to that of a jointly trained model (both starting from an identical initialization). Although we cannot exactly characterize to which set of points a two-layer ReLU net's decision boundary should converge only with our theorems, it gives an effective and efficient way to confirm our findings beyond a simple linear model.

To intuitively visualize the decision boundaries, we again use the 2D synthetic datasets. Most of the experimental setting is the same as in Appendix C.2.1, except for the following three differences:

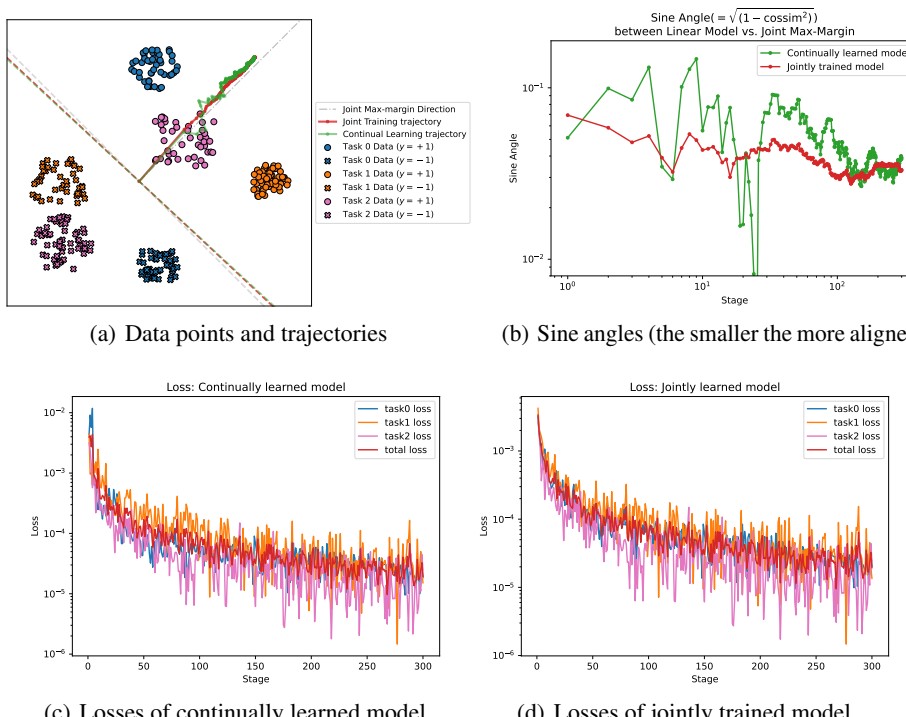

(a) Data points and trajectories

(b) Sine angles (the smaller the more aligned)

(c) Losses of continually learned model

(d) Losses of jointly trained model

Figure 8: 2D synthetic experiments: Random task ordering, jointly separable online dataset (which is drawn in an online/streaming fashion from a task's predefined data distribution).

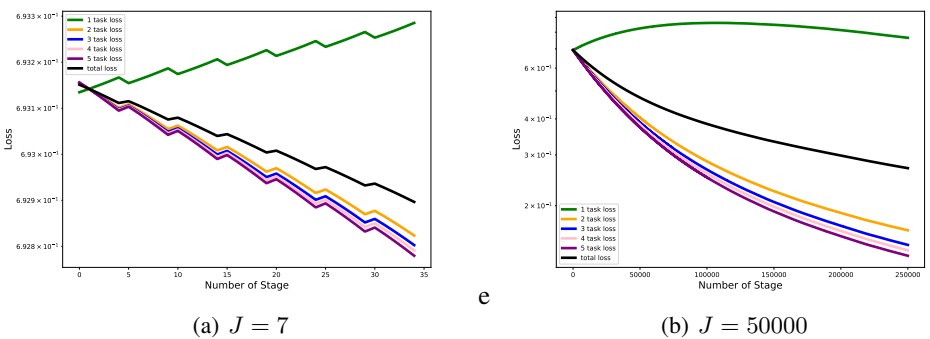

(a) $J = 7$

(b) $J = 50000$

Figure 9: We take average on total loss(black) for better visualization. The current task switches every 10 iterations. One cycle consists of 5 stages. Figure 9(a) shows the case where some task's loss increases within a cycle. However, it eventually decreases as Figure 9(b) shows.

1. The classifier's architecture is a two-layer neural network $f_{\boldsymbol{\theta}} : \mathbb{R}^2 \to \mathbb{R}$ consisting of 2-dimensional input, 500 hidden ReLU neurons, and scalar output:

$$f_{\boldsymbol{\theta}}(\boldsymbol{x}) = \boldsymbol{w}_2^\top \operatorname{ReLU}(\boldsymbol{W}_1 \boldsymbol{x} + \boldsymbol{b}_1) + b_2,$$

where $\boldsymbol{\theta} = (\boldsymbol{W}_1, \boldsymbol{b}_1, \boldsymbol{w}_2, b_2) \in \mathbb{R}^{500 \times 2} \times \mathbb{R}^{500} \times \mathbb{R}^{500} \times \mathbb{R}$ and $\operatorname{ReLU}(\boldsymbol{v})_i = \max\{\boldsymbol{v}_i, 0\}$.

2. To make the total dataset non-separable by a linear classifier with a positive margin but still classifiable by a neural net, we translate all data points with positive labels $(+1)$ by a vector $(-1.2, -1.2)$. In this case, the decision boundary should not be straight but bent in a curly L-shape to effectively distinguish two classes.

3. To prevent the sequential GD from behaving similarly to a mini-batch SGD with small-scale and lazy updates, we increase $K$ to 3,000 to guarantee that (1) the jointly trained model can

correctly classify all data points within only one stage (i.e., with initial $K$ updates), and (2) the continually learned model gets sufficiently trained on a specific task at each stage. As a result, the jointly trained model takes $MJK = 900,000$ iterations. ($M = 3$, $J = 100$)

As we did for a linear classifier, we classify the input data as $y = +1$ if the model output is positive and as $-1$ otherwise (thus, the decision boundary is a level set $\{x \in \mathbb{R}^2 : f_\theta(x) = 0\}$). We again use the usual logistic loss $\frac{1}{N} \sum_{i=1}^{N} \ell(y_i f_\theta(x_i))$.

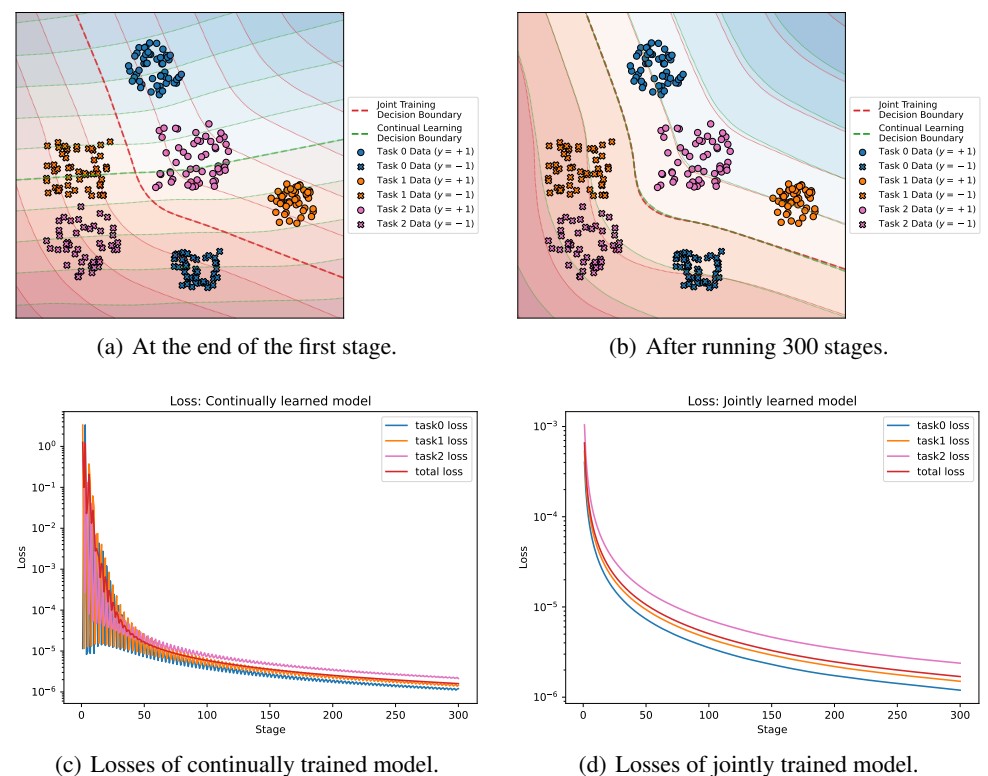

(a) At the end of the first stage.

(b) After running 300 stages.

(c) Losses of continually trained model.

(d) Losses of jointly trained model.

Figure 10: Two-layer ReLU network experiment under cyclic task ordering. **(Top.)** Each subfigure displays the decision boundaries (and other auxiliary level sets) of a jointly trained model (dashed red line) and a continually trained model (dashed green line). **(Bottom.)** Figure 10(c) demonstrates the large amounts of forgetting at the initial few cycles and convergence of loss and (cycle-averaged) forgetting to near zero. On the other hand, Figure 10(d) shows that the training loss of the jointly trained model is already small (e.g., less than $10^{-3}$) at the initial stages.

The result of the experiment for cyclic task order is visualized in Figure 10, exhibiting decision boundaries of a jointly trained model and a continually trained model (with sequential GD). As we expected, the updates are aggressive enough so that even a single stage (i.e., initial $K = 3,000$ iterations) is sufficient to perfectly classify the total dataset with the jointly trained model, and the same for the dataset of the task 0 with the continually trained model (Figure 10(a)). After some number of stages (Figure 10(b)), both models not only correctly classify every data point, but also have an almost identical decision boundary (note that the other level sets are not necessarily the same), implying that a similar phenomenon like implicit bias is happening here. We also observe almost the same tendencies under random task ordering and even for non-repeating dataset cases (Appendix C.2.4). We omit their detailed results from the paper, but one can find them in our supplementary materials.

## C.5 EXPERIMENT ON A REAL-WORLD DATASET

In this section, we present a result of the training linear model with CIFAR-10 (Krizhevsky, 2009).

We choose two classes from the CIFAR-10 dataset and design 3 tasks that have 512 data points from the two classes ('airplane' and 'automobile'). Our Theorem 5.2 on linearly non-separable data like CIFAR-10 shows that sequential GD iterates should not diverge and instead converge to the global minimum $\boldsymbol{w}^*$ under the properly chosen learning rate. To estimate the distance between sequential GD iterates and the global minimum, we first train a linear model using joint task data and obtain $\boldsymbol{w}_{\text{Joint}}$ as a proxy of $\boldsymbol{w}^*$; we do this because offline training is guaranteed to converge to the global minimum. Then, we train sequential GD and measure the distance between iterates and the jointly trained solution $\boldsymbol{w}_{\text{Joint}}$ at the end of every stage of sequential GD.

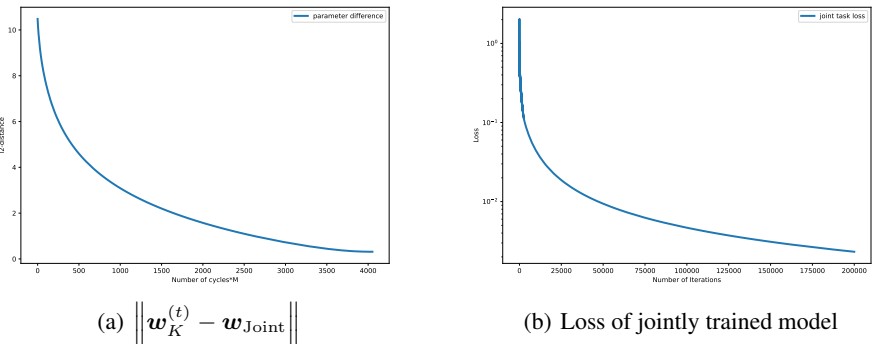

(a) $\left\| \boldsymbol{w}_K^{(t)} - \boldsymbol{w}_{\text{Joint}} \right\|$

(b) Loss of jointly trained model

Figure 11: **CIFAR-10 Experiments with a linear model.** We jointly train a model for 200000 iterations to achieve the global minimum. We then train each task with cyclic ordering. We set the number of GD for each stage as 50 ($K = 50$), and run 1350 cycles ($J = 1350$). Figure 11(a) shows that sequential GD iterate converges close to $\boldsymbol{w}_{\text{Joint}}$ as the training goes on. However, it does not fully converge to $\boldsymbol{w}_{\text{Joint}}$, as $\boldsymbol{w}_{\text{Joint}}$ is not equal to $\boldsymbol{w}^*$. Figure 11(b) reveals that the loss of the jointly trained model was decreasing after 200000 iterations.

As a result, we observe that the distance between sequential GD iterates and $\boldsymbol{w}_{\text{Joint}}$ converges close to 0, even when we adopt a learning rate $\eta = 0.01$, which is not as small as our theorem requires. Yet, we couldn't show convergence of distance to exactly 0 since the jointly trained model did not converge all the way to $\boldsymbol{w}^*$.

# D PROOF SKETCHES

In this appendix, we provide proof sketches of the main theorems of the asymptotic and non-asymptotic convergence under the cyclic task ordering. The proof sketch for the random task ordering is omitted, as the asymptotic loss convergence is straightforward, and the directional convergence can be established in much the same way as for the cyclic task ordering. In addition, we include proof sketches for Lemma E.5 whose proof is particularly intricate and challenging to follow.

## D.1 ASYMPTOTIC LOSS CONVERGENCE (PROOF SKETCH OF THEOREM 3.1)

Since $\mathcal{L}$ is a $\sigma_{\max}^2 \beta$-smooth function, we get

$$
\begin{aligned}
&\mathcal{L}(\boldsymbol{w}_0^{(Mj+M)}) - \mathcal{L}(\boldsymbol{w}_0^{(Mj)}) - \frac{\sigma_{\max}^2 \beta}{2} \left\| \boldsymbol{w}_0^{(Mj+M)} - \boldsymbol{w}_0^{(Mj)} \right\|^2 \\
&\leq \nabla \mathcal{L}(\boldsymbol{w}_0^{(Mj)})^\top (\boldsymbol{w}_0^{(Mj+M)} - \boldsymbol{w}_0^{(Mj)}) \\
&\leq -\eta K \left\| \nabla \mathcal{L}(\boldsymbol{w}_0^{(Mj)}) \right\|^2 + \left\| \nabla \mathcal{L}(\boldsymbol{w}_0^{(Mj)}) \right\| \left\| \boldsymbol{w}_0^{(Mj+M)} - \boldsymbol{w}_0^{(Mj)} + \eta K \nabla \mathcal{L}(\boldsymbol{w}_0^{(Mj)}) \right\|,
\end{aligned}
$$

for each $j$-th cycle ($j \geq 0$). Since the term $\left\| \boldsymbol{w}_0^{(Mj+M)} - \boldsymbol{w}_0^{(Mj)} + \eta K \nabla \mathcal{L}(\boldsymbol{w}_0^{(Mj)}) \right\|$ is bounded above by a factor (in $j$) of $\left\| \nabla \mathcal{L}(\boldsymbol{w}_0^{(Mj)}) \right\|$ (Lemma E.2), we have a form of descent lemma,

$$
\mathcal{L}(\boldsymbol{w}_0^{(Mj+M)}) - \mathcal{L}(\boldsymbol{w}_0^{(Mj)}) \leq -\eta K \left(1 - \eta K \beta'\right) \left\| \nabla \mathcal{L}(\boldsymbol{w}_0^{(Mj)}) \right\|^2, \tag{12}
$$

for some $\beta' \in \left(0, \frac{1}{\eta K}\right)$. Therefore,

$$
\sum_{j=0}^{\infty} \left\| \nabla \mathcal{L}(\boldsymbol{w}_0^{(Mj)}) \right\|^2 \leq \frac{\mathcal{L}(\boldsymbol{w}_0^{(0)})}{\eta K (1 - \eta K \beta')} < \infty.
$$

Again by Lemma E.2, the term $\left\| \nabla \mathcal{L}(\boldsymbol{w}_k^{(Mj+m)}) \right\|^2$ is bounded above by a constant factor (in $j$) of $\left\| \nabla \mathcal{L}(\boldsymbol{w}_0^{(Mj)}) \right\|$, which leads to the following convergence of an infinite sum of non-negative terms:

$$
\sum_{j=0}^{\infty} \sum_{m=0}^{M-1} \sum_{k=0}^{K-1} \left\| \nabla \mathcal{L}(\boldsymbol{w}_k^{(Mj+m)}) \right\|^2 < \infty.
$$

This implies $\forall k \in [0 : K-1] : \lim_{t \to \infty} \left\| \nabla \mathcal{L}(\boldsymbol{w}_k^{(t)}) \right\|^2 = 0$. Finally, Lemma E.1 leads to $\lim_{t \to \infty} \ell'(\boldsymbol{x}_i^\top \boldsymbol{w}_k^{(t)}) = 0$ for all $k \in [0 : K-1]$. Equivalently, $\boldsymbol{x}_i^\top \boldsymbol{w}_k^{(t)} \to \infty$.

## D.2 DIRECTIONAL CONVERGENCE (PROOF SKETCH OF THEOREM 4.2)

First, for each $k \in [0 : K-1]$, define $\boldsymbol{\rho}_k^{(t)}$ and $\boldsymbol{r}_k^{(t)}$ as

$$
\begin{aligned}
\boldsymbol{w}_k^{(t)} &= \ln(t) \, \hat{\boldsymbol{w}} + \boldsymbol{\rho}_k^{(t)} \\
&= \ln(t) \, \hat{\boldsymbol{w}} + \ln\left(\frac{K}{M}\right) \hat{\boldsymbol{w}} + \tilde{\boldsymbol{w}} + \frac{M}{K} \cdot \boldsymbol{m}_{t,k} + \boldsymbol{r}_k^{(t)},
\end{aligned}
$$

where $\hat{\boldsymbol{w}}$ is the joint $\ell_2$ max-margin solution (defined in Equation (3)), $\tilde{\boldsymbol{w}}$ is the solution of

$$
\forall i \in S : \eta \exp\left(-\boldsymbol{x}_i^\top \tilde{\boldsymbol{w}}\right) = \alpha_i, \quad \bar{P}(\tilde{\boldsymbol{w}} - \boldsymbol{w}_0^{(0)}) = 0,
$$

and $\boldsymbol{m}_{t,k}$ is a vector defined in Lemma E.4 in detail. Then by definition, the following holds.

$$
\boldsymbol{r}_k^{(t)} = \boldsymbol{w}_k^{(t)} - \frac{M}{K} \left( \frac{K}{M} \ln\left(\frac{K}{M} t\right) \hat{\boldsymbol{w}} + \boldsymbol{m}_{t,k} \right) - \tilde{\boldsymbol{w}} \tag{13}
$$

$$= \boldsymbol{w}_k^{(t)} - \frac{M}{K} \left( K \sum_{u=1}^{t-1} \frac{1}{u} \sum_{s \in S^{(u)}} \alpha_s \boldsymbol{x}_s + \frac{k}{t} \sum_{s \in S^{(t)}} \alpha_s \boldsymbol{x}_s \right) - \ln K \hat{\boldsymbol{w}} - \tilde{\boldsymbol{w}} + \check{\boldsymbol{w}}. \tag{14}$$

To show the boundedness of $\boldsymbol{\rho}_k^{(t)}$, it is enough to prove the boundedness of $\boldsymbol{r}_k^{(t)}$ as $\tilde{\boldsymbol{w}}$ is a constant vector and $\boldsymbol{m}_{t,k}$ converges to zero. From the fact

$$\left\| \boldsymbol{r}_{k+1}^{(t)} \right\|^2 - \left\| \boldsymbol{r}_k^{(t)} \right\|^2 = 2(\boldsymbol{r}_{k+1}^{(t)} - \boldsymbol{r}_k^{(t)})^\top \boldsymbol{r}_k^{(t)} + \left\| \boldsymbol{r}_{k+1}^{(t)} - \boldsymbol{r}_k^{(t)} \right\|^2,$$

we get

$$\left\| \boldsymbol{r}_0^{(t)} \right\|^2 - \left\| \boldsymbol{r}_0^{(t_1)} \right\|^2 = \sum_{u=t_1}^{t-1} \sum_{k=0}^{K-1} 2(\boldsymbol{r}_{k+1}^{(u)} - \boldsymbol{r}_k^{(u)})^\top \boldsymbol{r}_k^{(u)} + \sum_{u=t_1}^{t-1} \sum_{k=0}^{K-1} \left\| \boldsymbol{r}_{k+1}^{(u)} - \boldsymbol{r}_k^{(u)} \right\|^2. \tag{15}$$

According to Lemma E.5, the first term of the right-hand side of Equation (15) is bounded. By Equation (13),

$$\sum_{t=t_1}^{T} \sum_{k=0}^{K-1} \left\| \boldsymbol{r}_{k+1}^{(t)} - \boldsymbol{r}_k^{(t)} \right\|^2 = \sum_{t=t_1}^{T} \sum_{k=0}^{K-1} \left\| \boldsymbol{w}_{k+1}^{(t)} - \boldsymbol{w}_k^{(t)} - \frac{M}{K} (\boldsymbol{m}_{t,k+1} - \boldsymbol{m}_{t,k}) \right\|^2$$

$$\leq \sum_{t=t_1}^{T} \sum_{k=0}^{K-1} \left\| \boldsymbol{w}_{k+1}^{(t)} - \boldsymbol{w}_k^{(t)} \right\|^2 + \sum_{t=t_1}^{T} \sum_{k=0}^{K-1} \left\| \frac{M}{K} (\boldsymbol{m}_{t,k+1} - \boldsymbol{m}_{t,k}) \right\|^2$$

$$+ 2 \sqrt{ \sum_{t=t_1}^{T} \sum_{k=0}^{K-1} \left\| \boldsymbol{w}_k^{(t)} - \boldsymbol{w}_{k+1}^{(t)} \right\|^2 \sum_{t=t_1}^{T} \sum_{k=0}^{K-1} \left\| \frac{M}{K} (\boldsymbol{m}_{t,k+1} - \boldsymbol{m}_{t,k}) \right\|^2 }.$$

We use the Cauchy-Schwarz inequality for the inequality. $\sum_{t=t_1}^{T} \sum_{k=0}^{K-1} \left\| \boldsymbol{w}_k^{(t)} - \boldsymbol{w}_{k+1}^{(t)} \right\|^2$ and $\sum_{t=t_1}^{T} \sum_{k=0}^{K-1} \left\| \frac{M}{K} (\boldsymbol{m}_{t,k+1} - \boldsymbol{m}_{t,k}) \right\|^2$ are bounded as a consequence of Theorem 3.1 and Lemma E.4, respectively. Therefore, the second term of the right-hand side of Equation (15) is bounded. It concludes $\left\| \boldsymbol{r}_k^{(t)} \right\|$ is bounded.

### D.2.1 Proof Sketch of Lemma E.5

From Theorem 3.1, $\forall i \in I : \boldsymbol{x}_i^\top \boldsymbol{w}_k^{(t)} \to \infty$. Thus for a sufficiently large $t$, Assumption 3.4 gives

$$-\ell'(\boldsymbol{x}_i^\top \boldsymbol{w}_k^{(t)}) \approx e^{-\boldsymbol{x}_i^\top \boldsymbol{w}_k^{(t)}}.$$

Also, since $\boldsymbol{m}_{t,k}$ converges to zero, for a sufficiently large $t$,

$$\boldsymbol{w}_k^{(t)} = \ln(\frac{K}{M} t) \hat{\boldsymbol{w}} + \tilde{\boldsymbol{w}} + \frac{M}{K} \boldsymbol{m}_{t,k} + \boldsymbol{r}_k^{(t)} \approx \ln(\frac{K}{M} t) \hat{\boldsymbol{w}} + \tilde{\boldsymbol{w}} + \boldsymbol{r}_k^{(t)}.$$

By Equation (14), for all $k \in [0 : K-1]$, we get

$$\boldsymbol{r}_{k+1}^{(t)} - \boldsymbol{r}_k^{(t)} = \boldsymbol{w}_{k+1}^{(t)} - \boldsymbol{w}_k^{(t)} - \frac{M}{Kt} \sum_{s \in S^{(t)}} \alpha_s \boldsymbol{x}_s$$

$$= -\eta \sum_{s \in I^{(t)}} \ell'(\boldsymbol{x}_s^\top \boldsymbol{w}_k^{(t)}) \boldsymbol{x}_s - \frac{M}{Kt} \sum_{s \in S^{(t)}} \alpha_s \boldsymbol{x}_s$$

$$= -\eta \sum_{s \in I^{(t)} \setminus S^{(t)}} \ell'(\boldsymbol{x}_s^\top \boldsymbol{w}_k^{(t)}) \boldsymbol{x}_s - \sum_{s \in S^{(t)}} \left[ \eta \ell'(\boldsymbol{x}_s^\top \boldsymbol{w}_k^{(t)}) + \frac{M}{Kt} \alpha_s \right] \boldsymbol{x}_s.$$

Hence,

$$\left( \boldsymbol{r}_{k+1}^{(t)} - \boldsymbol{r}_k^{(t)} \right)^\top \boldsymbol{r}_k^{(t)} = -\eta \sum_{s \in I^{(t)} \setminus S^{(t)}} \ell'(\boldsymbol{x}_s^\top \boldsymbol{w}_k^{(t)}) \boldsymbol{x}_s^\top \boldsymbol{r}_k^{(t)} - \sum_{s \in S^{(t)}} \left[ \eta \ell'(\boldsymbol{x}_s^\top \boldsymbol{w}_k^{(t)}) + \frac{M}{Kt} \alpha_s \right] \boldsymbol{x}_s^\top \boldsymbol{r}_k^{(t)}. \tag{16}$$

The first term of the right-hand side of Equation (16) can be upper bounded as below:

$$- \eta \sum_{s \in I^{(t)} \setminus S^{(t)}} \ell'(\boldsymbol{x}_s^\top \boldsymbol{w}_k^{(t)}) \boldsymbol{x}_s^\top \boldsymbol{r}_k^{(t)} \approx \eta \sum_{\substack{s \in I^{(t)} \setminus S^{(t)} \\ \boldsymbol{x}_s^\top \boldsymbol{r}_k^{(t)} > 0}} \exp\left(-\ln(\frac{K}{M}t)\boldsymbol{x}_s^\top \hat{\boldsymbol{w}} - \boldsymbol{x}_s^\top \tilde{\boldsymbol{w}} - \boldsymbol{x}_s^\top \boldsymbol{r}_k^{(t)}\right) \boldsymbol{x}_s^\top \boldsymbol{r}_k^{(t)}$$

$$\leq \eta \sum_{\substack{s \in I^{(t)} \setminus S^{(t)} \\ \boldsymbol{x}_s^\top \boldsymbol{r}_k^{(t)} > 0}} \exp\left(-\ln(\frac{K}{M}t)\boldsymbol{x}_s^\top \hat{\boldsymbol{w}} - \boldsymbol{x}_s^\top \tilde{\boldsymbol{w}}\right) = \mathcal{O}(t^{-\theta}), \tag{17}$$

where in the last line we use the fact $\forall x \geq 0 : x\exp(-x) \leq 1$, and $\forall s \in I^{(t)} \setminus S^{(t)} : x_s^\top \hat{\boldsymbol{w}} \geq \theta$.

Now we examine the second term of the right-hand side of Equation (16),

$$- \sum_{s \in S^{(t)}} \left[\eta \ell'(\boldsymbol{x}_s^\top \boldsymbol{w}_k^{(t)}) + \frac{M}{Kt}\alpha_s\right] \boldsymbol{x}_s^\top \boldsymbol{r}_k^{(t)} \approx \sum_{s \in S^{(t)}} \frac{M}{Kt}\alpha_s \left[\exp\left(-\boldsymbol{x}_s^\top \boldsymbol{r}_k^{(t)}\right) - 1\right] \boldsymbol{x}_s^\top \boldsymbol{r}_k^{(t)}.$$

Set $\tilde{\mu} = \min\{\mu_+, \mu_-, 0.25\}$, where $\mu_+, \mu_-$ is defined from Assumption 3.4. We analyze each $s \in S^{(t)}$ by dividing into cases.

1. if $0 \leq \boldsymbol{x}_s^\top \boldsymbol{r}_k^{(t)} \leq t^{-0.5\tilde{\mu}}$:

$$\frac{M}{Kt}\alpha_s \left[\exp\left(-\boldsymbol{x}_s^\top \boldsymbol{r}_k^{(t)}\right) - 1\right] \boldsymbol{x}_s^\top \boldsymbol{r}_k^{(t)} = \mathcal{O}(t^{-1-0.5\tilde{\mu}}).$$

2. if $-t^{-0.5\tilde{\mu}} \leq \boldsymbol{x}_s^\top \boldsymbol{r}_k^{(t)} \leq 0$:

$$\frac{M}{Kt}\alpha_s \left[\exp\left(-\boldsymbol{x}_s^\top \boldsymbol{r}_k^{(t)}\right) - 1\right] \boldsymbol{x}_s^\top \boldsymbol{r}_k^{(t)} = \frac{M}{Kt}\alpha_s \left[1 - \exp\left(-\boldsymbol{x}_s^\top \boldsymbol{r}_k^{(t)}\right)\right] \cdot \left|\boldsymbol{x}_s^\top \boldsymbol{r}_k^{(t)}\right|$$

$$\leq \frac{M}{Kt}\alpha_s \left|\boldsymbol{x}_s^\top \boldsymbol{r}_k^{(t)}\right| = \mathcal{O}(t^{-1-0.5\tilde{\mu}}).$$

3. if $t^{-0.5\tilde{\mu}} < \boldsymbol{x}_s^\top \boldsymbol{r}_k^{(t)}$:

$$\frac{M}{Kt}\alpha_s \left[\exp\left(-\boldsymbol{x}_s^\top \boldsymbol{r}_k^{(t)}\right) - 1\right] \boldsymbol{x}_s^\top \boldsymbol{r}_k^{(t)} \leq \frac{M}{Kt}\alpha_s \left[\exp\left(-t^{-0.5\tilde{\mu}}\right) - 1\right] \boldsymbol{x}_s^\top \boldsymbol{r}_k^{(t)}$$

$$\leq \frac{M}{Kt}\alpha_s \left[-t^{-0.5\tilde{\mu}} + t^{-\tilde{\mu}}\right] \boldsymbol{x}_s^\top \boldsymbol{r}_k^{(t)},$$

where in the last line we use the fact $\forall x \leq 1 : \exp x \leq 1 + x + x^2$. Since $-t^{-0.5\tilde{\mu}}$ decreases to zero more slowly than the other term, the last term is negative for a sufficiently large $t$.

4. if $\boldsymbol{x}_s^\top \boldsymbol{r}_k^{(t)} < -t^{-0.5\tilde{\mu}}$:

Since $\boldsymbol{x}_s^\top \boldsymbol{r}_k^{(t)} < 0$, $\exp\left(-\boldsymbol{x}_s^\top \boldsymbol{r}_k^{(t)}\right) > 1$ holds. Therefore

$$\frac{M}{Kt}\alpha_s \left[\exp\left(-\boldsymbol{x}_s^\top \boldsymbol{r}_k^{(t)}\right) - 1\right] \boldsymbol{x}_s^\top \boldsymbol{r}_k^{(t)} \leq 0.$$

To sum up, there exist $C_1, C_2 > 0, \tilde{t} \geq \max\{t_+, t_-\}$ such that for all $t \geq \tilde{t}$,

$$(\boldsymbol{r}_{k+1}^{(t)} - \boldsymbol{r}_k^{(t)})^\top \boldsymbol{r}_k^{(t)} \leq C_1 t^{-\theta} + C_2 t^{-1-0.5\tilde{\mu}}, \forall k \in [0 : K-1].$$

### D.3 NON-ASYMPTOTIC LOSS CONVERGENCE ANALYSIS (PROOF SKETCH OF THEOREM 3.3)

The analysis in Appendix D.1 can be extended similarly to arbitrary cycle number. That is,

$$\forall j < J : \mathcal{L}(\boldsymbol{w}_0^{(Mj+M)}) \leq \mathcal{L}(\boldsymbol{w}_0^{(Mj)}) - \eta K (1 - \eta K \beta') \left\|\nabla \mathcal{L}(\boldsymbol{w}_0^{(Mj)})\right\|^2.$$

Thus by applying Lemma E.9, we obtain

$$2\sum_{j=0}^{J-1} \eta K \left(\mathcal{L}(\boldsymbol{w}_0^{(Mj+M)}) - \mathcal{L}(\boldsymbol{z})\right) - \frac{2\eta MK\sigma_{\max}^4 \beta}{\phi^2(1 - \eta MK\sigma_{\max}^2 \beta)^2} \sum_{j=0}^{J-1} \frac{\eta K}{1 - \eta K\beta'} \left(\mathcal{L}(\boldsymbol{w}_0^{(Mj)}) - \mathcal{L}(\boldsymbol{w}_0^{(Mj+M)})\right)$$

$$\leq \left\| \boldsymbol{w}_0^{(0)} - \boldsymbol{z} \right\|^2 - \left\| \boldsymbol{w}_0^{(MJ)} - \boldsymbol{z} \right\|^2, \tag{18}$$

for any $\boldsymbol{z} \in \mathbb{R}^d$. Thus

$$2\eta KJ \left( \mathcal{L}(\boldsymbol{w}_0^{(MJ)}) - \mathcal{L}(\boldsymbol{z}) \right) + \frac{8\sigma_{\max}^2}{\phi^2} \eta K \left( \mathcal{L}(\boldsymbol{w}_0^{(MJ)}) - \mathcal{L}(\boldsymbol{w}_0^{(0)}) \right)$$

$$\leq 2 \sum_{j=0}^{J-1} \eta K \left( \mathcal{L}(\boldsymbol{w}_0^{(Mj+M)}) - \mathcal{L}(\boldsymbol{z}) \right) + \frac{8\sigma_{\max}^2}{\phi^2} \eta K \left( \mathcal{L}(\boldsymbol{w}_0^{(MJ)}) - \mathcal{L}(\boldsymbol{w}_0^{(0)}) \right)$$

$$= 2 \sum_{j=0}^{J-1} \eta K \left( \mathcal{L}(\boldsymbol{w}_0^{(Mj+M)}) - \mathcal{L}(\boldsymbol{z}) \right) - \frac{8\sigma_{\max}^2}{\phi^2} \eta K \sum_{j=0}^{J-1} \left( \mathcal{L}(\boldsymbol{w}_0^{(Mj)}) - \mathcal{L}(\boldsymbol{w}_0^{(Mj+M)}) \right)$$

$$2 \sum_{j=0}^{J-1} \eta K \left( \mathcal{L}(\boldsymbol{w}_0^{(Mj+M)}) - \mathcal{L}(\boldsymbol{z}) \right) - \frac{2\eta MK\sigma_{\max}^4 \beta}{\phi^2 (1 - \eta MK\sigma_{\max}^2 \beta)^2} \sum_{j=0}^{J-1} \frac{\eta K}{1 - \eta K\beta'} \left( \mathcal{L}(\boldsymbol{w}_0^{(Mj)}) - \mathcal{L}(\boldsymbol{w}_0^{(Mj+M)}) \right)$$

$$\leq \left\| \boldsymbol{w}_0^{(0)} - \boldsymbol{z} \right\|^2 - \left\| \boldsymbol{w}_0^{(MJ)} - \boldsymbol{z} \right\|^2, \tag{19}$$

where in the first and second inequalities we use the fact that $\mathcal{L}(\boldsymbol{w}_0^{(Mj)})$ is decreasing and $\eta < \min\{\frac{1}{2MK\beta\sigma_{\max}^2}, \frac{1}{2K\beta'}\}$. Re-arranging gives

$$\mathcal{L}(\boldsymbol{w}_0^{(MJ)}) \leq \mathcal{L}(\boldsymbol{z}) + \frac{4\sigma_{\max}^2}{\phi^2} \frac{\mathcal{L}(\boldsymbol{w}_0^{(0)})}{J} + \frac{\left\| \boldsymbol{w}_0^{(0)} - \boldsymbol{z} \right\|^2}{2\eta KJ}.$$

Let $\boldsymbol{z} := \hat{\boldsymbol{w}} \ln(MJ)$. Using $\mathcal{L}(\hat{\boldsymbol{w}} \ln(MJ)) \leq |I| \ell(\ln(MJ))$, we can obtain $\mathcal{O}(\frac{\ln^2 J}{J})$ upper bound for the training loss.

# E PROOFS FOR SECTION 3: CYCLIC TASK ORDERING, JOINTLY SEPARABLE

For simplicity, we set $y_i = 1$ for all $i \in I = [1 : N]$ in the proofs, without loss of generality. If not, we can convert every data into $(\widetilde{x}_i, \widetilde{y}_i) = (y_i x_i, +1)$ by "absorbing" the original label into the input.

## E.1 ASYMPTOTIC LOSS CONVERGENCE ANALYSIS (PROOF OF THEOREM 3.1)

Let us restate the theorem here for the sake of readability, whose proof sketch is provided in Appendix D.1.

**Theorem 3.1.** *Let $\{w_k^{(t)}\}_{k \in [0:K-1], t \geq 0}$ be the sequence of GD iterates (2) from any starting point $w_0^{(0)}$, where tasks are given cyclically. Under Assumptions 3.1 and 3.3, if the learning rate satisfies $\eta < \frac{\phi^2}{2K\beta\sigma_{\max}^3(M\phi + \sigma_{\max})}$, then*

1. *Loss converges to zero:* $\lim_{t \to \infty} \mathcal{L}(w_k^{(t)}) = 0, \forall k \in [0 : K - 1]$.

2. *All data points are eventually classified correctly:* $\lim_{t \to \infty} y_i x_i^\top w_k^{(t)} = \infty, \forall k \in [0 : K - 1], i \in I$.

3. *Square sum of the change of weight is finite:* $\sum_{t=0}^{\infty} \sum_{k=0}^{K-1} \|w_{k+1}^{(t)} - w_k^{(t)}\|^2 < \infty$.

Let us recap some notation regarding the dataset and loss function. The training loss of task $m$, $\mathcal{L}_m(w) = \sum_{i \in I_m} \ell(y_i x_i^\top w)$, is $\beta_m$-smooth for $\beta_m := \beta\lambda_{\max}(X_m X_m^\top)$ where $X_m \in \mathbb{R}^{d \times |I_m|}$ is a data matrix of task $m$ consisting of columns $\{x_i : i \in I_m\}$. Let $\beta_{\max} = \max_{m \in [0:M-1]} \beta_m$; by definition, $\beta_{\max} \leq \beta\sigma_{\max}^2 \leq \sum_{m \in [0:M-1]} \beta_m$ holds.

In this proof, we rely on the key property of linearly separable data.

**Lemma E.1.** *Suppose that Assumptions 3.1 and 3.3 holds. For any $w \in \mathbb{R}^d$,*

$$\|\nabla\mathcal{L}(w)\| \geq \phi\sqrt{\sum_{i \in I} \left[\ell'(x_i^\top w)\right]^2}.$$

*Proof.* See Appendix E.1.1. □

Also, we use the following lemma, which holds in cyclic continual learning with $M$ tasks.

**Lemma E.2.** *Suppose that Assumptions 3.1 and 3.3 holds. Let any $t \in \mathbb{N}, m \in [0 : M - 1], k \in [0 : K - 1]$. Then, if we choose the step size as $\eta < \frac{1}{(mK+k)\sigma_{\max}^2\beta}$, we have*

$$\left\|w_k^{(t+m)} - w_0^{(t)} + \eta\left(K\sum_{i=0}^{m-1} \nabla\mathcal{L}^{(t+i)}(w_0^{(t)}) + k\nabla\mathcal{L}^{(t+m)}(w_0^{(t)})\right)\right\|$$
$$\leq \frac{\eta^2(mK+k)K\sigma_{\max}^3\beta}{\phi\{1 - \eta(mK+k)\sigma_{\max}^2\beta\}}\left\|\nabla\mathcal{L}(w_0^{(t)})\right\|,$$
$$\left\|w_k^{(t+m)} - w_0^{(t)}\right\| \leq \frac{\eta K\sigma_{\max}}{\phi\{1 - \eta(mK+k)\sigma_{\max}^2\beta\}}\left\|\nabla\mathcal{L}(w_0^{(t)})\right\|,$$
$$\left\|\nabla\mathcal{L}(w_k^{(t+m)}) - \nabla\mathcal{L}(w_0^{(t)})\right\| \leq \frac{\eta K\sigma_{\max}^3\beta}{\phi\{1 - \eta(mK+k)\sigma_{\max}^2\beta\}}\left\|\nabla\mathcal{L}(w_0^{(t)})\right\|.$$

*Proof.* See Appendix E.1.2. □

Since $\mathcal{L}$ is a $\sigma_{\max}^2\beta$-smooth function, we get

$$\mathcal{L}\left(w_0^{(Mj+M)}\right) - \mathcal{L}\left(w_0^{(Mj)}\right) - \frac{\sigma_{\max}^2\beta}{2}\left\|w_0^{(Mj+M)} - w_0^{(Mj)}\right\|^2$$
$$\leq \nabla\mathcal{L}(w_0^{(Mj)})^\top(w_0^{(Mj+M)} - w_0^{(Mj)})$$
$$= \nabla\mathcal{L}\left(w_0^{(Mj)}\right)^\top\left(w_0^{(Mj+M)} - w_0^{(Mj)} + \eta K\nabla\mathcal{L}\left(w_0^{(Mj)}\right) - \eta K\nabla\mathcal{L}\left(w_0^{(Mj)}\right)\right)$$

$$\leq -\eta K \left\| \nabla \mathcal{L}\left(\boldsymbol{w}_0^{(Mj)}\right) \right\|^2 + \left\| \nabla \mathcal{L}\left(\boldsymbol{w}_0^{(Mj)}\right) \right\| \left\| \boldsymbol{w}_0^{(Mj+M)} - \boldsymbol{w}_0^{(Mj)} + \eta K \nabla \mathcal{L}\left(\boldsymbol{w}_0^{(Mj)}\right) \right\|.$$

By Lemma E.2,

$$\mathcal{L}(\boldsymbol{w}_0^{(Mj+M)}) - \mathcal{L}(\boldsymbol{w}_0^{(Mj)}) - \frac{\sigma_{\max}^2 \beta}{2} \cdot \frac{(\eta \sigma_{\max} K)^2}{\phi^2(1 - \eta M K \sigma_{\max}^2 \beta)^2} \left\| \nabla \mathcal{L}(\boldsymbol{w}_0^{(Mj)}) \right\|^2$$

$$\leq -\eta K \left\| \nabla \mathcal{L}(\boldsymbol{w}_0^{(Mj)}) \right\|^2 + \frac{\eta^2 M K^2 \sigma_{\max}^3 \beta}{\phi(1 - \eta M K \sigma_{\max}^2 \beta)} \left\| \nabla \mathcal{L}(\boldsymbol{w}_0^{(Mj)}) \right\|^2.$$

Given that $\eta \leq \frac{1}{2MK\sigma_{\max}^2\beta}$,

$$\mathcal{L}(\boldsymbol{w}_0^{(Mj+M)}) - \mathcal{L}(\boldsymbol{w}_0^{(Mj)})$$

$$\leq -\eta K \left\{ 1 - \eta K \left( \frac{M\sigma_{\max}^3\beta}{\phi(1 - \eta M K \sigma_{\max}^2 \beta)} + \frac{\sigma_{\max}^4\beta}{2\phi^2(1 - \eta M K \sigma_{\max}^2\beta)^2} \right) \right\} \left\| \nabla \mathcal{L}(\boldsymbol{w}_0^{(Mj)}) \right\|^2$$

$$\leq -\eta K \left( 1 - \eta K \frac{2(M\phi + \sigma_{\max})\sigma_{\max}^3\beta}{\phi^2} \right) \left\| \nabla \mathcal{L}(\boldsymbol{w}_0^{(Mj)}) \right\|^2$$

$$= -\eta K \left( 1 - \eta K \beta' \right) \left\| \nabla \mathcal{L}(\boldsymbol{w}_0^{(Mj)}) \right\|^2, \tag{20}$$

where we set $\beta' := \frac{2(M\phi+\sigma_{\max})\sigma_{\max}^3\beta}{\phi^2}$. Given that $\eta < \frac{1}{K\beta'}$, $\mathcal{L}(\boldsymbol{w}_0^{(Mj+M)}) < \mathcal{L}(\boldsymbol{w}_0^{(Mj)})$ holds. Note that $\frac{1}{K\beta'} < \frac{1}{2MK\sigma_{\max}^2\beta}$ because, in the proof of Lemma E.1, we can show that

$$\phi \leq \min_{\boldsymbol{v} \in \mathbb{R}_{\geq 0}^d : \|\boldsymbol{v}\|=1} \|\boldsymbol{X}\boldsymbol{v}\| \leq \max_{\boldsymbol{v} \in \mathbb{R}_{\geq 0}^d : \|\boldsymbol{v}\|=1} \|\boldsymbol{X}\boldsymbol{v}\| \leq \sigma_{\max}.$$

Also, by Equation (20),

$$\sum_{j=0}^{\infty} \left\| \nabla \mathcal{L}(\boldsymbol{w}_0^{(Mj)}) \right\|^2 \leq \frac{\mathcal{L}(\boldsymbol{w}_0^{(0)}) - \lim_{t\to\infty}\mathcal{L}(\boldsymbol{w}_0^{(Mj)})}{\eta K(1 - \eta K\beta')} \leq \frac{\mathcal{L}(\boldsymbol{w}_0^{(0)})}{\eta K(1 - \eta K\beta')} < \infty.$$

Coupled with Lemma E.2,

$$\sum_{j=0}^{\infty} \sum_{m=0}^{M-1} \sum_{k=0}^{K-1} \left\| \nabla \mathcal{L}(\boldsymbol{w}_k^{(Mj+m)}) \right\|^2$$

$$\leq \sum_{j=0}^{\infty} \sum_{m=0}^{M-1} \sum_{k=0}^{K-1} \left( \left\| \nabla \mathcal{L}(\boldsymbol{w}_0^{(Mj)}) \right\| + \left\| \nabla \mathcal{L}(\boldsymbol{w}_k^{(Mj+m)}) - \nabla \mathcal{L}(\boldsymbol{w}_0^{(Mj)}) \right\| \right)^2$$

$$\leq \sum_{j=0}^{\infty} \sum_{m=0}^{M-1} \sum_{k=0}^{K-1} \left( 1 + \frac{\eta K \sigma_{\max}^3\beta}{\phi\{1 - \eta(mK+k)\sigma_{\max}^2\beta\}} \right)^2 \left\| \nabla \mathcal{L}(\boldsymbol{w}_0^{(Mj)}) \right\|^2$$

$$\leq \left( 1 + \frac{\eta K \sigma_{\max}^3\beta}{\phi\{1 - \eta M K \sigma_{\max}^2\beta\}} \right)^2 MK \sum_{j=0}^{\infty} \left\| \nabla \mathcal{L}(\boldsymbol{w}_0^{(Mj)}) \right\|^2 < \infty.$$

The boundedness of infinite sum of nonzero elements means $\lim_{t\to\infty} \left\| \nabla \mathcal{L}(\boldsymbol{w}_k^{(t)}) \right\|^2 = 0, \forall k \in [0 : K-1]$. This leads to $\lim_{t\to\infty} \ell'(\boldsymbol{x}_i^\top \boldsymbol{w}_k^{(t)}) = 0, \forall i \in I, k \in [0 : K-1]$ by Lemma E.1. Since $\ell'(u) \to 0$ only when $u \to \infty$, we obtain $\boldsymbol{x}_i^\top \boldsymbol{w}_k^{(t)} \to \infty, \forall i \in I, k \in [0 : K-1]$ and $\lim_{t\to\infty} \mathcal{L}(\boldsymbol{w}_k^{(t)}) = 0, \forall k \in [0 : K-1]$. Finally, we obtain that $\sum_{j=0}^{\infty} \sum_{k=0}^{K-1} \left\| \boldsymbol{w}_{k+1}^{(t)} - \boldsymbol{w}_k^{(t)} \right\|^2 < \infty$ followed by

$$\left\| \nabla \mathcal{L}(\boldsymbol{w}_k^{(t)}) \right\| \geq \phi \sqrt{\sum_{i\in I} \left[ \ell'(\boldsymbol{x}_i^\top \boldsymbol{w}_k^{(t)}) \right]^2} \geq \phi \sqrt{\sum_{i\in I^{(t)}} \left[ \ell'(\boldsymbol{x}_i^\top \boldsymbol{w}_k^{(t)}) \right]^2}$$

$$\geq \frac{\phi}{\beta_{m_t}} \left\| \sum_{i\in I^{(t)}} \ell'(\boldsymbol{x}_i^\top \boldsymbol{w}_k^{(t)}) \boldsymbol{x}_i \right\| = \frac{\phi}{\beta_{m_t}\eta} \left\| \boldsymbol{w}_{k+1}^{(t)} - \boldsymbol{w}_k^{(t)} \right\|,$$

where we use Lemma E.1 in the first inequality, while we use the fact $\forall \lambda_s \in \mathbb{R} : \left\| \sum_{s\in I} \lambda_s \boldsymbol{x}_s \right\|_2 \leq \sigma_{\max} \sqrt{\sum_{s\in I} \lambda_s^2}$ in the third inequality. The last equality is true by the definition of gradient descent.

### E.1.1 PROOF OF LEMMA E.1

Let us recall the statement of the lemma for the sake of readability.

**Lemma E.1.** *Suppose that Assumptions 3.1 and 3.3 holds. For any $\boldsymbol{w} \in \mathbb{R}^d$,*

$$\|\nabla\mathcal{L}(\boldsymbol{w})\| \geq \phi\sqrt{\sum_{i\in I}\left[\ell'(\boldsymbol{x}_i^\top\boldsymbol{w})\right]^2}.$$

Let any $\boldsymbol{w} \in \mathbb{R}^d$. Observe that if we let $\boldsymbol{v}' \in \mathbb{R}_{\geq 0}^N$ be a vector whose $i$-th entry is $-\ell'(\boldsymbol{x}_i^\top\boldsymbol{w})$, then $-\nabla\mathcal{L}(\boldsymbol{w}) = \boldsymbol{X}\boldsymbol{v}'$. Because of this,

$$
\begin{aligned}
\|\nabla\mathcal{L}(\boldsymbol{w})\| &= \|\boldsymbol{X}\boldsymbol{v}'\| \\
&\geq \|\boldsymbol{v}'\| \cdot \min_{\boldsymbol{v}\in\mathbb{R}_{\geq0}^N:\|\boldsymbol{v}\|=1}\|\boldsymbol{X}\boldsymbol{v}\| \\
&= \sqrt{\sum_{i\in I}\left[\ell'(\boldsymbol{x}_i^\top\boldsymbol{w})\right]^2} \cdot \min_{\boldsymbol{v}\in\mathbb{R}_{\geq0}^N:\|\boldsymbol{v}\|=1}\|\boldsymbol{X}\boldsymbol{v}\|.
\end{aligned}
$$

Let $\hat{\boldsymbol{v}} := \arg\min_{\boldsymbol{v}\in\mathbb{R}_{\geq0}^N:\|\boldsymbol{v}\|=1}\|\boldsymbol{X}\boldsymbol{v}\|$. Then for max-margin direction $\hat{\boldsymbol{w}}$, the following holds.

$$\|\boldsymbol{X}\hat{\boldsymbol{v}}\| \geq \frac{\hat{\boldsymbol{w}}^\top\boldsymbol{X}\hat{\boldsymbol{v}}}{\|\hat{\boldsymbol{w}}\|} \geq \phi\|\hat{\boldsymbol{v}}\| = \phi.$$

We used Cauchy-Schwarz for the first inequality and the definition of $\hat{\boldsymbol{w}}$ for the second one. This concludes the proof of the lemma.

### E.1.2 PROOF OF LEMMA E.2

We restate the lemma here for readability.

**Lemma E.2.** *Suppose that Assumptions 3.1 and 3.3 holds. Let any $t \in \mathbb{N}, m \in [0 : M - 1], k \in [0 : K - 1]$. Then, if we choose the step size as $\eta < \frac{1}{(mK+k)\sigma_{\max}^2\beta}$, we have*

$$
\left\|\boldsymbol{w}_k^{(t+m)} - \boldsymbol{w}_0^{(t)} + \eta\left(K\sum_{i=0}^{m-1}\nabla\mathcal{L}^{(t+i)}(\boldsymbol{w}_0^{(t)}) + k\nabla\mathcal{L}^{(t+m)}(\boldsymbol{w}_0^{(t)})\right)\right\|
$$
$$
\leq \frac{\eta^2(mK+k)K\sigma_{\max}^3\beta}{\phi\{1-\eta(mK+k)\sigma_{\max}^2\beta\}}\left\|\nabla\mathcal{L}(\boldsymbol{w}_0^{(t)})\right\|,
$$
$$
\left\|\boldsymbol{w}_k^{(t+m)} - \boldsymbol{w}_0^{(t)}\right\| \leq \frac{\eta K\sigma_{\max}}{\phi\{1-\eta(mK+k)\sigma_{\max}^2\beta\}}\left\|\nabla\mathcal{L}(\boldsymbol{w}_0^{(t)})\right\|,
$$
$$
\left\|\nabla\mathcal{L}(\boldsymbol{w}_k^{(t+m)}) - \nabla\mathcal{L}(\boldsymbol{w}_0^{(t)})\right\| \leq \frac{\eta K\sigma_{\max}^3\beta}{\phi\{1-\eta(mK+k)\sigma_{\max}^2\beta\}}\left\|\nabla\mathcal{L}(\boldsymbol{w}_0^{(t)})\right\|.
$$

To start the proof, observe that

$$\boldsymbol{w}_k^{(t+m)} = \boldsymbol{w}_0^{(t)} - \eta\left(\sum_{i=0}^{m-1}\sum_{h=0}^{K-1}\nabla\mathcal{L}^{(t+i)}(\boldsymbol{w}_h^{(t+i)}) + \sum_{h'=0}^{k-1}\nabla\mathcal{L}^{(t+m)}(\boldsymbol{w}_{h'}^{(t+m)})\right).$$

for all $t \geq 0$, $m \in [0 : M]$, $k \in [0 : K]$. Using this, we can deduce the following with $\beta_{\max}$-smoothness and triangle inequality.

$$
\left\|\boldsymbol{w}_k^{(t+m)} - \boldsymbol{w}_0^{(t)} + \eta\left(K\sum_{i=0}^{m-1}\nabla\mathcal{L}^{(t+i)}(\boldsymbol{w}_0^{(t)}) + k\nabla\mathcal{L}^{(t+m)}(\boldsymbol{w}_0^{(t)})\right)\right\|
$$
$$
= \left\|\eta\sum_{i=0}^{m-1}\sum_{h=0}^{K-1}\left(\nabla\mathcal{L}^{(t+i)}(\boldsymbol{w}_0^{(t)}) - \nabla\mathcal{L}^{(t+i)}(\boldsymbol{w}_h^{(t+i)})\right) + \eta\sum_{h'=0}^{k-1}\left(\nabla\mathcal{L}^{(t+m)}(\boldsymbol{w}_0^{(t)}) - \nabla\mathcal{L}^{(t+m)}(\boldsymbol{w}_{h'}^{(t+m)})\right)\right\|
$$
$$
\leq \eta\sum_{i=0}^{m-1}\sum_{h=0}^{K-1}\left\|\nabla\mathcal{L}^{(t+i)}(\boldsymbol{w}_0^{(t)}) - \nabla\mathcal{L}^{(t+i)}(\boldsymbol{w}_h^{(t+i)})\right\| + \eta\sum_{h'=0}^{k-1}\left\|\nabla\mathcal{L}^{(t+m)}(\boldsymbol{w}_0^{(t)}) - \nabla\mathcal{L}^{(t+m)}(\boldsymbol{w}_{h'}^{(t+m)})\right\|
$$

$$\leq \eta\beta_{\max} \left( \sum_{i=0}^{m-1} \sum_{h=0}^{K-1} \left\| \boldsymbol{w}_0^{(t)} - \boldsymbol{w}_h^{(t+i)} \right\| + \sum_{h'=0}^{k-1} \left\| \boldsymbol{w}_0^{(t)} - \boldsymbol{w}_{h'}^{(t+m)} \right\| \right). \tag{21}$$

Moreover, using the fact $\forall \lambda_s \in \mathbb{R} : \left\| \sum_{s \in I} \lambda_s \boldsymbol{x}_s \right\|_2 \leq \sigma_{\max} \sqrt{\sum_{s \in I} \lambda_s^2}$,

$$\left\| \boldsymbol{w}_k^{(t+m)} - \boldsymbol{w}_0^{(t)} \right\|$$

$$\leq \left\| -\eta \left( K \sum_{i=0}^{m-1} \nabla \mathcal{L}^{(t+i)}(\boldsymbol{w}_0^{(t)}) + k \nabla \mathcal{L}^{(t+m)}(\boldsymbol{w}_0^{(t)}) \right) \right\|$$

$$+ \left\| \boldsymbol{w}_k^{(t+m)} - \boldsymbol{w}_0^{(t)} + \eta \left( K \sum_{i=0}^{m-1} \nabla \mathcal{L}^{(t+i)}(\boldsymbol{w}_0^{(t)}) + k \nabla \mathcal{L}^{(t+m)}(\boldsymbol{w}_0^{(t)}) \right) \right\|$$

$$\leq \eta \left\| K \sum_{i=0}^{m-1} \sum_{s \in I^{(t+i)}} \ell'(\boldsymbol{x}_s^\top \boldsymbol{w}_0^{(t)}) \boldsymbol{x}_s + k \sum_{s \in I^{(t+m)}} \ell'(\boldsymbol{x}_s^\top \boldsymbol{w}_0^{(t)}) \boldsymbol{x}_s \right\|$$

$$+ \left\| \boldsymbol{w}_k^{(t+m)} - \boldsymbol{w}_0^{(t)} + \eta \left( K \sum_{i=0}^{m-1} \nabla \mathcal{L}^{(t+i)}(\boldsymbol{w}_0^{(t)}) + k \nabla \mathcal{L}^{(t+m)}(\boldsymbol{w}_0^{(t)}) \right) \right\|$$

$$\leq \eta \sigma_{\max} \sqrt{ \sum_{i=0}^{m-1} \sum_{s \in I^{(t+i)}} \left( K \ell'(\boldsymbol{x}_s^\top \boldsymbol{w}_0^{(t)}) \right)^2 + \sum_{s \in I^{(t+m)}} \left( k \ell'(\boldsymbol{x}_s^\top \boldsymbol{w}_0^{(t)}) \right)^2 }$$

$$+ \left\| \boldsymbol{w}_k^{(t+m)} - \boldsymbol{w}_0^{(t)} + \eta \left( K \sum_{i=0}^{m-1} \nabla \mathcal{L}^{(t+i)}(\boldsymbol{w}_0^{(t)}) + k \nabla \mathcal{L}^{(t+m)}(\boldsymbol{w}_0^{(t)}) \right) \right\|$$

$$\leq \eta K \sigma_{\max} \sqrt{ \sum_{s \in I} \left( \ell'(\boldsymbol{x}_s^\top \boldsymbol{w}_0^{(t)}) \right)^2 } + \left\| \boldsymbol{w}_k^{(t+m)} - \boldsymbol{w}_0^{(t)} + \eta \left( K \sum_{i=0}^{m-1} \nabla \mathcal{L}^{(t+i)}(\boldsymbol{w}_0^{(t)}) + k \nabla \mathcal{L}^{(t+m)}(\boldsymbol{w}_0^{(t)}) \right) \right\|.$$

Thus, by Equation (21) and Lemma E.1, we obtain

$$\left\| \boldsymbol{w}_k^{(t+m)} - \boldsymbol{w}_0^{(t)} \right\|$$

$$\leq \frac{\eta K \sigma_{\max}}{\phi} \left\| \nabla \mathcal{L}(\boldsymbol{w}_0^{(t)}) \right\| + \eta \sigma_{\max}^2 \beta \left( \sum_{i=0}^{m-1} \sum_{j=0}^{K-1} \left\| \boldsymbol{w}_j^{(t+i)} - \boldsymbol{w}_0^{(t)} \right\| + \sum_{j=0}^{k-1} \left\| \boldsymbol{w}_j^{(t+m)} - \boldsymbol{w}_0^{(t)} \right\| \right). \tag{22}$$

Here, we use the following technical lemma.

**Lemma E.3** (Nacson et al., 2019). *Let $\epsilon$ and $\theta$ be positive constants. Suppose $\delta_k \leq \theta + \epsilon \sum_{u=0}^{k-1} \delta_u$ holds for all non-negative integers $k < \frac{1}{\epsilon}$. Then*

$$\delta_k \leq \frac{\theta}{1 - k\epsilon} \qquad and \qquad \sum_{u=0}^{k-1} \delta_u \leq \frac{k\theta}{1 - k\epsilon}.$$

By applying the lemma to (22), we obtain

$$\left\| \boldsymbol{w}_k^{(t+m)} - \boldsymbol{w}_0^{(t)} \right\| \leq \frac{\eta K \sigma_{\max}}{\phi \{ 1 - \eta(mK + k)\sigma_{\max}^2 \beta \}} \left\| \nabla \mathcal{L}(\boldsymbol{w}_0^{(t)}) \right\|$$

and

$$\left\| \boldsymbol{w}_k^{(t+m)} - \boldsymbol{w}_0^{(t)} + \eta \left( K \sum_{i=0}^{m-1} \nabla \mathcal{L}^{(t+i)}(\boldsymbol{w}_0^{(t)}) + k \nabla \mathcal{L}^{(t+m)}(\boldsymbol{w}_0^{(t)}) \right) \right\|$$

$$\leq \eta \sigma_{\max}^2 \beta \left( \sum_{i=0}^{m-1} \sum_{j=0}^{K-1} \left\| \boldsymbol{w}_j^{(t+i)} - \boldsymbol{w}_0^{(t)} \right\| + \sum_{j=0}^{k-1} \left\| \boldsymbol{w}_j^{(t+m)} - \boldsymbol{w}_0^{(t)} \right\| \right)$$

$$\leq \frac{\eta^2(mK+k)K\sigma_{\max}^3\beta}{\phi\{1-\eta(mK+k)\sigma_{\max}^2\beta\}}\left\|\nabla\mathcal{L}(\boldsymbol{w}_0^{(t)})\right\|.$$

Finally, by smoothness,

$$\left\|\nabla\mathcal{L}(\boldsymbol{w}_k^{(t+m)}) - \nabla\mathcal{L}(\boldsymbol{w}_0^{(t)})\right\| \leq \sigma_{\max}^2\beta\left\|\boldsymbol{w}_k^{(t+m)} - \boldsymbol{w}_0^{(t)}\right\|$$

$$\leq \frac{\eta K\sigma_{\max}^3\beta}{\phi\{1-\eta(mK+k)\sigma_{\max}^2\beta\}}\left\|\nabla\mathcal{L}(\boldsymbol{w}_0^{(t)})\right\|,$$

which concludes the proof of the lemma.

### E.2 DIRECTIONAL CONVERGENCE ANALYSIS (PROOF OF THEOREM 3.2)

In this section, we prove our implicit bias result: Theorem 3.2. Moreover, we further discuss the convergence of the residual vector $\boldsymbol{\rho}_k^{(t)}$, beyond its boundedness, under some additional assumption on the dataset (see Appendix E.2.3).

**Theorem 3.2.** *Let $\{\boldsymbol{w}_k^{(t)}\}_{k\in[0:K-1],t\geq0}$ be the sequence of GD iterates (2) from any initial point $\boldsymbol{w}_0^{(0)}$, where tasks are given cyclically. Suppose that Assumptions 3.1, 3.2, 3.3, and 3.4 hold. Then, under the same learning rate condition as in Theorem 3.1, $\boldsymbol{w}_k^{(t)}$ will behave as*

$$\boldsymbol{w}_k^{(t)} = \ln(t)\hat{\boldsymbol{w}} + \boldsymbol{\rho}_k^{(t)},$$

*for all sufficiently large $t$, where $\|\boldsymbol{\rho}_k^{(t)}\|$ stays bounded as $t\to\infty$.*

Note that we use Assumption 3.2, the unique existence of SVM dual variables $\boldsymbol{\alpha}$ that satisfies

$$\hat{\boldsymbol{w}} = \sum_{s\in S}\alpha_s\boldsymbol{x}_s,$$

$$\forall s\in S:\alpha_s>0, \forall s\notin S:\alpha_s=0.$$

This assumption holds for almost all data (Soudry et al., 2018).

When the tasks are given in a cyclic order, the following lemma holds. Note that the lemma does not depend on the algorithm.

**Lemma E.4.** *When tasks are given cyclic, there exists $\check{\boldsymbol{w}}, \boldsymbol{m}_{t,k}\in\mathbb{R}^d$ the following holds for all $t\in\mathbb{N}$, $k\in[0:K-1]$.*

$$K\sum_{u=1}^{t-1}\frac{1}{u}\sum_{s\in S^{(u)}}\alpha_s\boldsymbol{x}_s + \frac{k}{t}\sum_{s\in S^{(t)}}\alpha_s\boldsymbol{x}_s = \frac{K}{M}\ln\left(\frac{t}{M}\right)\hat{\boldsymbol{w}} + \frac{K}{M}\check{\boldsymbol{w}} + \boldsymbol{m}_{t,k},$$

$$\boldsymbol{m}_{t,K} := \boldsymbol{m}_{t+1,0},$$

*such that $\|\boldsymbol{m}_{t,k}\| = o(t^{-0.5+\epsilon})$, and $\|\boldsymbol{m}_{t,k+1} - \boldsymbol{m}_{t,k}\| = \mathcal{O}(t^{-1})$ for all $k\in[0:K-1], \epsilon>0$, and $\check{\boldsymbol{w}}$ only depends on the order of tasks and constant with respect to $t$.*

*Proof.* See Appendix E.2.1. $\qquad\square$

We set $\boldsymbol{m}_{t,k}$ and $\check{\boldsymbol{w}}$ along Lemma E.4, and define $\boldsymbol{\rho}_k^{(t)}$ and $\boldsymbol{r}_k^{(t)}$ as

$$\begin{aligned}
\boldsymbol{w}_k^{(t)} &= \ln(t)\hat{\boldsymbol{w}} + \boldsymbol{\rho}_k^{(t)} \\
&= \ln(t)\hat{\boldsymbol{w}} + \ln\left(\frac{K}{M}\right)\hat{\boldsymbol{w}} + \tilde{\boldsymbol{w}} + \frac{M}{K}\boldsymbol{m}_{t,k} + \boldsymbol{r}_k^{(t)},
\end{aligned} \tag{23}$$

$$\boldsymbol{\rho}_K^{(t)} = \boldsymbol{\rho}_0^{(t+1)}, \quad \boldsymbol{r}_K^{(t)} = \boldsymbol{r}_0^{(t+1)},$$

where $\tilde{w}$ is the solution of

$$\forall i \in S : \eta \exp\left(-\boldsymbol{x}_i^\top \tilde{\boldsymbol{w}}\right) = \alpha_i, \qquad \bar{P}(\tilde{\boldsymbol{w}} - \boldsymbol{w}_0^{(0)}) = 0,$$

which is unique under Assumption 3.2. Then by the definition,

$$\boldsymbol{r}_k^{(t)} = \boldsymbol{w}_k^{(t)} - \frac{M}{K}\left(\frac{K}{M}\ln\left(\frac{K}{M}t\right)\hat{\boldsymbol{w}} + \boldsymbol{m}_{t,k}\right) - \tilde{\boldsymbol{w}}$$

$$= \boldsymbol{w}_k^{(t)} - \frac{M}{K}\left(K\sum_{u=1}^{t-1}\frac{1}{u}\sum_{s\in S^{(u)}}\alpha_s\boldsymbol{x}_s + \frac{k}{t}\sum_{s\in S^{(t)}}\alpha_s\boldsymbol{x}_s\right) - \ln K\hat{\boldsymbol{w}} - \tilde{\boldsymbol{w}} + \check{\boldsymbol{w}}.$$

Under these definitions, we can get the primary lemma of $\boldsymbol{r}_k^{(t)}$.

**Lemma E.5.** *Under Assumption 3.1, 3.3, 3.4, and Assumption 3.2, if learning rate is $\eta < \min\{\frac{1}{2MK\beta\sigma_{\max}^2}, \frac{\phi^2}{4K\beta\sigma_{\max}^3(M\phi+\sigma_{\max})}\}$, then*

*1. $\exists \tilde{t}, C_1, C_2 > 0$ such that $\forall t > \tilde{t}$,*

$$(\boldsymbol{r}_{k+1}^{(t)} - \boldsymbol{r}_k^{(t)})^\top \boldsymbol{r}_k^{(t)} \leq C_1 t^{-\theta} + C_2 t^{-1-0.5\tilde{\mu}}, \forall k \in [0:K-1].$$

*2. Moreover, for all $\epsilon_1 > 0$, $\exists \tilde{t}^*, C_3 > 0$ such that if $\left\|P\boldsymbol{r}_k^{(t)}\right\| \geq \epsilon_1$ and $S^{(t)} \neq \emptyset$,*

$$(\boldsymbol{r}_{k+1}^{(t)} - \boldsymbol{r}_k^{(t)})^\top \boldsymbol{r}_k^{(t)} \leq -C_3 t^{-1}, \forall t > \tilde{t}^*, k \in [0:K-1].$$

*Proof.* See Appendix E.2.2. The proof sketch is provided in Appendix D.2.1. $\square$

By the definition of $\rho_k^{(t)}$, it is enough to prove $\left\|\boldsymbol{r}_k^{(t)}\right\|$ is bounded above in $t$. We use the fact

$$\left\|\boldsymbol{r}_{k+1}^{(t)}\right\|^2 - \left\|\boldsymbol{r}_k^{(t)}\right\|^2 = 2(\boldsymbol{r}_{k+1}^{(t)} - \boldsymbol{r}_k^{(t)})^\top \boldsymbol{r}_k^{(t)} + \left\|\boldsymbol{r}_{k+1}^{(t)} - \boldsymbol{r}_k^{(t)}\right\|^2$$

to show the boundedness of $\left\|\boldsymbol{r}_k^{(t)}\right\|$. For all $k \in [0:K-2]$, let $\boldsymbol{a}_k^{(t)} := \frac{M}{K}(\boldsymbol{m}_{t,k+1} - \boldsymbol{m}_{t,k})$. And let $\boldsymbol{a}_{K-1}^{(t)} := \ln(1+\frac{1}{t})\hat{\boldsymbol{w}} + \frac{M}{K}(\boldsymbol{m}_{t+1,0} - \boldsymbol{m}_{t,K-1})$. Since $\boldsymbol{w}_k^{(t)} = \ln\left(\frac{K}{M}t\right)\hat{\boldsymbol{w}} + \tilde{\boldsymbol{w}} + \frac{M}{K}\boldsymbol{m}_{t,k} + \boldsymbol{r}_k^{(t)}$, $\left\|\boldsymbol{r}_{k+1}^{(t)} - \boldsymbol{r}_k^{(t)}\right\|^2 = \left\|\boldsymbol{w}_{k+1}^{(t)} - \boldsymbol{w}_k^{(t)} - \boldsymbol{a}_k^{(t)}\right\|^2$. Also, by Lemma E.4, $\left\|\boldsymbol{a}_k^{(t)}\right\| = \mathcal{O}(t^{-1})$. Thus, $\exists t_1$ such that $\forall t \geq t_1, \forall k \in [0:K-1] : \left\|\boldsymbol{a}_k^{(t)}\right\| \leq t^{-1}$.

Now we can get the following for all $T \geq t_1$.

$$\sum_{t=t_1}^{T}\sum_{k=0}^{K-1}\left\|\boldsymbol{r}_{k+1}^{(t)} - \boldsymbol{r}_k^{(t)}\right\|^2 = \sum_{t=t_1}^{T}\sum_{k=0}^{K-1}\left\|\boldsymbol{w}_{k+1}^{(t)} - \boldsymbol{w}_k^{(t)} - \boldsymbol{a}_k^{(t)}\right\|^2$$

$$= \sum_{t=t_1}^{T}\sum_{k=0}^{K-1}\left\|\boldsymbol{w}_{k+1}^{(t)} - \boldsymbol{w}_k^{(t)}\right\|^2 + \sum_{t=t_1}^{T}\sum_{k=0}^{K-1}2(\boldsymbol{w}_k^{(t)} - \boldsymbol{w}_{k+1}^{(t)})^\top \boldsymbol{a}_k^{(t)} + \sum_{t=t_1}^{T}\sum_{k=0}^{K-1}\left\|\boldsymbol{a}_k^{(t)}\right\|^2$$

$$\leq \sum_{t=t_1}^{T}\sum_{k=0}^{K-1}\left\|\boldsymbol{w}_{k+1}^{(t)} - \boldsymbol{w}_k^{(t)}\right\|^2 + 2\sqrt{\sum_{t=t_1}^{T}\sum_{k=0}^{K-1}\left\|\boldsymbol{w}_k^{(t)} - \boldsymbol{w}_{k+1}^{(t)}\right\|^2 \sum_{t=t_1}^{T}\sum_{k=0}^{K-1}\left\|\boldsymbol{a}_k^{(t)}\right\|^2} + \sum_{t=t_1}^{T}\sum_{k=0}^{K-1}\left\|\boldsymbol{a}_k^{(t)}\right\|^2$$

$$\leq \sum_{t=t_1}^{T}\sum_{k=0}^{K-1}\left\|\boldsymbol{w}_{k+1}^{(t)} - \boldsymbol{w}_k^{(t)}\right\|^2 + 2\sqrt{\sum_{t=t_1}^{T}\sum_{k=0}^{K-1}\left\|\boldsymbol{w}_k^{(t)} - \boldsymbol{w}_{k+1}^{(t)}\right\|^2 \sum_{t=t_1}^{T}\sum_{k=0}^{K-1}t^{-2}} + \sum_{t=t_1}^{T}\sum_{k=0}^{K-1}t^{-2}$$

$$< \infty. \tag{24}$$

We use Cauchy-Schwarz inequality for the first inequality and the factor that $\sum_{t=t_1}^{T}t^{-2} < \infty$ and $\sum_{t=t_1}^{T}\sum_{k=0}^{K-1}\left\|\boldsymbol{w}_k^{(t)} - \boldsymbol{w}_{k+1}^{(t)}\right\|^2 < \infty$ by Theorem 3.1.

Combined with Lemma E.5 and the fact that $\forall c > 1 : \sum_{t=1}^{\infty} t^{-c} < \infty$, we get

$$
\left\| \boldsymbol{r}_0^{(t)} \right\|^2 - \left\| \boldsymbol{r}_0^{(t_1)} \right\|^2 = \sum_{u=t_1}^{t-1} \sum_{k=0}^{K-1} \left( \left\| \boldsymbol{r}_{k+1}^{(u)} \right\|^2 - \left\| \boldsymbol{r}_k^{(u)} \right\|^2 \right)
$$

$$
= \sum_{u=t_1}^{t-1} \sum_{k=0}^{K-1} \left( 2(\boldsymbol{r}_{k+1}^{(u)} - \boldsymbol{r}_k^{(u)})^\top \boldsymbol{r}_k^{(u)} + \left\| \boldsymbol{r}_{k+1}^{(u)} - \boldsymbol{r}_k^{(u)} \right\|^2 \right) < \infty,
$$

hence $\left\| \boldsymbol{r}_k^{(t)} \right\|$ is bounded.

### E.2.1 Proof of Lemma E.4

$$
K \sum_{u=1}^{t-1} \frac{1}{u} \sum_{s \in S^{(u)}} \alpha_s \boldsymbol{x}_s + \frac{k}{t} \sum_{s \in S^{(t)}} \alpha_s \boldsymbol{x}_s
$$

$$
= K \sum_{u=1}^{\lfloor \frac{t-1}{M} \rfloor M} \frac{1}{u} \sum_{s \in S^{(u)}} \alpha_s \boldsymbol{x}_s + \underbrace{K \sum_{u=\lfloor \frac{t-1}{M} \rfloor M + 1}^{t-1} \frac{1}{u} \sum_{s \in S^{(u)}} \alpha_s \boldsymbol{x}_s + \frac{k}{t} \sum_{s \in S^{(t)}} \alpha_s \boldsymbol{x}_s}_{=: \boldsymbol{m}'_{t,k}}
$$

$$
= K \sum_{u=1}^{\lfloor \frac{t-1}{M} \rfloor M} \frac{1}{u} \sum_{s \in S^{(u)}} \alpha_s \boldsymbol{x}_s + \boldsymbol{m}'_{t,k}
$$

$$
= K \sum_{u=1}^{\lfloor \frac{t-1}{M} \rfloor} \left[ \sum_{v=1}^{M} \frac{1}{v + M(u-1)} \left( \sum_{s \in S^{(v)}} \alpha_s \boldsymbol{x}_s \right) \right] + \boldsymbol{m}'_{t,k}
$$

$$
= K \sum_{v=1}^{M} \left[ \sum_{u=1}^{\lfloor \frac{t-1}{M} \rfloor} \frac{1}{v + M(u-1)} \left( \sum_{s \in S^{(v)}} \alpha_s \boldsymbol{x}_s \right) \right] + \boldsymbol{m}'_{t,k}.
$$

Note that $\boldsymbol{m}'_{t,k}$ and $\boldsymbol{m}'_{t,k+1} - \boldsymbol{m}'_{t,k}$ are both $\mathcal{O}(t^{-1})$ for all $k \in [0 : K-1]$. For every $v$,

$$
\sum_{u=1}^{\lfloor \frac{t-1}{M} \rfloor} \frac{1}{v + M(u-1)} \left( \sum_{s \in S^{(v)}} \alpha_s \boldsymbol{x}_s \right)
$$

$$
= \sum_{u=1}^{\lfloor \frac{t-1}{M} \rfloor} \left[ \frac{1}{Mu} + \frac{1 - \frac{v}{M}}{Mu^2 + (v-M)u} \right] \left( \sum_{s \in S^{(v)}} \alpha_s \boldsymbol{x}_s \right)
$$

$$
= \left[ \frac{1}{M} \left( \ln \left( \left\lfloor \frac{t-1}{M} \right\rfloor \right) + \gamma + \mathcal{O}(t^{-1}) \right) + \sum_{u=1}^{\lfloor \frac{t-1}{M} \rfloor} \frac{1 - \frac{v}{M}}{Mu^2 + (v-M)u} \right] \left( \sum_{s \in S^{(v)}} \alpha_s \boldsymbol{x}_s \right)
$$

$$
= \left[ \frac{1}{M} \left( \ln \left( \frac{t-1}{M} \right) + \gamma + \mathcal{O}(t^{-1}) \right) + \sum_{u=1}^{\lfloor \frac{t-1}{M} \rfloor} \frac{1 - \frac{v}{M}}{Mu^2 + (v-M)u} \right] \left( \sum_{s \in S^{(v)}} \alpha_s \boldsymbol{x}_s \right)
$$

$$
= \left[ \frac{1}{M} \left( \ln \left( \frac{t}{M} \right) + \gamma + \mathcal{O}(t^{-1}) \right) + \sum_{u=1}^{\lfloor \frac{t-1}{M} \rfloor} \frac{1 - \frac{v}{M}}{Mu^2 + (v-M)u} \right] \left( \sum_{s \in S^{(v)}} \alpha_s \boldsymbol{x}_s \right),
$$

where, in the last three equalities, we use the fact

$$
\sum_{u=1}^{t} \frac{1}{u} = \ln t + \gamma + \mathcal{O}(t^{-1}),
$$

$$\ln(t) - \ln(\lfloor t \rfloor) = \mathcal{O}(t^{-1}),$$
$$\ln(t) - \ln(t-1) = \mathcal{O}(t^{-1}),$$

where $\gamma$ is the Euler-Mascheroni constant. Since $1 \le v \le M$, the inequality $\frac{1-\frac{v}{M}}{Mu^2+(v-M)u} \le \frac{1-\frac{v}{M}}{vu^2}$ holds. Therefore, the series $\sum_u \frac{1-\frac{v}{M}}{Mu^2+(v-M)u}$ converges with a rate $\mathcal{O}(t^{-1})$.

$$\sum_{u=1}^{\lfloor \frac{t-1}{M} \rfloor} \frac{1-\frac{v}{M}}{Mu^2+(v-M)u} = \sum_{u=1}^{\infty} \frac{1-\frac{v}{M}}{Mu^2+(v-M)u} - \sum_{u=\lfloor \frac{t-1}{M} \rfloor +1}^{\infty} \frac{1-\frac{v}{M}}{Mu^2+(v-M)u}$$
$$= \sum_{u=1}^{\infty} \frac{1-\frac{v}{M}}{Mu^2+(v-M)u} + \mathcal{O}(t^{-1}).$$

Hence,

$$K \sum_{v=1}^{M} \left[ \sum_{u=1}^{\lfloor \frac{t-1}{M} \rfloor} \frac{1}{v+M(u-1)} \left( \sum_{s \in S^{(v)}} \alpha_s \boldsymbol{x}_s \right) \right]$$
$$= \frac{K}{M} \left( \ln \frac{t}{M} + \gamma \right) \left( \sum_{s \in S} \alpha_s \boldsymbol{x}_s \right) + K \sum_{v=1}^{M} \sum_{u=1}^{\infty} \frac{1-\frac{v}{M}}{Mu^2+(v-M)u} \left( \sum_{s \in S^{(v)}} \alpha_s \boldsymbol{x}_s \right) + \boldsymbol{m}_t''$$
$$= \frac{K}{M} \left( \ln \frac{t}{M} + \gamma \right) \hat{\boldsymbol{w}} + K \sum_{v=1}^{M} \sum_{u=1}^{\infty} \frac{1-\frac{v}{M}}{Mu^2+(v-M)u} \left( \sum_{s \in S^{(v)}} \alpha_s \boldsymbol{x}_s \right) + \boldsymbol{m}_t''$$
$$= \frac{K}{M} \ln(\frac{t}{M}) \hat{\boldsymbol{w}} + \frac{K}{M} \check{\boldsymbol{w}} + \boldsymbol{m}_t'',$$

where $\check{\boldsymbol{w}} := \gamma \hat{\boldsymbol{w}} + M \sum_{v=1}^{M} \sum_{u=1}^{\infty} \frac{1-\frac{v}{M}}{Mu^2+(v-M)u} \left( \sum_{s \in S^{(v)}} \alpha_s \boldsymbol{x}_s \right)$, and $\|\boldsymbol{m}_t''\| = \mathcal{O}(t^{-1})$.

Finally, for all $k \in [0:K-1]$, let

$$\boldsymbol{m}_{t,k} := K \sum_{u=1}^{t-1} \frac{1}{u} \sum_{s \in S^{(u)}} \alpha_s \boldsymbol{x}_s + \frac{k}{t} \sum_{s \in S^{(t)}} \alpha_s \boldsymbol{x}_s - \frac{K}{M} \ln(\frac{t}{M}) \hat{\boldsymbol{w}} - \frac{K}{M} \check{\boldsymbol{w}}$$

and

$$\boldsymbol{m}_{t,K} := \boldsymbol{m}_{t+1,0}.$$

Then $\|\boldsymbol{m}_{t,k}\| = \left\| \boldsymbol{m}_{t,k}' + \boldsymbol{m}_t'' \right\| = \mathcal{O}(t^{-1})$, and

$$\|\boldsymbol{m}_{t,k+1} - \boldsymbol{m}_{t,k}\| = \left\| \frac{1}{t} \sum_{s \in S^{(t)}} \alpha_s \boldsymbol{x}_s \right\| = \mathcal{O}(t^{-1}), \quad (k = 0, ..., K-2)$$

$$\|\boldsymbol{m}_{t+1,0} - \boldsymbol{m}_{t,K-1}\| = \left\| \frac{1}{t} \sum_{s \in S^{(t)}} \alpha_s \boldsymbol{x}_s - \frac{K}{M} \ln(1+t^{-1}) \hat{\boldsymbol{w}} \right\| = \mathcal{O}(t^{-1}).$$

### E.2.2 PROOF OF LEMMA E.5

We use Assumption 3.4 here. That is, there exist positive constants $\mu_+, \mu_-,$ and $\bar{u}$ such that $\forall u > \bar{u}$:

$$(1 - \exp(-\mu_- u))e^{-u} \le -\ell'(u) \le (1 + \exp(-\mu_+ u))e^{-u}.$$

By definition,

$$\forall k \in [0:K-1]: \boldsymbol{r}_k^{(t)} = \boldsymbol{w}_k^{(t)} - \frac{M}{K} \left( K \sum_{u=1}^{t-1} \frac{1}{u} \sum_{s \in S^{(u)}} \alpha_s \boldsymbol{x}_s + \frac{k}{t} \sum_{s \in S^{(t)}} \alpha_s \boldsymbol{x}_s \right) - \ln K \hat{\boldsymbol{w}} - \tilde{\boldsymbol{w}} + \check{\boldsymbol{w}},$$

$$\boldsymbol{r}_K^{(t)} = \boldsymbol{r}_0^{(t+1)}.$$

Then for all $k \in [0 : K-1]$, we get

$$\boldsymbol{r}_{k+1}^{(t)} - \boldsymbol{r}_k^{(t)} = \boldsymbol{w}_{k+1}^{(t)} - \boldsymbol{w}_k^{(t)} - \frac{M}{Kt} \sum_{s \in S^{(t)}} \alpha_s \boldsymbol{x}_s$$

$$= -\eta \sum_{s \in I^{(t)}} \ell'(\boldsymbol{x}_s^\top \boldsymbol{w}_k^{(t)}) \boldsymbol{x}_s - \frac{M}{Kt} \sum_{s \in S^{(t)}} \alpha_s \boldsymbol{x}_s$$

$$= -\eta \sum_{s \in I^{(t)} \setminus S^{(t)}} \ell'(\boldsymbol{x}_s^\top \boldsymbol{w}_k^{(t)}) \boldsymbol{x}_s - \sum_{s \in S^{(t)}} \left[ \eta \ell'(\boldsymbol{x}_s^\top \boldsymbol{w}_k^{(t)}) + \frac{M}{Kt} \alpha_s \right] \boldsymbol{x}_s.$$

Hence,

$$\left( \boldsymbol{r}_{k+1}^{(t)} - \boldsymbol{r}_k^{(t)} \right)^\top \boldsymbol{r}_k^{(t)}$$

$$= -\eta \sum_{s \in I^{(t)} \setminus S^{(t)}} \ell'(\boldsymbol{x}_s^\top \boldsymbol{w}_k^{(t)}) \boldsymbol{x}_s^\top \boldsymbol{r}_k^{(t)} - \sum_{s \in S^{(t)}} \left[ \eta \ell'(\boldsymbol{x}_s^\top \boldsymbol{w}_k^{(t)}) + \frac{M}{Kt} \alpha_s \right] \boldsymbol{x}_s^\top \boldsymbol{r}_k^{(t)}$$

$$= -\eta \sum_{s \in I^{(t)} \setminus S^{(t)}} \ell' \left( \ln\left(\frac{K}{M} t\right) \boldsymbol{x}_s^\top \hat{\boldsymbol{w}} + \frac{M}{K} \boldsymbol{x}_s^\top \boldsymbol{m}_{t,k} + \boldsymbol{x}_s^\top \tilde{\boldsymbol{w}} + \boldsymbol{x}_s^\top \boldsymbol{r}_k^{(t)} \right) \boldsymbol{x}_s^\top \boldsymbol{r}_k^{(t)} \tag{25}$$

$$- \sum_{s \in S^{(t)}} \left[ \eta \ell' \left( \ln\left(\frac{K}{M} t\right) + \frac{M}{K} \boldsymbol{x}_s^\top \boldsymbol{m}_{t,k} + \boldsymbol{x}_s^\top \tilde{\boldsymbol{w}} + \boldsymbol{x}_s^\top \boldsymbol{r}_k^{(t)} \right) + \frac{M}{Kt} \alpha_s \right] \boldsymbol{x}_s^\top \boldsymbol{r}_k^{(t)}. \tag{26}$$

The behavior of each term can be analyzed when stage $t$ is large. To achieve this, we first characterize five stages.

$$t_5 := \min\{t' \mid \forall t \geq t', \forall k \in [0 : K-1], \forall s \in I : \boldsymbol{x}_s^\top \boldsymbol{w}_k^{(t)} \geq \bar{u}\}$$

$$t_6 := \min\{t' \mid \forall t \geq t', \forall k \in [0 : K-1], \forall s \in I : \boldsymbol{x}_s^\top \boldsymbol{w}_k^{(t)} \geq 0\}$$

$$t_7 := \min\{t' \mid \forall t \geq t', \forall k \in [0 : K-1], \forall s \in I : \exp\left(-\frac{M}{K} \boldsymbol{x}_s^\top \boldsymbol{m}_{t,k}\right) \leq 2\}$$

$$t_8 := \min\{t' \mid \forall t \geq t', \forall k \in [0 : K-1], \forall s \in I : \exp\left(-\frac{M}{K} \boldsymbol{x}_s^\top \boldsymbol{m}_{t,k}\right) \geq \frac{1}{2}\}$$

$$t_9 := \min\{t' \mid \forall t \geq t', \forall k \in [0 : K-1], \forall s \in I : \exp\left(-\mu_- \boldsymbol{x}_s^\top \boldsymbol{w}_k^{(t)}\right) \leq \frac{1}{2}\}$$

Such $t_5 \sim t_9$ exist since $\forall s \in I, \forall k \in [0 : K-1] : \lim_{t \to \infty} \boldsymbol{x}_s^\top \boldsymbol{w}_k^{(t)} = \infty$ by Theorem 3.1, and $\forall k \in [0 : K-1] : \lim_{t \to \infty} \|\boldsymbol{m}_{t,k}\| = 0$ by Lemma E.4.

Then for all $t \geq \max\{t_5, t_6, t_7, t_8, t_9\}$, the first term (25) can be upper bounded as below:

$$-\eta \sum_{s \in I^{(t)} \setminus S^{(t)}} \ell'(\boldsymbol{x}_s^\top \boldsymbol{w}_k^{(t)}) \boldsymbol{x}_s^\top \boldsymbol{r}_k^{(t)} \leq -\eta \sum_{\substack{s \in I^{(t)} \setminus S^{(t)} \\ \boldsymbol{x}_s^\top \boldsymbol{r}_k^{(t)} > 0}} \ell'(\boldsymbol{x}_s^\top \boldsymbol{w}_k^{(t)}) \boldsymbol{x}_s^\top \boldsymbol{r}_k^{(t)}$$

$$\leq \eta \sum_{\substack{s \in I^{(t)} \setminus S^{(t)} \\ \boldsymbol{x}_s^\top \boldsymbol{r}_k^{(t)} > 0}} \left( 1 + \exp(-\mu_+ \boldsymbol{x}_s^\top \boldsymbol{w}_k^{(t)}) \right) \exp(-\boldsymbol{x}_s^\top \boldsymbol{w}_k^{(t)}) \boldsymbol{x}_s^\top \boldsymbol{r}_k^{(t)} \qquad t \geq t_5$$

$$\leq \eta \sum_{\substack{s \in I^{(t)} \setminus S^{(t)} \\ \boldsymbol{x}_s^\top \boldsymbol{r}_k^{(t)} > 0}} 2 \exp\left( -\ln\left(\frac{K}{M} t\right) \boldsymbol{x}_s^\top \hat{\boldsymbol{w}} - \frac{M}{K} \boldsymbol{x}_s^\top \boldsymbol{m}_{t,k} - \boldsymbol{x}_s^\top \tilde{\boldsymbol{w}} - \boldsymbol{x}_s^\top \boldsymbol{r}_k^{(t)} \right) \boldsymbol{x}_s^\top \boldsymbol{r}_k^{(t)} \quad t \geq t_6$$

$$\leq \sum_{\substack{s \in I^{(t)} \setminus S^{(t)} \\ \boldsymbol{x}_s^\top \boldsymbol{r}_k^{(t)} > 0}} 2 \alpha_s \exp\left( -\ln\left(\frac{K}{M} t\right) \boldsymbol{x}_s^\top \hat{\boldsymbol{w}} - \frac{M}{K} \boldsymbol{x}_s^\top \boldsymbol{m}_{t,k} - \boldsymbol{x}_s^\top \boldsymbol{r}_k^{(t)} \right) \boldsymbol{x}_s^\top \boldsymbol{r}_k^{(t)} \tag{27}$$

$$\leq \sum_{\substack{s \in I^{(t)} \setminus S^{(t)} \\ \boldsymbol{x}_s^\top \boldsymbol{r}_k^{(t)} > 0}} 2\alpha_s \exp\left(-\ln\left(\frac{K}{M}t\right)\boldsymbol{x}_s^\top \hat{\boldsymbol{w}} - \frac{M}{K}\boldsymbol{x}_s^\top \boldsymbol{m}_{t,k}\right) \tag{28}$$

$$\leq \sum_{\substack{s \in I^{(t)} \setminus S^{(t)} \\ \boldsymbol{x}_s^\top \boldsymbol{r}_k^{(t)} > 0}} 4\alpha_s \exp\left(-\ln\left(\frac{K}{M}t\right)\boldsymbol{x}_s^\top \hat{\boldsymbol{w}}\right) \qquad\qquad t \geq t_7$$

$$\tag{29}$$

$$\leq 4N(\max_s \alpha_s)\left(\frac{Kt}{M}\right)^{-\theta}, \tag{30}$$

where in (27) we use the definition of $\tilde{\boldsymbol{w}}$, in (28) we use the fact $\forall x \geq 0 : x \exp(-x) \leq 1$, and in (30) we use $\forall s \in I^{(t)} \setminus S^{(t)} : x_s^\top \hat{\boldsymbol{w}} \geq \theta$. Now we examine the second term (26). Given $t \geq t_5$, it can be divided into two cases.

$$-\ell'(\boldsymbol{x}_s^\top \boldsymbol{w}_k^{(t)})\boldsymbol{x}_s^\top \boldsymbol{r}_k^{(t)} \leq \begin{cases} \left(1 + \exp(-\mu_+ \boldsymbol{x}_s^\top \boldsymbol{w}_k^{(t)})\right)\exp(-\boldsymbol{x}_s^\top \boldsymbol{w}_k^{(t)})\boldsymbol{x}_s^\top \boldsymbol{r}_k^{(t)} & \text{if } \boldsymbol{x}_s^\top \boldsymbol{r}_k^{(t)} > 0 \\ \left(1 - \exp(-\mu_- \boldsymbol{x}_s^\top \boldsymbol{w}_k^{(t)})\right)\exp(-\boldsymbol{x}_s^\top \boldsymbol{w}_k^{(t)})\boldsymbol{x}_s^\top \boldsymbol{r}_k^{(t)} & \text{if } \boldsymbol{x}_s^\top \boldsymbol{r}_k^{(t)} \leq 0 \end{cases}$$

For each $s \in S$, define $A_{s,k}^{(t)}$ as

$$A_{s,k}^{(t)} := \begin{cases} 1 + \exp(-\mu_+ \boldsymbol{x}_s^\top \boldsymbol{w}_k^{(t)}) & \text{if } \boldsymbol{x}_s^\top \boldsymbol{r}_k^{(t)} > 0 \\ 1 - \exp(-\mu_- \boldsymbol{x}_s^\top \boldsymbol{w}_k^{(t)}) & \text{if } \boldsymbol{x}_s^\top \boldsymbol{r}_k^{(t)} \leq 0 \end{cases}$$

Then, we can use

$$-\ell'(\boldsymbol{x}_s^\top \boldsymbol{w}_k^{(t)})\boldsymbol{x}_s^\top \boldsymbol{r}_k^{(t)} \leq A_{s,k}^{(t)} \exp(-\boldsymbol{x}_s^\top \boldsymbol{w}_k^{(t)})\boldsymbol{x}_s^\top \boldsymbol{r}_k^{(t)}$$

in any $s \in S, k \in [0 : K-1]$. Therefore the second term (26) is bounded

$$-\sum_{s \in S^{(t)}}\left[\eta\ell'\left(\ln\left(\frac{K}{M}t\right) + \frac{M}{K}\boldsymbol{x}_s^\top \boldsymbol{m}_{t,k} + \boldsymbol{x}_s^\top \tilde{\boldsymbol{w}} + \boldsymbol{x}_s^\top \boldsymbol{r}_k^{(t)}\right) + \frac{M}{Kt}\alpha_s\right]\boldsymbol{x}_s^\top \boldsymbol{r}_k^{(t)}$$

$$\leq \sum_{s \in S^{(t)}}\left[\eta A_{s,k}^{(t)}\exp\left(-\ln\left(\frac{K}{M}t\right) - \frac{M}{K}\boldsymbol{x}_s^\top \boldsymbol{m}_{t,k} - \boldsymbol{x}_s^\top \tilde{\boldsymbol{w}} - \boldsymbol{x}_s^\top \boldsymbol{r}_k^{(t)}\right) - \frac{M}{Kt}\alpha_s\right]\boldsymbol{x}_s^\top \boldsymbol{r}_k^{(t)}$$

$$= \sum_{s \in S^{(t)}}\left[A_{s,k}^{(t)}\frac{M\alpha_s}{Kt}\exp\left(-\frac{M}{K}\boldsymbol{x}_s^\top \boldsymbol{m}_{t,k} - \boldsymbol{x}_s^\top \boldsymbol{r}_k^{(t)}\right) - \frac{M}{Kt}\alpha_s\right]\boldsymbol{x}_s^\top \boldsymbol{r}_k^{(t)}$$

$$= \sum_{s \in S^{(t)}}\frac{M}{Kt}\alpha_s\left[A_{s,k}^{(t)}\exp\left(-\frac{M}{K}\boldsymbol{x}_s^\top \boldsymbol{m}_{t,k} - \boldsymbol{x}_s^\top \boldsymbol{r}_k^{(t)}\right) - 1\right]\boldsymbol{x}_s^\top \boldsymbol{r}_k^{(t)}.$$

Now we analyze each $s \in S^{(t)}$ by dividing into cases. Note that $\left|\frac{M}{K}\boldsymbol{x}_s^\top \boldsymbol{m}_{t,k}\right| = o(t^{-0.5+\epsilon})$ for all $\epsilon > 0$. Therefore if we set $\tilde{\mu} = \min\{\mu_+, \mu_-, 0.25\}$, then $\left|\frac{M}{K}\boldsymbol{x}_s^\top \boldsymbol{m}_{t,k}\right| = o(t^{-\tilde{\mu}})$.

1. if $0 \leq \boldsymbol{x}_s^\top \boldsymbol{r}_k^{(t)} \leq C_7 t^{-0.5\tilde{\mu}}$:

$$\frac{M}{Kt}\alpha_s\left[A_{s,k}^{(t)}\exp\left(-\frac{M}{K}\boldsymbol{x}_s^\top \boldsymbol{m}_{t,k} - \boldsymbol{x}_s^\top \boldsymbol{r}_k^{(t)}\right) - 1\right]\boldsymbol{x}_s^\top \boldsymbol{r}_k^{(t)}$$

$$\leq \frac{M}{Kt}\alpha_s\left[2\exp\left(-\frac{M}{K}\boldsymbol{x}_s^\top \boldsymbol{m}_{t,k} - \boldsymbol{x}_s^\top \boldsymbol{r}_k^{(t)}\right) - 1\right]\boldsymbol{x}_s^\top \boldsymbol{r}_k^{(t)} \qquad t \geq t_6$$

$$\leq \frac{M}{Kt}\alpha_s\left[4\exp\left(-\boldsymbol{x}_s^\top \boldsymbol{r}_k^{(t)}\right) - 1\right]\boldsymbol{x}_s^\top \boldsymbol{r}_k^{(t)} \qquad t \geq t_7$$

$$\leq \left(\max_s \alpha_s\right)\frac{4MC_7}{K}t^{-1-0.5\tilde{\mu}}.$$

The last inequality holds by the case condition $0 \leq \boldsymbol{x}_s^\top \boldsymbol{r}_k^{(t)} \leq C_7 t^{-0.5\tilde{\mu}}$.

2. if $-C_7 t^{-0.5\tilde{\mu}} \leq \boldsymbol{x}_s^\top \boldsymbol{r}_k^{(t)} \leq 0$:

$$\frac{M}{Kt}\alpha_s \left[ A_{s,k}^{(t)} \exp\left( -\frac{M}{K}\boldsymbol{x}_s^\top \boldsymbol{m}_{t,k} - \boldsymbol{x}_s^\top \boldsymbol{r}_k^{(t)} \right) - 1 \right] \boldsymbol{x}_s^\top \boldsymbol{r}_k^{(t)}$$

$$= \frac{M}{Kt}\alpha_s \left[ 1 - A_{s,k}^{(t)} \exp\left( -\frac{M}{K}\boldsymbol{x}_s^\top \boldsymbol{m}_{t,k} - \boldsymbol{x}_s^\top \boldsymbol{r}_k^{(t)} \right) \right] \left| \boldsymbol{x}_s^\top \boldsymbol{r}_k^{(t)} \right|$$

$$\leq \frac{M}{Kt}\alpha_s \left| \boldsymbol{x}_s^\top \boldsymbol{r}_k^{(t)} \right| \leq \frac{M}{Kt}\alpha_s \cdot C_7 t^{-0.5\tilde{\mu}}$$

$$\leq \left( \max_s \alpha_s \right) \frac{MC_7}{K} t^{-1-0.5\tilde{\mu}}.$$

3. if $C_7 t^{-0.5\tilde{\mu}} < \boldsymbol{x}_s^\top \boldsymbol{r}_k^{(t)}$:

Here, we first examine $A_{s,k}^{(t)}$.

$$A_{s,k}^{(t)} = 1 + \exp(-\mu_+ \boldsymbol{x}_s^\top \boldsymbol{w}_k^{(t)})$$

$$= 1 + \exp\left( -\mu_+ \left( \ln\left( \frac{K}{M}t \right) + \frac{M}{K}\boldsymbol{x}_s^\top \boldsymbol{m}_{t,k} + \boldsymbol{x}_s^\top \tilde{\boldsymbol{w}} + \boldsymbol{x}_s^\top \boldsymbol{r}_k^{(t)} \right) \right)$$

$$\leq 1 + \exp\left( -\mu_+ \left( \ln\left( \frac{K}{M}t \right) + \frac{M}{K}\boldsymbol{x}_s^\top \boldsymbol{m}_{t,k} + \boldsymbol{x}_s^\top \tilde{\boldsymbol{w}} \right) \right)$$

$$\leq 1 + 2^{\mu_+} \exp\left( -\mu_+ \left( \ln\left( \frac{K}{M}t \right) + \boldsymbol{x}_s^\top \tilde{\boldsymbol{w}} \right) \right) \qquad\qquad t \geq t_7$$

$$\leq 1 + C_8 t^{-\mu_+}.$$

Therefore,

$$\frac{M}{Kt}\alpha_s \left[ A_{s,k}^{(t)} \exp\left( -\frac{M}{K}\boldsymbol{x}_s^\top \boldsymbol{m}_{t,k} - \boldsymbol{x}_s^\top \boldsymbol{r}_k^{(t)} \right) - 1 \right] \boldsymbol{x}_s^\top \boldsymbol{r}_k^{(t)}$$

$$\leq \frac{M}{Kt}\alpha_s \left[ (1 + C_8 t^{-\mu_+}) \exp\left( -\frac{M}{K}\boldsymbol{x}_s^\top \boldsymbol{m}_{t,k} - \boldsymbol{x}_s^\top \boldsymbol{r}_k^{(t)} \right) - 1 \right] \boldsymbol{x}_s^\top \boldsymbol{r}_k^{(t)}$$

$$\leq \frac{M}{Kt}\alpha_s \left[ (1 + C_8 t^{-\mu_+}) \exp\left( -\frac{M}{K}\boldsymbol{x}_s^\top \boldsymbol{m}_{t,k} - C_7 t^{-0.5\tilde{\mu}} \right) - 1 \right] \boldsymbol{x}_s^\top \boldsymbol{r}_k^{(t)}. \qquad (31)$$

Since $t \geq t_7$, $-\frac{M}{K}\boldsymbol{x}_s^\top \boldsymbol{m}_{t,k} \leq 1$. Now by the fact $\forall x \leq 1 : \exp x \leq 1 + x + x^2$,

$$\exp\left( -\frac{M}{K}\boldsymbol{x}_s^\top \boldsymbol{m}_{t,k} \right) \leq 1 - \frac{M}{K}\boldsymbol{x}_s^\top \boldsymbol{m}_{t,k} + \left( \frac{M}{K}\boldsymbol{x}_s^\top \boldsymbol{m}_{t,k} \right)^2,$$

$$\exp\left( -C_7 t^{-0.5\tilde{\mu}} \right) \leq 1 - C_7 t^{-0.5\tilde{\mu}} + C_7^2 t^{-\tilde{\mu}}.$$

Then we get

$$\left( 1 + C_8 t^{-\mu_+} \right) \exp\left( -\frac{M}{K}\boldsymbol{x}_s^\top \boldsymbol{m}_{t,k} - C_7 t^{-0.5\tilde{\mu}} \right)$$

$$\leq \left( 1 - \frac{M}{K}\boldsymbol{x}_s^\top \boldsymbol{m}_{t,k} + \left( \frac{M}{K}\boldsymbol{x}_s^\top \boldsymbol{m}_{t,k} \right)^2 \right) \left( 1 - C_7 t^{-0.5\tilde{\mu}} \right) + o(t^{-\mu_+})$$

$$\leq 1 - \frac{M}{K}\boldsymbol{x}_s^\top \boldsymbol{m}_{t,k} + \left( \frac{M}{K}\boldsymbol{x}_s^\top \boldsymbol{m}_{t,k} \right)^2 - C_7 t^{-0.5\tilde{\mu}} + o(t^{-\mu_+})$$

$$\leq 1 - C_7 t^{-0.5\tilde{\mu}} + o(t^{-\tilde{\mu}})$$

where in the last two inequality, we use $\left| \frac{M}{K}\boldsymbol{x}_s^\top \boldsymbol{m}_{t,k} \right| = o(t^{-\tilde{\mu}})$.

Finally, Equation (31) is bounded

$$\frac{M}{Kt}\alpha_s \left[ (1 + C_8 t^{-\mu_+}) \exp\left( -\frac{M}{K}\boldsymbol{x}_s^\top \boldsymbol{m}_{t,k} - C_7 t^{-0.5\tilde{\mu}} \right) - 1 \right] \boldsymbol{x}_s^\top \boldsymbol{r}_k^{(t)}$$

$$\leq \frac{M}{Kt}\alpha_s \left[-C_7 t^{-0.5\tilde{\mu}} + o(t^{-\tilde{\mu}})\right] \boldsymbol{x}_s^\top \boldsymbol{r}_k^{(t)}.$$

Since $-C_7 t^{-0.5\tilde{\mu}}$ decrease to zero slower than the other term, $\exists t_+ \geq \max\{t_5, t_6, t_7, t_8, t_9\}$ such that for all $t \geq t_+$, the last term is negative.

4. if $\boldsymbol{x}_s^\top \boldsymbol{r}_k^{(t)} < -C_7 t^{-0.5\tilde{\mu}}$:

Since $\boldsymbol{x}_s^\top \boldsymbol{r}_k^{(t)} < 0$, it is enough to show that $A_{s,k}^{(t)} \exp\left(-\frac{M}{K}\boldsymbol{x}_s^\top \boldsymbol{m}_{t,k} - \boldsymbol{x}_s^\top \boldsymbol{r}_k^{(t)}\right) > 1$ for sufficiently large $t$. Note that $A_{s,k}^{(t)} = 1 - \exp(-\mu_- \boldsymbol{x}_s^\top \boldsymbol{w}_k^{(t)}) > 0$ since $t \geq t_6$. If $\exp\left(-\boldsymbol{x}_s^\top \boldsymbol{r}_k^{(t)}\right) \geq 4$,

$$A_{s,k}^{(t)} \exp\left(-\frac{M}{K}\boldsymbol{x}_s^\top \boldsymbol{m}_{t,k} - \boldsymbol{x}_s^\top \boldsymbol{r}_k^{(t)}\right)$$
$$\geq 4(1 - \exp(-\mu_- \boldsymbol{x}_s^\top \boldsymbol{w}_k^{(t)})) \exp\left(-\frac{M}{K}\boldsymbol{x}_s^\top \boldsymbol{m}_{t,k}\right) \geq 1.$$

The last inequality holds by $t \geq \max\{t_8, t_9\}$. Now, if $\exp\left(-\boldsymbol{x}_s^\top \boldsymbol{r}_k^{(t)}\right) < 4$,

$$A_{s,k}^{(t)} = 1 - \exp\left(-\mu_-\left(\ln\left(\frac{K}{M}t\right) + \frac{M}{K}\boldsymbol{x}_s^\top \boldsymbol{m}_{t,k} + \boldsymbol{x}_s^\top \tilde{\boldsymbol{w}} + \boldsymbol{x}_s^\top \boldsymbol{r}_k^{(t)}\right)\right)$$
$$\geq 1 - \left(\frac{4Kt}{M}\right)^{-\mu_-} \exp\left(-\mu_-\left(\frac{M}{K}\boldsymbol{x}_s^\top \boldsymbol{m}_{t,k} + \boldsymbol{x}_s^\top \tilde{\boldsymbol{w}}\right)\right)$$
$$\geq 1 - \left(\frac{8Kt}{M}\right)^{-\mu_-} \exp\left(-\mu_- \boldsymbol{x}_s^\top \tilde{\boldsymbol{w}}\right) \geq 1 - C_9 t^{-\mu_-}. \qquad t \geq t_7$$

Also, by the fact $\forall x : \exp x \geq 1 + x$,

$$\exp\left(-\frac{M}{K}\boldsymbol{x}_s^\top \boldsymbol{m}_{t,k} - \boldsymbol{x}_s^\top \boldsymbol{r}_k^{(t)}\right) \geq \left(1 - \frac{M}{K}\boldsymbol{x}_s^\top \boldsymbol{m}_{t,k}\right)\left(1 - \boldsymbol{x}_s^\top \boldsymbol{r}_k^{(t)}\right).$$

Combined with the former inequality,

$$A_{s,k}^{(t)} \exp\left(-\frac{M}{K}\boldsymbol{x}_s^\top \boldsymbol{m}_{t,k} - \boldsymbol{x}_s^\top \boldsymbol{r}_k^{(t)}\right)$$
$$\geq \left(1 - C_9 t^{-\mu_-}\right)\left(1 - \frac{M}{K}\boldsymbol{x}_s^\top \boldsymbol{m}_{t,k}\right)\left(1 - \boldsymbol{x}_s^\top \boldsymbol{r}_k^{(t)}\right)$$
$$\geq \left(1 - C_9 t^{-\mu_-}\right)\left(1 + o(t^{-\tilde{\mu}})\right)\left(1 + C_7 t^{-0.5\tilde{\mu}}\right)$$
$$= 1 + C_7 t^{-0.5\tilde{\mu}} - o(t^{-\tilde{\mu}}).$$

Since $C_7 t^{-0.5\tilde{\mu}}$ decrease to zero slower than the other term, $\exists t_- \geq \max\{t_5, t_6, t_7, t_8, t_9\}$ such that for all $t \geq t_-$, the last equation is larger than 1.

To sum up, there exist $C_1, C_2 > 0, \tilde{t} \geq \max\{t_+, t_-\}$ such that for all $t \geq \tilde{t}$,

$$(\boldsymbol{r}_{k+1}^{(t)} - \boldsymbol{r}_k^{(t)})^\top \boldsymbol{r}_k^{(t)} \leq C_1 t^{-\theta} + C_2 t^{-1-0.5\tilde{\mu}}, \forall k \in [0 : K-1].$$

Now we consider special cases to finish the lemma. For any $\epsilon_2 > 0$, the following analysis holds.

1. If $\boldsymbol{x}_s^\top \boldsymbol{r}_k^{(t)} \geq \epsilon_2 > 0$:

Since $\lim_{t\to\infty} \boldsymbol{m}_{t,k} = 0$, there exist $t_1^* \geq \max\{t_+, t_-\}$ such that $\forall t \geq t_1^*, \forall s \in S, \forall k \in [0 : K-1] : |\frac{M}{K}\boldsymbol{x}_s^\top \boldsymbol{m}_{t,k}| < 0.5\epsilon_2$. Also since $\lim_{t\to\infty} \boldsymbol{x}_s^\top \boldsymbol{w}_k^{(t)} \to \infty$, there exist $t_+^* \geq t_1^*$ such that $\forall t \geq t_+^*, \forall s \in S, \forall k \in [0 : K-1] : \exp\left(-\mu_+ \boldsymbol{x}_s^\top \boldsymbol{w}_k^{(t)}\right) \leq \exp(0.25\epsilon_2) - 1$. Therefore for $t \geq t_+^*$,

$$\frac{M}{Kt}\alpha_s \left[A_{s,k}^{(t)} \exp\left(-\frac{M}{K}\boldsymbol{x}_s^\top \boldsymbol{m}_{t,k} - \boldsymbol{x}_s^\top \boldsymbol{r}_k^{(t)}\right) - 1\right]\boldsymbol{x}_s^\top \boldsymbol{r}_k^{(t)}$$

$$\leq \frac{M}{Kt}\alpha_s \left[\left(1 + \exp(-\mu_+ \boldsymbol{x}_s^\top \boldsymbol{w}_k^{(t)})\right)\exp(-0.5\epsilon_2) - 1\right]\boldsymbol{x}_s^\top \boldsymbol{r}_k^{(t)} \qquad t \geq t_1^*$$

$$\leq \frac{M}{Kt}\alpha_s \left(\exp(-0.25\epsilon_2) - 1\right)\boldsymbol{x}_s^\top \boldsymbol{r}_k^{(t)} \qquad t \geq t_+^*$$

$$\leq \min_s \alpha_s \frac{M}{K}\left(\exp(-0.25\epsilon_2) - 1\right)\epsilon_2 \frac{1}{t} = -C_+'' t^{-1}.$$

2. If $\boldsymbol{x}_s^\top \boldsymbol{r}_k^{(t)} \leq -\epsilon_2 < 0$:

Again, since $\lim_{t\to\infty} \boldsymbol{x}_s^\top \boldsymbol{w}_k^{(t)} \to \infty$, there exist $t_-^* \geq t_1^*$ such that $\forall t \geq t_-^*, \forall s \in S, \forall k \in [0 : K-1] : 1 - \exp\left(-\mu_- \boldsymbol{x}_s^\top \boldsymbol{w}_k^{(t)}\right) \geq \exp(-0.25\epsilon_2)$. Therefore for $t \geq t_-^*$,

$$\frac{M}{Kt}\alpha_s \left[A_{s,k}^{(t)}\exp\left(-\frac{M}{K}\boldsymbol{x}_s^\top \boldsymbol{m}_{t,k} - \boldsymbol{x}_s^\top \boldsymbol{r}_k^{(t)}\right) - 1\right]\boldsymbol{x}_s^\top \boldsymbol{r}_k^{(t)}$$

$$\leq \frac{M}{Kt}\alpha_s \left[\left(1 - \exp(-\mu_- \boldsymbol{x}_s^\top \boldsymbol{w}_k^{(t)})\right)\exp(0.5\epsilon_2) - 1\right]\boldsymbol{x}_s^\top \boldsymbol{r}_k^{(t)} \qquad t \geq t_1^*$$

$$\leq \frac{M}{Kt}\alpha_s \left(\exp(0.25\epsilon_2) - 1\right)\boldsymbol{x}_s^\top \boldsymbol{r}_k^{(t)} \qquad t \geq t_-^*$$

$$\leq -\min_s \alpha_s \frac{M}{K}\left(\exp(0.25\epsilon_2) - 1\right)\epsilon_2 \frac{1}{t} = -C_-'' t^{-1}.$$

In conclusion, for any $\epsilon_1 > 0$, if $\left\|P\boldsymbol{r}_k^{(t)}\right\| \geq \epsilon_1$ and $S^{(t)} \neq \emptyset$, then

$$\max_{s \in S^{(t)}}\left|\boldsymbol{x}_s^\top \boldsymbol{r}_k^{(t)}\right|^2 = \max_{s \in S^{(t)}}\left|\left(P^\top \boldsymbol{x}_s\right)^\top \boldsymbol{r}_k^{(t)}\right|^2 \geq \frac{1}{\left|S^{(t)}\right|}\sum_{s \in S^{(t)}}\left|\boldsymbol{x}_s^\top P\boldsymbol{r}_k^{(t)}\right|^2$$

$$= \frac{1}{\left|S^{(t)}\right|}\left\|X_{S^{(t)}}^\top P\boldsymbol{r}_k^{(t)}\right\|^2 \geq \frac{1}{\left|S^{(t)}\right|}\sigma_{\min}^2(X_{S^{(t)}})\left\|P\boldsymbol{r}_k^{(t)}\right\|^2 \geq \frac{1}{\left|S^{(t)}\right|}\sigma_{\min}^2(X_{S^{(t)}})\epsilon_1^2$$

where $X_{S^{(t)}} \in \mathbb{R}^{d \times |S^{(t)}|}$ is a matrix which has $\{x_s \mid s \in S^{(t)}\}$ as its columns. By Assumption 3.2, $\sigma_{\min}(X_{S^{(t)}})$ is non-zero. Therefore, for all $\epsilon_1 > 0, \exists \tilde{t}^*, C_3 > 0$ such that if $\left\|P\boldsymbol{r}_k^{(t)}\right\| \geq \epsilon_1$ and $S^{(t)} \neq \emptyset$,

$$(\boldsymbol{r}_{k+1}^{(t)} - \boldsymbol{r}_k^{(t)})^\top \boldsymbol{r}_k^{(t)} \leq -C_3 t^{-1}, \quad \forall t > \tilde{t}^*, \ k \in [0 : K-1].$$

### E.2.3 Convergence of $\boldsymbol{\rho}_k^{(t)}$

Theorem 3.2 only shows boundedness of $\boldsymbol{\rho}_k^{(t)}$. Yet, if an additional mild assumption on data is given, it can be guaranteed that $\boldsymbol{\rho}_k^{(t)}$ converges to the particular vector.

**Assumption E.1.** Support vectors span dataset. That is, $\mathrm{rank}\{\boldsymbol{x}_i : i \in S\} = \mathrm{rank}\{\boldsymbol{x}_i : i \in I\}$.

**Proposition E.6.** *Under the same setting as Theorem 3.2 with an additional Assumption E.1, the "residual" converges to $\lim_{t\to\infty}\boldsymbol{\rho}_k^{(t)} = \tilde{\boldsymbol{w}}, \forall k \in [0 : K-1]$. Here, $\tilde{\boldsymbol{w}}$ is the unique solution of the following system of equations*

$$\forall i \in S : \eta \exp\left(-\boldsymbol{x}_i^\top \tilde{\boldsymbol{w}}\right) = \alpha_i, \quad (I - P)(\tilde{\boldsymbol{w}} - \boldsymbol{w}_0^{(0)}) = 0,$$

*where $P \in \mathbb{R}^{d \times d}$ is the orthogonal projection matrix to the space spanned by the joint support vectors indexed by $S$.*

We set $\bar{P} = I - P$ for the convenience of proof.

*Proof.* By the definition of $\boldsymbol{\rho}_k^{(t)} = \tilde{\boldsymbol{w}} + \frac{M}{K}\boldsymbol{m}_{t,k} + \boldsymbol{r}_k^{(t)}$, it is enough to prove $\lim_{t\to\infty}\boldsymbol{r}_k^{(t)} = 0$.

First of all, since $\boldsymbol{w}_k^{(t)} = \ln\left(\frac{K}{M}t\right)\hat{\boldsymbol{w}} + \tilde{\boldsymbol{w}} + \frac{M}{K}\boldsymbol{m}_{t,k} + \boldsymbol{r}_k^{(t)}$,

$$\bar{P}\boldsymbol{r}_k^{(t)} = \bar{P}\boldsymbol{w}_k^{(t)} - \ln\left(\frac{K}{M}t\right)\bar{P}\hat{\boldsymbol{w}} - \bar{P}\tilde{\boldsymbol{w}} - \frac{M}{K}\bar{P}\boldsymbol{m}_{t,k}$$

$$= \bar{P}\boldsymbol{w}_0^{(0)} - \ln\left(\frac{K}{M}t\right)\bar{P}\hat{\boldsymbol{w}} - \bar{P}\tilde{\boldsymbol{w}} - \frac{M}{K}\bar{P}\boldsymbol{m}_{t,k}$$

$$= \bar{P}\boldsymbol{w}_0^{(0)} - \bar{P}\tilde{\boldsymbol{w}} = 0.$$

The first line holds under the Assumption E.1 since $\nabla\mathcal{L}(w)$ is a linear combination of the columns of $X$. that is, $\forall l < t : \bar{P}\nabla\mathcal{L}^{(l)}(w) = 0$. The remaining lines are true by the definition.

Second, we get to show $Pr_k^{(t)} \to 0$. By Equation (24), $\lim_{T\to\infty}\sum_{t=t_1}^{T}\sum_{k=0}^{K-1}\left\|r_{k+1}^{(t)} - r_k^{(t)}\right\|^2 = C_4$. That means $\forall k \in [0 : K-1] : \lim_{T\to\infty}\left\|r_{k+1}^{(T)} - r_k^{(T)}\right\| = 0$. Therefore, for any $\epsilon_0$, there exists $t_2 > 0$ such that $\left\|r_{k+1}^{(t)} - r_k^{(t)}\right\| < \frac{\epsilon_0}{K}$ for all $t \geq t_2, k \in [0 : K-1]$. As a result,

$$\left\|Pr_0^{(t)}\right\| + \frac{k}{K}\epsilon_0 \geq \left\|Pr_k^{(t)}\right\| \geq \left\|Pr_0^{(t)}\right\| - \frac{k}{K}\epsilon_0.$$

For $t \geq \max\{t_1, t_2, \tilde{t}^*\}$, if $\left\|Pr_0^{(t)}\right\| \geq \epsilon_1 + \epsilon_0$ and $S^{(t)} \neq \emptyset$, then $\forall k \in [0 : K-1] : \left\|Pr_k^{(t)}\right\| \geq \epsilon_1$. By Lemma E.5 (2),

$$\forall m \in [0 : M-1] : \sum_{u=t}^{t+m}\sum_{v=0}^{K-1}(r_{v+1}^{(u)} - r_v^{(u)})^\top r_v^{(u)} \leq -KC_3 t^{-1} + Km\left(C_1 t^{-\theta} + C_2 t^{-1-0.5\tilde{\mu}}\right).$$

Since $t^{-1}$ decrease to zero slower than $t^{-\theta}$ and $t^{-1-0.5\tilde{\mu}}$, there exists $t_3 > \max\{t_1, t_2, \tilde{t}^*\}, C_4 > 0$ such that $-KC_3 t^{-1} + Km\left(C_1 t^{-\theta} + C_2 t^{-1-0.5\tilde{\mu}}\right) \leq -C_5 t^{-1}$. To sum up, for any $\epsilon_0, \epsilon_2 > 0$, there exists $t_3 > \max\{t_1, t_2, \tilde{t}^*\}$ such that if $\left\|Pr_0^{(t)}\right\| \geq \epsilon_0 + \epsilon_1$ and $S^{((t))} \neq \emptyset$, then

$$\forall m \in [0 : M-1] : \sum_{u=t}^{t+m}\sum_{v=0}^{K-1}(r_{v+1}^{(u)} - r_v^{(u)})^\top r_v^{(u)} \leq -C_5 t^{-1},$$

Now, define two sets for each $k \in [0 : K-1]$

$$\mathcal{T}_k := \{t > t_3 : \left\|Pr_k^{(t)}\right\| < \epsilon_0 + \epsilon_1\}$$

$$\bar{\mathcal{T}}_k := \{t > t_3 : \left\|Pr_k^{(t)}\right\| \geq \epsilon_0 + \epsilon_1\}$$

We will finish our proof by showing $\bar{\mathcal{T}}_k$ is finite.

First, every $\mathcal{T}_k$ is neither empty nor finite. If there exists some $k'$ that $\mathcal{T}_{k'}$ is empty or finite, then $\exists t_{\max} \in \bar{\mathcal{T}}_{k'}$. Then

$$\left\|Pr_0^{(t)}\right\|^2 - \left\|Pr_0^{(t_{\max})}\right\|^2 = \left\|r_0^{(t)}\right\|^2 - \left\|r_0^{(t_{\max})}\right\|^2$$

$$= \sum_{u=t_{\max}}^{t-1}\sum_{k=0}^{K-1}\left[\left\|r_{k+1}^{(u)}\right\|^2 - \left\|r_k^{(u)}\right\|^2\right]$$

$$= \sum_{u=t_{\max}}^{t-1}\sum_{k=0}^{K-1}\left[\left\|r_{k+1}^{(u)} - r_k^{(u)}\right\|^2\right] + 2\sum_{u=t_{\max}}^{t-1}\sum_{k=0}^{K-1}(r_{k+1}^{(u)} - r_k^{(u)})^\top r_k^{(u)}$$

$$\leq C_4 + 2\sum_{u=t_{\max}}^{t-1}\left(\sum_{k\neq k'}(r_{k+1}^{(u)} - r_k^{(u)})^\top r_k^{(u)} + (r_{k'+1}^{(u)} - r_{k'}^{(u)})^\top r_{k'}^{(u)}\right)$$

$$\leq C_4 + C_6 + 2\sum_{\substack{t_{\max}\leq u\leq t-1 \\ S^{(u)}\neq\emptyset}}(r_{k'+1}^{(u)} - r_{k'}^{(u)})^\top r_{k'}^{(u)}$$

$$\leq C_4 + C_6 - 2C_3 \sum_{\substack{t_{\max} \leq u \leq t-1 \\ S^{(u)} \neq \emptyset}} u^{-1}.$$

The first inequality is true because of Equation (24), while the other inequalities hold due to Lemma E.5. As $t$ goes to infinity, the upper bound goes to negative infinity. However, it contradicts the fact that $\left\| \boldsymbol{r}_0^{(t)} \right\|$ is bounded.

Before we move on to the final step, note that $\lim_{T \to \infty} \sum_{t=t_1}^{T} \sum_{k=0}^{K-1} \left\| \boldsymbol{r}_{k+1}^{(t)} - \boldsymbol{r}_k^{(t)} \right\|^2 = C_4$ implies

$$\sum_{u=t_1}^{t} \sum_{k=0}^{K-1} \left\| \boldsymbol{r}_{k+1}^{(u)} - \boldsymbol{r}_k^{(u)} \right\|^2 = C_4 - h(t)$$

where $h(t)$ is a positive function monotonic decreasing to zero.

Now, assume that there exists some $k'$ that $\bar{\mathcal{T}}_k$ is infinite. WLOG, we set $k' = 0$. Since $\mathcal{T}_0$ is infinite, for any $t \in \bar{\mathcal{T}}_0$ there exists $t', t'' \in \mathcal{T}_0$ such that $t \in [t'+1 : t''-1] \subset \bar{\mathcal{T}}_0$. We divide it into two cases: For all $t \in [t'+1 : t''-1]$,

1. if $|[t'+1 : t]| < M$, then $\left\| P\boldsymbol{r}_0^{(t)} \right\|^2 \leq \left\| P\boldsymbol{r}_0^{(t')} \right\|^2 + M\epsilon_0 \leq (M+1)\epsilon_0 + \epsilon_1$.

2. if $|[t'+1 : t]| \geq M$, let $t^* = \min\{u \in [t'+1 : t] : S^{(u)} \neq \emptyset\}$. Then

$$\left\| P\boldsymbol{r}_0^{(t)} \right\|^2 = \left\| P\boldsymbol{r}_0^{(t^*)} \right\|^2 + \sum_{u=t^*}^{t-1} \sum_{k=0}^{K-1} \left[ \left\| \boldsymbol{r}_{k+1}^{(u)} \right\|^2 - \left\| \boldsymbol{r}_k^{(u)} \right\|^2 \right]$$

$$= \left\| P\boldsymbol{r}_0^{(t^*)} \right\|^2 + \sum_{u=t^*}^{t-1} \sum_{k=0}^{K-1} \left[ \left\| \boldsymbol{r}_{k+1}^{(u)} - \boldsymbol{r}_k^{(u)} \right\|^2 + 2(\boldsymbol{r}_{k+1}^{(u)} - \boldsymbol{r}_k^{(u)})^\top \boldsymbol{r}_k^{(u)} \right]$$

$$= \left\| P\boldsymbol{r}_0^{(t^*)} \right\|^2 + h(t) - h(t^*) + 2 \sum_{u=t^*}^{t-1} \sum_{k=0}^{K-1} \left[ (\boldsymbol{r}_{k+1}^{(u)} - \boldsymbol{r}_k^{(u)})^\top \boldsymbol{r}_k^{(u)} \right]$$

$$\leq (M\epsilon_0 + \epsilon_0 + \epsilon_1)^2 + h(t) - 2C_5 \sum_{u=0}^{\lfloor \frac{t-1-t^*}{M} \rfloor} \frac{1}{Mu + t^*}$$

$$\leq (M\epsilon_0 + \epsilon_0 + \epsilon_1)^2 + h(t).$$

Since $h(t)$ is monotonic decreasing function, for any $\epsilon_2 > 0$, there exists $t_4$ such that $\forall t \geq t_4 : h(t) < \epsilon_2$.

Therefore, $\forall t \geq \max\{t_3, t_4\} : \left\| P\boldsymbol{r}_0^{(t)} \right\|^2 \leq (M\epsilon_0 + \epsilon_0 + \epsilon_1)^2 + \epsilon_2$. Since it holds for any $\epsilon_0, \epsilon_1, \epsilon_2$, it contradicts with the assumption that $\bar{\mathcal{T}}_0$ is infinite. $\qquad\square$

### E.3 ASYMPTOTIC LOSS CONVERGENCE RATE AFTER MANY CYCLES

Here we discuss and prove an asymptotic convergence rate of sequential GD, in terms of the joint training loss, after a large enough number of cycles $J$. This convergence rate is derived from the implicit bias result in Theorem 3.2, so it relies on a different set of assumptions from those used in the proof of Theorem 3.3. Namely, in this section, we rely on the the shape of the loss function $\ell(\cdot)$ described in Assumptions 3.3 and 3.4, thereby having $\mathcal{O}(1/J)$ rate without any logarithmic factor in $J$ in it, while the convexity of $\ell$ is *not* a necessity. On the other hand, in Theorem 3.3 having $\mathcal{O}(\ln^2(J)/J)$ convergence rate, the loss function $\ell$ is (non-strongly) convex (Assumption 3.5) and has a shape described in Assumption 3.3, but it does *not* have to satisfy the tight exponential tail assumption (Assumption 3.4).

We start the proof of the asymptotic rate by obtaining an upper bound of $\ell(u)$ based on Assumptions 3.3 and 3.4. Recall that, for any $v > \bar{u}$,

$$-\ell'(v) \leq (1 + \exp(-\mu_+ v))e^{-v}.$$

Taking the integration over $v \in [u, \infty)$ for $u > \bar{u}$, we have

$$
\begin{aligned}
\ell(u) &\leq \left(1 + \frac{1}{1 + \mu_+} \exp(-\mu_+ u)\right) e^{-u} \\
&\leq C_{\mu_+} e^{-u}, \qquad \left(\text{where } C_{\mu_+} = 1 + \frac{1}{1+\mu_+} e^{-\mu_+ \bar{u}} < 2\right)
\end{aligned}
\tag{32}
$$

since $\ell(u) \to 0$ as $u \to \infty$.

We move our attention to Equation (23), appeared in the proof of Theorem 3.2:

$$
\begin{aligned}
\boldsymbol{w}_k^{(t)} &= \ln\left(\frac{Kt}{M}\right)\hat{\boldsymbol{w}} + \underbrace{\tilde{\boldsymbol{w}} + \frac{M}{K}\boldsymbol{m}_{t,k} + \boldsymbol{r}_k^{(t)}}_{=:\widehat{\boldsymbol{\rho}}_k^{(t)}} \\
&= \ln\left(\frac{Kt}{M}\right)\hat{\boldsymbol{w}} + \widehat{\boldsymbol{\rho}}_k^{(t)}.
\end{aligned}
\tag{33}
$$

Recall that Theorem 3.2 proves the equation above for all sufficiently large $t$, as well as the boundedness of $\widehat{\boldsymbol{\rho}}_k^{(t)}$ as $t$ tends to infinity.

In addition, from Theorem 3.1.2, we can deduce that the inner products $\boldsymbol{x}_i^\top \boldsymbol{w}_k^{(t)} > \bar{u}$ for all sufficiently large $t$. Now, let $T_\star \in \left[\frac{M}{K}, \infty\right)$ be a time step such that $\boldsymbol{x}_i^\top \boldsymbol{w}_k^{(t)} > \bar{u}$ and Equation (33) hold for all $t \geq T_\star$. By definition of the joint training loss, for all $t \geq T_\star$,

$$
\begin{aligned}
\mathcal{L}\left(\boldsymbol{w}_k^{(t)}\right) &= \sum_{i \in I} \ell\left(\boldsymbol{x}_i^\top \boldsymbol{w}_k^{(t)}\right) \\
&\overset{(32)}{\leq} C_{\mu_+} \sum_{i \in I} \exp\left(-\boldsymbol{x}_i^\top \boldsymbol{w}_k^{(t)}\right) \\
&\overset{(33)}{=} C_{\mu_+} \sum_{i \in I} \exp\left(-\boldsymbol{x}_i^\top \boldsymbol{\rho}_k^{(t)}\right) \exp\left(-\ln\left(\frac{Kt}{M}\right)\boldsymbol{x}_i^\top \hat{\boldsymbol{w}}\right) \\
&= C_{\mu_+} \sum_{i \in I} \exp\left(-\boldsymbol{x}_i^\top \boldsymbol{\rho}_k^{(t)}\right)\left(\frac{M}{Kt}\right)^{\boldsymbol{x}_i^\top \hat{\boldsymbol{w}}} \\
&= C_{\mu_+} \sum_{i \in S} \exp\left(-\boldsymbol{x}_i^\top \widehat{\boldsymbol{\rho}}_k^{(t)}\right)\left(\frac{M}{Kt}\right)^{\boldsymbol{x}_i^\top \hat{\boldsymbol{w}}} + C_{\mu_+} \sum_{i \in I \setminus S} \exp\left(-\boldsymbol{x}_i^\top \widehat{\boldsymbol{\rho}}_k^{(t)}\right)\left(\frac{M}{Kt}\right)^{\boldsymbol{x}_i^\top \hat{\boldsymbol{w}}} \\
&\leq C_{\mu_+} \underbrace{\left(\sum_{i \in S} \exp\left(-\boldsymbol{x}_i^\top \widehat{\boldsymbol{\rho}}_k^{(t)}\right)\right) \cdot \frac{M}{Kt}}_{\text{This is the leading term in } t.} + C_{\mu_+} \left(\sum_{i \in I \setminus S} \exp\left(-\boldsymbol{x}_i^\top \widehat{\boldsymbol{\rho}}_k^{(t)}\right)\right) \cdot \left(\frac{M}{Kt}\right)^\theta.
\end{aligned}
$$

The last inequality above holds because $\boldsymbol{x}_i^\top \hat{\boldsymbol{w}} = 1$ if $\boldsymbol{x}_i$ is a support vector (i.e., $i \in S$) but $\boldsymbol{x}_i^\top \hat{\boldsymbol{w}} \geq \theta > 1$ otherwise. Since the magnitude of $\boldsymbol{\rho}_k^{(t)}$ is bounded above by a constant independent to $t$, we can summarize the bound above as $\mathcal{L}(\boldsymbol{w}_k^{(t)}) = \mathcal{O}(1/t)$; if we plug in $t = MJ$ where $J$ is the number of cycles, we yield $\mathcal{L}(\boldsymbol{w}_0^{(MJ)}) = \mathcal{O}(1/J)$.

A noteworthy implication of the asymptotic loss convergence bound above is about the vanishing rate of cycle-averaged forgetting (Definition 3.8) under the same setting of cyclic task occurrence. That is, we can obtain an asymptotic vanishing rate of the (absolute value of) cycle-averaged forgetting in terms of the cycle count $J$ as $\mathcal{O}(1/J^2)$ without any logarithmic factor in its leading term. The proof is effectively identical to that of our Theorem 3.4. In essence, we use the loss convergence upper bound $L(J)$ in the middle of the proof: see Equation (45). Thus, letting $L(J) = \mathcal{O}(1/J)$ and

following the same logic as the rest of the proof leads to the desired result: $|\mathcal{F}_{\text{cyc}}(J)| = \mathcal{O}(1/J^2)$. Although this bound seems better than that in Theorem 3.4 in terms of the vanishing speed, it inherits the same problem that the asymptotic $\mathcal{O}(1/J)$ loss convergence bound has: it holds only when $J$ is sufficiently large, thereby may fail to capture the dynamics in early (observable) cycles.

### E.4 NON-ASYMPTOTIC LOSS CONVERGENCE ANALYSIS (PROOF OF THEOREM 3.3)

In this section, we show non-asymptotic loss convergence, as stated below:

**Theorem 3.3.** *Suppose that Assumptions 3.1, 3.3, and 3.5 hold under the cyclic task ordering setup. Then, if we choose a learning rate satisfying $\eta < \frac{\phi^2}{4K\beta\sigma_{\max}^3(M\phi+\sigma_{\max})}$, we have for any $m \in [0:M-1]$ and $k \in [0:K-1]$ that*

$$\mathcal{L}(\boldsymbol{w}_k^{(MJ+m)}) \leq \left(|S| + \frac{\sum_{i=0}^{m-1}|S_i| + \frac{k}{K}|S_m|}{J}\right)\ell(\ln MJ) + \frac{\|\boldsymbol{w}_0^{(0)} - \hat{\boldsymbol{w}}\ln MJ\|^2}{2\eta KJ} + \frac{D_1}{J}$$
$$+ \left(|I| - |S| + \frac{\sum_{i=0}^{m-1}(|I_i| - |S_i|) + \frac{k}{K}(|I_m| - |S_m|)}{J}\right)\ell(\theta\ln MJ),$$

*where $\theta > 1$ is the second margin defined in Section 3.1, and $D_1$ is a constant independent to $J$.*

We first present three main lemmas to prove Theorem 3.3. The first one is an extension of Lemma E.2. When $M$ tasks are given in a cyclic order, the following lemma holds.

**Lemma E.7.** *Let any $t \in \mathbb{N}$, $l \in [0:K-1]$, $m \in [0:M-1]$ and $k \in [0:K-1]$ such that $l \geq k$ if $m = 0$ but $l \leq k$ if $m = M$. Then, for any $(t, l, m, k)$ satisfying these conditions,*

$$\left\|\boldsymbol{w}_l^{(t+m)} - \boldsymbol{w}_k^{(t)} + \eta\left((K-k+1)\nabla\mathcal{L}^{(t)}(\boldsymbol{w}_k^{(t)})K\sum_{i=1}^{m-1}\nabla\mathcal{L}^{(t+i)}(\boldsymbol{w}_k^{(t)}) + l\nabla\mathcal{L}^{(t+m)}(\boldsymbol{w}_k^{(t)})\right)\right\|$$

$$\leq \frac{\eta^2(mK+l-k)K\sigma_{\max}^3\beta}{\phi\{1-\eta(mK+l-k)\sigma_{\max}^2\beta\}}\left\|\nabla\mathcal{L}(\boldsymbol{w}_k^{(t)})\right\|,$$

$$\left\|\boldsymbol{w}_l^{(t+m)} - \boldsymbol{w}_k^{(t)}\right\| \leq \frac{\eta K\sigma_{\max}}{\phi\{1-\eta(mK+l-k)\sigma_{\max}^2\beta\}}\left\|\nabla\mathcal{L}(\boldsymbol{w}_k^{(t)})\right\|,$$

$$\left\|\nabla\mathcal{L}(\boldsymbol{w}_l^{(t+m)}) - \nabla\mathcal{L}(\boldsymbol{w}_k^{(t)})\right\| \leq \frac{\eta K\sigma_{\max}^3\beta}{\phi\{1-\eta(mK+l-k)\sigma_{\max}^2\beta\}}\left\|\nabla\mathcal{L}(\boldsymbol{w}_k^{(l)})\right\|.$$

*Proof.* We omitted the proof since there are only a few changes from the proof of Appendix E.1.2. $\square$

The second and third lemmas represent two similar versions with respect to the common Gradient Descent setting, and the Continual Learning setting.

**Lemma E.8.** *Let $\mathcal{L}$ be a convex function. Suppose that there exists a $\beta \geq 0$ satisfying that, if $\eta \in \left(0, \frac{1}{\beta}\right)$, the iterates $(\boldsymbol{w}_0, \ldots, \boldsymbol{w}_t)$ defined with an update rule $\boldsymbol{w}_{j+1} := \boldsymbol{w}_j - \eta\nabla\mathcal{L}(\boldsymbol{w}_j)$ satisfy*

$$\mathcal{L}(\boldsymbol{w}_{j+1}) \leq \mathcal{L}(\boldsymbol{w}_j) - \eta\left(1 - \eta\beta\right)\left\|\nabla\mathcal{L}(\boldsymbol{w}_j)\right\|^2$$

*for all $j \in [0:t-1]$. Then, for any $\boldsymbol{z} \in \mathbb{R}^d$,*

$$2\sum_{j=0}^{t-1}\eta\left(\mathcal{L}(\boldsymbol{w}_j) - \mathcal{L}(\boldsymbol{z})\right) - \sum_{j=0}^{t-1}\frac{\eta}{1-\eta\beta}\left(\mathcal{L}(\boldsymbol{w}_j) - \mathcal{L}(\boldsymbol{w}_{j+1})\right) \leq \|\boldsymbol{w}_0 - \boldsymbol{z}\|^2 - \|\boldsymbol{w}_t - \boldsymbol{z}\|^2.$$

*Proof.* See Appendix E.4.1. $\square$

**Lemma E.9.** *Consider the jointly separable continual linear classification problem with cyclic task ordering. Assume that $\ell(\cdot)$ is convex and $\beta$-smooth, there by the total training loss $\mathcal{L}$ is a convex & $\sigma_{\max}^2\beta$-smooth function. Assume that there exists $\beta' \geq 0$ satisfying that: if*

$\eta \le \min\{\frac{1}{2MK\sigma_{\max}^2\beta}, \frac{1}{2K\beta'}\}$, *the iterates* $\left(\boldsymbol{w}_0^{(0)}, \ldots, \boldsymbol{w}_{K-1}^{(0)}, \boldsymbol{w}_0^{(1)}, \ldots, \boldsymbol{w}_{K-1}^{(MJ+M)}\right)$ *defined with the update rules* $\boldsymbol{w}_{q+1}^{(p)} := \boldsymbol{w}_q^{(p)} - \eta \nabla \mathcal{L}^{(p)}(\boldsymbol{w}_q^{(p)})$ *and* $\boldsymbol{w}_0^{(p+1)} := \boldsymbol{w}_K^{(p)}$ *satisfy*

$$\mathcal{L}(\boldsymbol{w}_k^{(Mj+M+m)}) \le \mathcal{L}(\boldsymbol{w}_k^{(Mj+m)}) - \eta K (1 - \eta K \beta') \left\|\nabla \mathcal{L}(\boldsymbol{w}_k^{(Mj+m)})\right\|^2$$

*for each* $j \in [0:J-1]$, $m \in [0:M-1]$, *and* $k \in [0:K-1]$. **Then,** *for any* $\boldsymbol{z} \in \mathbb{R}^d$,

$$2 \sum_{j=0}^{J-1} \eta K \left(\mathcal{L}(\boldsymbol{w}_k^{(Mj+M+m)}) - \mathcal{L}(\boldsymbol{z})\right)$$

$$- \frac{2\eta M K \sigma_{\max}^4 \beta}{\phi^2(1 - \eta M K \sigma_{\max}^2\beta)^2} \sum_{j=0}^{J-1} \frac{\eta K}{1 - \eta K \beta'} \left(\mathcal{L}(\boldsymbol{w}_k^{(Mj+m)}) - \mathcal{L}(\boldsymbol{w}_k^{(Mj+M+m)})\right)$$

$$\le \left\|\boldsymbol{w}_k^{(m)} - \boldsymbol{z}\right\|^2 - \left\|\boldsymbol{w}_k^{(MJ+m)} - \boldsymbol{z}\right\|^2.$$

*Proof.* See Appendix E.4.2. □

Note that Lemma E.9 holds only when jointly separable tasks are given cyclic, while Lemma E.8 always holds.

We follow the process of Appendix E.1 to show that it satisfies the condition in Lemma E.9. Since $\mathcal{L}$ is a $\sigma_{\max}^2\beta$-smooth function, For all $j \in [0:J-1], m \in [0:M-1], k \in [0:K-1]$, we get

$$\mathcal{L}(\boldsymbol{w}_k^{(Mj+M+m)}) - \mathcal{L}(\boldsymbol{w}_k^{(Mj+m)}) - \frac{\sigma_{\max}^2\beta}{2} \left\|\boldsymbol{w}_k^{(Mj+M+m)} - \boldsymbol{w}_k^{(Mj+m)}\right\|^2$$

$$\le \nabla \mathcal{L}(\boldsymbol{w}_k^{(Mj+m)})^\top (\boldsymbol{w}_k^{(Mj+M+m)} - \boldsymbol{w}_k^{(Mj+m)})$$

$$= \nabla \mathcal{L}(\boldsymbol{w}_k^{(Mj+m)})^\top (\boldsymbol{w}_k^{(Mj+M+m)} - \boldsymbol{w}_k^{(Mj+m)} - \eta K \nabla \mathcal{L}(\boldsymbol{w}_k^{(Mj+m)}) + \eta K \nabla \mathcal{L}(\boldsymbol{w}_k^{(Mj+m)}))$$

$$\le -\eta K \left\|\nabla \mathcal{L}(\boldsymbol{w}_k^{(Mj+m)})\right\|^2 + \left\|\nabla \mathcal{L}(\boldsymbol{w}_k^{(Mj+m)})\right\| \left\|\boldsymbol{w}_k^{(Mj+M+m)} - \boldsymbol{w}_k^{(Mj+m)} + \eta K \nabla \mathcal{L}(\boldsymbol{w}_k^{(Mj+m)})\right\|.$$

By Lemma E.7,

$$\mathcal{L}(\boldsymbol{w}_k^{(Mj+M+m)}) - \mathcal{L}(\boldsymbol{w}_k^{(Mj+m)}) - \frac{\sigma_{\max}^2\beta}{2} \cdot \frac{(\eta \sigma_{\max} K)^2}{\phi^2(1 - \eta M K \sigma_{\max}^2\beta)^2} \left\|\nabla \mathcal{L}(\boldsymbol{w}_k^{(Mj+m)})\right\|^2$$

$$\le -\eta K \left\|\nabla \mathcal{L}(\boldsymbol{w}_k^{(Mj+m)})\right\|^2 + \frac{\eta^2 M K^2 \sigma_{\max}^3\beta}{\phi(1 - \eta M K \sigma_{\max}^2\beta)} \left\|\nabla \mathcal{L}(\boldsymbol{w}_k^{(Mj+m)})\right\|^2.$$

Given that $\eta \le \frac{\phi^2}{4K\beta\sigma_{\max}^3(M\phi+\sigma_{\max})} < \frac{1}{2MK\sigma_{\max}^2\beta}$,

$$\mathcal{L}(\boldsymbol{w}_k^{(Mj+M+m)}) - \mathcal{L}(\boldsymbol{w}_k^{(Mj+m)})$$

$$\le -\eta K \{1 - \eta K \left(\frac{M\sigma_{\max}^3\beta}{\phi(1 - \eta M K \sigma_{\max}^2\beta)} + \frac{\sigma_{\max}^4\beta}{2\phi^2(1 - \eta M K \sigma_{\max}^2\beta)^2}\right)\} \left\|\nabla \mathcal{L}(\boldsymbol{w}_k^{(Mj+m)})\right\|^2$$

$$\le -\eta K \left(1 - \eta K \frac{2(M\phi + \sigma_{\max})\sigma_{\max}^3\beta}{\phi^2}\right) \left\|\nabla \mathcal{L}(\boldsymbol{w}_k^{(Mj+m)})\right\|^2$$

$$= -\eta K (1 - \eta K \beta') \left\|\nabla \mathcal{L}(\boldsymbol{w}_k^{(Mj+m)})\right\|^2, \tag{34}$$

where we set $\beta' := \frac{2(M\phi+\sigma_{\max})\sigma_{\max}^3\beta}{\phi^2}$.

Since Equation (34) holds for all $j \in [0:J-1], m \in [0:M-1], k \in [0:K-1]$ and $\eta < 1/2K\beta'$ is given, by Lemma E.9, we get

$$2 \sum_{j=0}^{J-1} \eta K \left(\mathcal{L}(\boldsymbol{w}_k^{(Mj+M+m)}) - \mathcal{L}(\boldsymbol{z})\right)$$

$$-\frac{2\eta MK\sigma_{\max}^4\beta}{\phi^2(1-\eta MK\sigma_{\max}^2\beta)^2}\sum_{j=0}^{J-1}\frac{\eta K}{1-\eta K\beta'}\left(\mathcal{L}(\boldsymbol{w}_k^{(Mj+m)})-\mathcal{L}(\boldsymbol{w}_k^{(Mj+M+m)})\right)$$

$$\leq\left\|\boldsymbol{w}_k^{(m)}-\boldsymbol{z}\right\|^2-\left\|\boldsymbol{w}_k^{(MJ+m)}-\boldsymbol{z}\right\|^2. \tag{35}$$

Given that $\eta<\frac{1}{2K\beta'}<\frac{1}{2MK\beta\sigma_{\max}^2}$ and $\mathcal{L}(\boldsymbol{w}_k^{(Mj+m)})$ is decreasing,

$$\frac{2\eta MK\sigma_{\max}^4\beta}{\phi^2(1-\eta MK\sigma_{\max}^2\beta)^2}\cdot\frac{\eta K}{1-\eta K\beta'}\left(\mathcal{L}(\boldsymbol{w}_k^{(Mj+m)})-\mathcal{L}(\boldsymbol{w}_k^{(Mj+M+m)})\right)$$

$$\leq\frac{8\sigma_{\max}^2}{\phi^2}\eta K\left(\mathcal{L}(\boldsymbol{w}_k^{(Mj+m)})-\mathcal{L}(\boldsymbol{w}_k^{(Mj+M+m)})\right). \tag{36}$$

Also,

$$\frac{8\sigma_{\max}^2}{\phi^2}\eta K\mathcal{L}(\boldsymbol{w}_k^{(MJ+m)})+2\eta KJ\mathcal{L}(\boldsymbol{w}_k^{(MJ+m)})-\frac{8\sigma_{\max}^2}{\phi^2}\eta K\mathcal{L}(\boldsymbol{w}_k^{(m)})$$

$$\leq\frac{8\sigma_{\max}^2}{\phi^2}\eta K\mathcal{L}(\boldsymbol{w}_k^{(MJ+m)})+2\eta K\sum_{j=1}^{J}\mathcal{L}(\boldsymbol{w}_k^{(Mj+m)})-\frac{8\sigma_{\max}^2}{\phi^2}\eta K\mathcal{L}(\boldsymbol{w}_k^{(m)})$$

$$=2\eta K\sum_{j=0}^{J-1}\mathcal{L}(\boldsymbol{w}_k^{(Mj+M+m)})-\frac{8\sigma_{\max}^2}{\phi^2}\eta K\sum_{j=0}^{J-1}\left(\mathcal{L}(\boldsymbol{w}_k^{(Mj+m)})-\mathcal{L}(\boldsymbol{w}_k^{(Mj+M+m)})\right). \tag{37}$$

Combining Equations (35), (36), and (37), we obtain

$$2\eta KJ\left(\mathcal{L}(\boldsymbol{w}_k^{(MJ+m)})-\mathcal{L}(\boldsymbol{z})\right)+\frac{8\sigma_{\max}^2}{\phi^2}\eta K\left(\mathcal{L}(\boldsymbol{w}_k^{(MJ+m)})-\mathcal{L}(\boldsymbol{w}_k^{(m)})\right)$$

$$\leq 2\sum_{j=0}^{J-1}\eta K\left(\mathcal{L}(\boldsymbol{w}_k^{(Mj+M+m)})-\mathcal{L}(\boldsymbol{z})\right)-\frac{8\sigma_{\max}^2}{\phi^2}\eta K\sum_{j=0}^{J-1}\left(\mathcal{L}(\boldsymbol{w}_k^{(Mj+m)})-\mathcal{L}(\boldsymbol{w}_k^{(Mj+M+m)})\right)$$

$$\leq\left\|\boldsymbol{w}_k^{(m)}-\boldsymbol{z}\right\|^2-\left\|\boldsymbol{w}_k^{(MJ+m)}-\boldsymbol{z}\right\|^2. \tag{38}$$

Now we examine the loss change in a cycle. For any $j\in[0:M-1],l\in[0:K-1]$,

$$\mathcal{L}_j(\boldsymbol{w}_{l+1}^{(j)})\leq\mathcal{L}_j(\boldsymbol{w}_l^{(j)})-\eta(1-\frac{\eta\sigma_{\max}^2\beta}{2})\left\|\nabla\mathcal{L}_j(\boldsymbol{w}_l^{(j)})\right\|^2.$$

Since $\eta<\frac{1}{2MK\beta\sigma_{\max}^2}$, $\mathcal{L}_j(\boldsymbol{w}_l^{(j)})$ decreases. Therefore for any $p\in[0:M-1],q\in[0:K-1]$,

$$2\eta q(\mathcal{L}_p(\boldsymbol{w}_{q+1}^{(p)})-\mathcal{L}_p(\boldsymbol{z}))\leq 2\sum_{l=1}^{q}\eta\left(\mathcal{L}_p(\boldsymbol{w}_l^{(p)})-\mathcal{L}_p(\boldsymbol{z})\right)$$

$$=2\sum_{l=0}^{q-1}\eta\left(\mathcal{L}_p(\boldsymbol{w}_l^{(p)})-\mathcal{L}_p(\boldsymbol{z})\right)+2\sum_{l=0}^{q-1}\eta\left(\mathcal{L}_p(\boldsymbol{w}_{l+1}^{(p)})-\mathcal{L}_p(\boldsymbol{w}_l^{(p)})\right)$$

$$\leq 2\sum_{l=0}^{q-1}\eta\left(\mathcal{L}_p(\boldsymbol{w}_l^{(p)})-\mathcal{L}_p(\boldsymbol{z})\right)-\sum_{l=0}^{t-1}\frac{\eta}{1-\eta\beta}\left(\mathcal{L}_p(\boldsymbol{w}_l^{(p)})-\mathcal{L}_p(\boldsymbol{w}_{l+1}^{(p)})\right)$$

$$\leq\left\|\boldsymbol{w}_0^{(p)}-\boldsymbol{z}\right\|^2-\left\|\boldsymbol{w}_q^{(p)}-\boldsymbol{z}\right\|^2,$$

where in the third line, we use $\frac{1}{1-\eta\beta}<2$, and in the last line, we use Lemma E.8.

By summing up, we obtain

$$\sum_{p=0}^{m-1}2\eta K\left(\mathcal{L}_p(\boldsymbol{w}_K^{(p)})-\mathcal{L}_p(\boldsymbol{z})\right)+2\eta k\left(\mathcal{L}_m(\boldsymbol{w}_k^{(m)})-\mathcal{L}_m(\boldsymbol{z})\right)\leq\left\|\boldsymbol{w}_0^{(0)}-\boldsymbol{z}\right\|^2-\left\|\boldsymbol{w}_k^{(m)}-\boldsymbol{z}\right\|^2. \tag{39}$$

At last, $\mathcal{L}(\boldsymbol{w}_k^{(m)})$ is bounded by $\mathcal{L}(\boldsymbol{w}_0^{(0)})$ as follows:

$$
\begin{aligned}
\mathcal{L}(\boldsymbol{w}_k^{(m)}) - \mathcal{L}(\boldsymbol{w}_0^{(0)}) &\leq \nabla \mathcal{L}(\boldsymbol{w}_0^{(0)})^\top \left(\boldsymbol{w}_k^{(m)} - \boldsymbol{w}_0^{(0)}\right) + \frac{\sigma_{\max}^2 \beta}{2} \left\|\boldsymbol{w}_k^{(m)} - \boldsymbol{w}_0^{(0)}\right\|^2 \\
&\leq \left(\left\|\nabla \mathcal{L}(\boldsymbol{w}_0^{(0)})\right\| + \frac{\sigma_{\max}^2 \beta}{2} \left\|\boldsymbol{w}_k^{(m)} - \boldsymbol{w}_0^{(0)}\right\|\right) \left\|\boldsymbol{w}_k^{(m)} - \boldsymbol{w}_0^{(0)}\right\| \\
&\leq \left(1 + \frac{\eta K \sigma_{\max}^3 \beta}{\phi(1 - \eta M K \sigma_{\max}^2 \beta)}\right) \frac{\eta K \sigma_{\max}}{\phi(1 - \eta M K \sigma_{\max}^2 \beta)} \left\|\nabla \mathcal{L}(\boldsymbol{w}_0^{(0)})\right\|^2 \\
&= D_0 \left\|\nabla \mathcal{L}(\boldsymbol{w}_0^{(0)})\right\|^2,
\end{aligned}
\tag{40}
$$

where we use the smoothness in the first inequality, we use Cauchy-Schwarz in the second inequality, and in the last line, we use the fact $\left\|\boldsymbol{w}_k^{(m)} - \boldsymbol{w}_0^{(0)}\right\| \leq \frac{\eta K \sigma_{\max}}{\phi(1 - \eta M K \sigma_{\max}^2 \beta)} \left\|\nabla \mathcal{L}(\boldsymbol{w}_0^{(0)})\right\|$ held by Lemma E.7. We set $D_0 := \left(1 + \frac{\eta K \sigma_{\max}^3 \beta}{\phi(1 - \eta M K \sigma_{\max}^2 \beta)}\right) \frac{\eta K \sigma_{\max}}{\phi(1 - \eta M K \sigma_{\max}^2 \beta)}$.

Combining Equations (38), (39), and (40), we obtain

$$
\begin{aligned}
\mathcal{L}(\boldsymbol{w}_k^{(MJ+m)}) \leq{}& \mathcal{L}(\boldsymbol{z}) + \frac{1}{J} \sum_{p=0}^{m-1} \mathcal{L}_p(\boldsymbol{z}) + \frac{1}{J} \cdot \frac{q}{K} \mathcal{L}_m(\boldsymbol{z}) \\
&+ \frac{4\sigma_{\max}^2}{\phi^2} \frac{\mathcal{L}(w_0^{(0)}) + D_0 \left\|\nabla \mathcal{L}(\boldsymbol{w}_0^{(0)})\right\|^2}{J} + \frac{\left\|\boldsymbol{w}_0^{(0)} - \boldsymbol{z}\right\|^2}{2\eta K J}.
\end{aligned}
$$

Now let us choose $\boldsymbol{z} := \hat{\boldsymbol{w}} \ln MJ$. Using the fact $\mathcal{L}_j(\hat{\boldsymbol{w}} \ln MJ) \leq |S_j| \ell(\ln MJ) + (|I_j| - |S_j|) \ell(\theta \ln MJ)$ (since $\ell$ is monotonically decreasing), we can finish the proof with

$$
D_1 := \frac{4\sigma_{\max}^2}{\phi^2} \left(\mathcal{L}(w_0^{(0)}) + \left(1 + \frac{\eta K \sigma_{\max}^3 \beta}{\phi(1 - \eta M K \sigma_{\max}^2 \beta)}\right) \frac{\eta K \sigma_{\max}}{\phi(1 - \eta M K \sigma_{\max}^2 \beta)} \left\|\nabla \mathcal{L}(\boldsymbol{w}_0^{(0)})\right\|^2\right).
$$

### E.4.1 PROOF OF LEMMA E.8

Here is the restatement of the lemma.

**Lemma E.8.** *Let $\mathcal{L}$ be a convex function. Suppose that there exists a $\beta \geq 0$ satisfying that, if $\eta \in \left(0, \frac{1}{\beta}\right)$, the iterates $(\boldsymbol{w}_0, \ldots, \boldsymbol{w}_t)$ defined with an update rule $\boldsymbol{w}_{j+1} := \boldsymbol{w}_j - \eta \nabla \mathcal{L}(\boldsymbol{w}_j)$ satisfy*

$$
\mathcal{L}(\boldsymbol{w}_{j+1}) \leq \mathcal{L}(\boldsymbol{w}_j) - \eta(1 - \eta \beta) \|\nabla \mathcal{L}(\boldsymbol{w}_j)\|^2
$$

*for all $j \in [0 : t-1]$. Then, for any $\boldsymbol{z} \in \mathbb{R}^d$,*

$$
2 \sum_{j=0}^{t-1} \eta\left(\mathcal{L}(\boldsymbol{w}_j) - \mathcal{L}(\boldsymbol{z})\right) - \sum_{j=0}^{t-1} \frac{\eta}{1 - \eta \beta} \left(\mathcal{L}(\boldsymbol{w}_j) - \mathcal{L}(\boldsymbol{w}_{j+1})\right) \leq \|\boldsymbol{w}_0 - \boldsymbol{z}\|^2 - \|\boldsymbol{w}_t - \boldsymbol{z}\|^2.
$$

This is a well-known property of gradient descent iterates applied to a smooth convex objective function. However, we contain the proof for completeness.

For any $j \geq 0$ and $\boldsymbol{z} \in \mathbb{R}^d$,

$$
\begin{aligned}
\|\boldsymbol{w}_{j+1} - \boldsymbol{z}\|^2 &= \|\boldsymbol{w}_j - \boldsymbol{z}\|^2 + 2\eta \langle \nabla \mathcal{L}(\boldsymbol{w}_j), \boldsymbol{z} - \boldsymbol{w}_j \rangle + \eta^2 \|\nabla \mathcal{L}(\boldsymbol{w}_j)\|^2 \\
&\leq \|\boldsymbol{w}_j\|^2 + 2\eta(\mathcal{L}(\boldsymbol{z}) - \mathcal{L}(\boldsymbol{w}_j)) + \eta^2 \|\nabla \mathcal{L}(\boldsymbol{w}_j)\|^2 \\
&\leq \|\boldsymbol{w}_j\|^2 + 2\eta(\mathcal{L}(\boldsymbol{z}) - \mathcal{L}(\boldsymbol{w}_j)) + \frac{\eta}{1 - \eta \beta} \left(\mathcal{L}(\boldsymbol{w}_j) - \mathcal{L}(\boldsymbol{w}_{j+1})\right),
\end{aligned}
$$

where the first line comes from convexity and the second line comes from the condition of our lemma. Summing up the inequality over all $j \in [0 : t-1]$, we eventually get the desired result:

$$
2 \sum_{j=0}^{t-1} \eta\left(\mathcal{L}(\boldsymbol{w}_j) - \mathcal{L}(\boldsymbol{z})\right) - \sum_{j=0}^{t-1} \frac{\eta}{1 - \eta \beta} \left(\mathcal{L}(\boldsymbol{w}_j) - \mathcal{L}(\boldsymbol{w}_{j+1})\right) \leq \|\boldsymbol{w}_0 - \boldsymbol{z}\|^2 - \|\boldsymbol{w}_t - \boldsymbol{z}\|^2.
$$

### E.4.2 PROOF OF LEMMA E.9

Here is the restatement of the lemma.

**Lemma E.9.** *Consider the jointly separable continual linear classification problem with cyclic task ordering.* *Assume* *that $\ell(\cdot)$ is convex and $\beta$-smooth, there by the total training loss $\mathcal{L}$ is a convex & $\sigma_{\max}^2\beta$-smooth function.* *Assume* *that there exists $\beta' \geq 0$ satisfying that: if $\eta \leq \min\{\frac{1}{2MK\sigma_{\max}^2\beta}, \frac{1}{2K\beta'}\}$, the iterates $\left(\boldsymbol{w}_0^{(0)}, \ldots, \boldsymbol{w}_{K-1}^{(0)}, \boldsymbol{w}_0^{(1)}, \ldots, \boldsymbol{w}_{K-1}^{(MJ+M)}\right)$ defined with the update rules $\boldsymbol{w}_{q+1}^{(p)} := \boldsymbol{w}_q^{(p)} - \eta\nabla\mathcal{L}^{(p)}(\boldsymbol{w}_q^{(p)})$ and $\boldsymbol{w}_0^{(p+1)} := \boldsymbol{w}_K^{(p)}$ satisfy*

$$\mathcal{L}(\boldsymbol{w}_k^{(Mj+M+m)}) \leq \mathcal{L}(\boldsymbol{w}_k^{(Mj+m)}) - \eta K\left(1 - \eta K\beta'\right)\left\|\nabla\mathcal{L}(\boldsymbol{w}_k^{(Mj+m)})\right\|^2$$

*for each $j \in [0 : J-1]$, $m \in [0 : M-1]$, and $k \in [0 : K-1]$.* ***Then,*** *for any $\boldsymbol{z} \in \mathbb{R}^d$,*

$$2\sum_{j=0}^{J-1}\eta K\left(\mathcal{L}(\boldsymbol{w}_k^{(Mj+M+m)}) - \mathcal{L}(\boldsymbol{z})\right)$$

$$-\frac{2\eta MK\sigma_{\max}^4\beta}{\phi^2(1-\eta MK\sigma_{\max}^2\beta)^2}\sum_{j=0}^{J-1}\frac{\eta K}{1-\eta K\beta'}\left(\mathcal{L}(\boldsymbol{w}_k^{(Mj+m)}) - \mathcal{L}(\boldsymbol{w}_k^{(Mj+M+m)})\right)$$

$$\leq \left\|\boldsymbol{w}_k^{(m)} - \boldsymbol{z}\right\|^2 - \left\|\boldsymbol{w}_k^{(MJ+m)} - \boldsymbol{z}\right\|^2.$$

Without loss of generality, let us assume $m = 0, k = 0$.

For any $j$ and $\boldsymbol{z} \in \mathbb{R}^d$,

$$\left\|\boldsymbol{w}_0^{(Mj+M)} - \boldsymbol{z}\right\|^2 = \left\|\boldsymbol{w}_0^{(Mj)} - \eta\sum_{p=0}^{M-1}\sum_{q=0}^{K-1}\nabla\mathcal{L}_p(\boldsymbol{w}_q^{(Mj+p)}) - \boldsymbol{z}\right\|^2$$

$$= \left\|\boldsymbol{w}_0^{(Mj)} - \boldsymbol{z}\right\|^2 + 2\eta\sum_{p=0}^{M-1}\sum_{q=0}^{K-1}\left\langle\nabla\mathcal{L}_p(\boldsymbol{w}_q^{(Mj+p)}), \boldsymbol{z} - \boldsymbol{w}_0^{(Mj)}\right\rangle + \eta^2\left\|\sum_{p=0}^{M-1}\sum_{q=0}^{K-1}\nabla\mathcal{L}_p(\boldsymbol{w}_q^{(Mj+p)})\right\|^2$$

$$= \left\|\boldsymbol{w}_0^{(Mj)} - \boldsymbol{z}\right\|^2 + 2\eta\sum_{p=0}^{M-1}\sum_{q=0}^{K-1}\left\langle\nabla\mathcal{L}_p(\boldsymbol{w}_q^{(Mj+p)}), \boldsymbol{z} - \boldsymbol{w}_q^{(Mj+p)}\right\rangle$$

$$+ 2\eta\sum_{p=0}^{M-1}\sum_{q=0}^{K-1}\left\langle\nabla\mathcal{L}_p(\boldsymbol{w}_q^{(Mj+p)}), \boldsymbol{w}_q^{(Mj+p)} - \boldsymbol{w}_0^{(Mj)}\right\rangle + \eta^2\left\|\sum_{p=0}^{M-1}\sum_{q=0}^{K-1}\nabla\mathcal{L}_p(\boldsymbol{w}_q^{(Mj+p)})\right\|^2. \quad (41)$$

By convexity,

$$\sum_{p=0}^{M-1}\sum_{q=0}^{K-1}\left\langle\nabla\mathcal{L}_p(\boldsymbol{w}_q^{(Mj+p)}), \boldsymbol{z} - \boldsymbol{w}_q^{(Mj+p)}\right\rangle \leq MK\mathcal{L}(\boldsymbol{z}) - \sum_{p=0}^{M-1}\sum_{q=0}^{K-1}\mathcal{L}_p(\boldsymbol{w}_q^{(Mj+p)}). \quad (42)$$

Combined with Equations (41) and (42), we get

$$\left\|\boldsymbol{w}_0^{(Mj+M)} - \boldsymbol{z}\right\|^2 - \left\|\boldsymbol{w}_0^{(Mj)} - \boldsymbol{z}\right\|^2 \leq 2\eta K\mathcal{L}(\boldsymbol{z}) - 2\eta\sum_{p=0}^{M-1}\sum_{q=0}^{K-1}\mathcal{L}_p(\boldsymbol{w}_q^{(Mj+p)})$$

$$+ 2\eta\sum_{p=0}^{M-1}\sum_{q=0}^{K-1}\langle\nabla\mathcal{L}_p(\boldsymbol{w}_q^{(Mj+p)}), \boldsymbol{w}_q^{(Mj+p)} - \boldsymbol{w}_0^{(Mj)}\rangle + \eta^2\left\|\sum_{p=0}^{M-1}\sum_{q=0}^{K-1}\nabla\mathcal{L}_p(\boldsymbol{w}_q^{(Mj+p)})\right\|^2. \quad (43)$$

By $\sigma_{\max}^2\beta$-smoothness,

$$\sum_{p=0}^{M-1}\sum_{q=0}^{K-1}\langle\nabla\mathcal{L}_p(\boldsymbol{w}_q^{(Mj+p)}), \boldsymbol{z} - \boldsymbol{w}_q^{(Mj+p)}\rangle$$

$$\geq K\mathcal{L}(\boldsymbol{z}) - \sum_{p=0}^{M-1}\sum_{q=0}^{K-1}\mathcal{L}_p(\boldsymbol{w}_q^{(Mj+p)}) - \frac{\sigma_{\max}^2\beta}{2}\sum_{p=0}^{M-1}\sum_{q=0}^{K-1}\left\|\boldsymbol{z}-\boldsymbol{w}_q^{(Mj+p)}\right\|^2.$$

Apply smoothness on Equation (41), and let $\boldsymbol{z} := \boldsymbol{w}_0^{(Mj+M)}$ then we get

$$
\begin{aligned}
0 \geq{}& \left\|\boldsymbol{w}_0^{(Mj)} - \boldsymbol{w}_0^{(Mj+M)}\right\|^2 + 2\eta K\mathcal{L}(\boldsymbol{w}_0^{(Mj+M)}) \\
&- 2\eta\sum_{p=0}^{M-1}\sum_{q=0}^{K-1}\mathcal{L}_p(\boldsymbol{w}_q^{(Mj+p)}) - \eta\sigma_{\max}^2\beta\sum_{p=0}^{M-1}\sum_{q=0}^{K-1}\left\|\boldsymbol{w}_0^{(Mj+M)} - \boldsymbol{w}_q^{(Mj+p)}\right\|^2 \\
&+ 2\eta\sum_{p=0}^{M-1}\sum_{q=0}^{K-1}\left\langle\nabla\mathcal{L}_p(\boldsymbol{w}_q^{(Mj+p)}), \boldsymbol{w}_q^{(Mj+p)} - \boldsymbol{w}_0^{(Mj)}\right\rangle + \eta^2\left\|\sum_{p=0}^{M-1}\sum_{q=0}^{K-1}\nabla\mathcal{L}_p(\boldsymbol{w}_q^{(Mj+p)})\right\|^2.
\end{aligned}
\tag{44}
$$

Combined with Equations (43) and (44),

$$
\begin{aligned}
& \left\|\boldsymbol{w}_0^{(Mj+M)} - \boldsymbol{z}\right\|^2 - \left\|\boldsymbol{w}_0^{(Mj)} - \boldsymbol{z}\right\|^2 \leq 2\eta K\mathcal{L}(\boldsymbol{z}) - 2\eta K\mathcal{L}(\boldsymbol{w}_0^{(Mj+M)}) \\
&- \left\|\boldsymbol{w}_0^{(Mj)} - \boldsymbol{w}_0^{(Mj+M)}\right\|^2 + \eta\sigma_{\max}^2\beta\sum_{p=0}^{M-1}\sum_{q=0}^{K-1}\left\|\boldsymbol{w}_0^{(Mj+M)} - \boldsymbol{w}_q^{(Mj+p)}\right\|^2 \\[4pt]
&\leq 2\eta K\mathcal{L}(\boldsymbol{z}) - 2\eta K\mathcal{L}(\boldsymbol{w}_0^{(Mj+M)}) - \left\|\boldsymbol{w}_0^{(Mj)} - \boldsymbol{w}_0^{(Mj+M)}\right\|^2 \\
&+ 2\eta MK\sigma_{\max}^2\beta\left\|\boldsymbol{w}_0^{(Mj)} - \boldsymbol{w}_0^{(Mj+M)}\right\|^2 + 2\eta\sigma_{\max}^2\beta\sum_{p=0}^{M-1}\sum_{q=0}^{K-1}\left\|\boldsymbol{w}_0^{(Mj)} - \boldsymbol{w}_q^{(Mj+p)}\right\|^2 \\[4pt]
&\leq 2\eta K\mathcal{L}(\boldsymbol{z}) - 2\eta K\mathcal{L}(\boldsymbol{w}_0^{(Mj+M)}) + 2\eta\sigma_{\max}^2\beta\sum_{p=0}^{M-1}\sum_{q=0}^{K-1}\left\|\boldsymbol{w}_0^{(Mj)} - \boldsymbol{w}_q^{(Mj+p)}\right\|^2,
\end{aligned}
$$

where in the last line we use $\eta \leq \frac{1}{2MK\sigma_{\max}^2\beta}$. Finally, by Lemma E.7,

$$
\begin{aligned}
& \left\|\boldsymbol{w}_0^{(Mj+M)} - \boldsymbol{z}\right\|^2 - \left\|\boldsymbol{w}_0^{(Mj)} - \boldsymbol{z}\right\|^2 \\
&\leq 2\eta K\left(\mathcal{L}(\boldsymbol{z}) - \mathcal{L}(\boldsymbol{w}_0^{(Mj+M)})\right) + \frac{2\eta^3 MK^3\sigma_{\max}^4\beta}{\phi^2(1-\eta MK\sigma_{\max}^2\beta)^2}\left\|\nabla\mathcal{L}(\boldsymbol{w}_0^{(Mj)})\right\|^2 \\
&\leq 2\eta K\left(\mathcal{L}(\boldsymbol{z}) - \mathcal{L}(\boldsymbol{w}_0^{(Mj+M)})\right) - \frac{2\eta MK\sigma_{\max}^4\beta}{\phi^2(1-\eta MK\sigma_{\max}^2\beta)^2}\frac{\eta K}{1-\eta K\beta'}\left(\mathcal{L}(\boldsymbol{w}_0^{(Mj)}) - \mathcal{L}(\boldsymbol{w}_0^{(Mj+M)})\right).
\end{aligned}
$$

By adding all $j \in \{0,\cdots,J-1\}$, we can finish the proof.

## E.5 FORGETTING ANALYSIS (PROOF OF THEOREM 3.4)

We prove Theorem 3.4 here, which is restated below for readability.

**Theorem 3.4.** *Suppose Assumption 3.1 and let $\ell(u) = \ln(1 + e^{-u})$ be the logistic loss, which satisfies Assumptions 3.3, 3.4, and 3.5. If the learning rate satisfies $\eta < \frac{\phi^2}{4K\beta\sigma_{\max}^3(M\phi+\sigma_{\max})}$, then the cycle-averaged forgetting $\mathcal{F}_{\mathrm{cyc}}(J)$ for cycle $J$ satisfies the following upper and lower bounds:*

$$-\eta K\cdot\{L(J)\}^2\cdot\frac{\sum_{p\neq q}A_{p,q}^+}{M} \leq \mathcal{F}_{\mathrm{cyc}}(J) \leq \eta K\cdot\{L(J)\}^2\cdot\frac{\sum_{p\neq q}A_{p,q}^-}{M},$$

*where $L(J) = \mathcal{O}\left(\frac{\ln^2 J}{J}\right)$ and $A_{p,q}^+ := \sum_{\substack{(i,j)\in I_p\times I_q \\ \boldsymbol{x}_i^\top\boldsymbol{x}_j>0}}\boldsymbol{x}_i^\top\boldsymbol{x}_j$, $A_{p,q}^- := \sum_{\substack{(i,j)\in I_p\times I_q \\ \boldsymbol{x}_i^\top\boldsymbol{x}_j<0}}\left|\boldsymbol{x}_i^\top\boldsymbol{x}_j\right|$.*

By Theorem 3.3, the joint training loss at cycle $J$ is bounded as

$$\mathcal{L}(\boldsymbol{w}_k^{(MJ+m)}) \leq L(J) \tag{45}$$

where

$$L(J) := \frac{1}{J}\left(\left(|S| + \frac{|I| - |S|}{(MJ)^{\theta-1}}\right)\left(1 + \frac{1}{MJ}\right) + \frac{\|\boldsymbol{w}_0^{(0)} - \hat{\boldsymbol{w}}\ln MJ\|^2}{2\eta K} + D_1\right) = \mathcal{O}\left(\frac{\ln^2 J}{J}\right),$$

$$D_1 := \frac{4\sigma_{\max}^2}{\phi^2}\left(\mathcal{L}(w_0^{(0)}) + D_0\left\|\nabla\mathcal{L}(\boldsymbol{w}_0^{(0)})\right\|^2\right).$$

Therefore, the following holds:

$$\forall s \in I, \forall m \in [0:M-1], \forall k \in [0:K-1]: \quad \boldsymbol{x}_s^\top \boldsymbol{w}_k^{(MJ+m)} \geq \ell^{-1}\left(L(t)\right).$$

Now, we analyze the change of each task in one cycle. For the upper bound,

$$\mathcal{L}_m(\boldsymbol{w}_0^{(MJ+M)}) - \mathcal{L}_m(\boldsymbol{w}_K^{(MJ+m)})$$

$$\leq -\eta\sum_{p=m+1}^{M-1}\sum_{q=0}^{K-1}\nabla\mathcal{L}_m(\boldsymbol{w}_0^{(MJ+M)})^\top\nabla\mathcal{L}_p(\boldsymbol{w}_q^{(MJ+p)}) \tag{46}$$

$$\leq -\eta\sum_{p=m+1}^{M-1}\sum_{q=0}^{K-1}\sum_{\substack{(i,j)\in I_m\times I_p \\ \boldsymbol{x}_i^\top\boldsymbol{x}_j<0}}\ell'(\boldsymbol{x}_i^\top\boldsymbol{w}_0^{(MJ+M)})\ell'(\boldsymbol{x}_j^\top\boldsymbol{w}_q^{(MJ+p)})\boldsymbol{x}_i^\top\boldsymbol{x}_j$$

$$\leq -\eta\sum_{p=m+1}^{M-1}\sum_{q=0}^{K-1}\left[\ell'\left(\ell^{-1}\left(L(J)\right)\right)\right]^2\sum_{\substack{(i,j)\in I_m\times I_p \\ \boldsymbol{x}_i^\top\boldsymbol{x}_j<0}}\boldsymbol{x}_i^\top\boldsymbol{x}_j \tag{47}$$

$$\leq -\eta K\{L(J)\}^2\sum_{p=m+1}^{M-1}\sum_{\substack{(i,j)\in I_m\times I_p \\ \boldsymbol{x}_i^\top\boldsymbol{x}_j<0}}\boldsymbol{x}_i^\top\boldsymbol{x}_j, \tag{48}$$

where in (46) we use convexity, in (47) we use the condition that $\ell'$ is a negative function monotonically increasing to zero. Equation (48) holds by the fact $\forall x : \ell'(x) = \ell_{\log}'(x) \geq -\exp(-x)$ and $\forall x : \ell^{-1}(x) = \ell_{\log}^{-1}(x) \geq -\log(x)$. Likewise, we can get a lower bound as follows:

$$\mathcal{L}_m(\boldsymbol{w}_0^{(MJ+M)}) - \mathcal{L}_m(\boldsymbol{w}_K^{(MJ+m)})$$

$$\geq -\eta\sum_{p=m+1}^{M-1}\sum_{q=0}^{K-1}\nabla\mathcal{L}_m(\boldsymbol{w}_k^{(MJ+m)})^\top\nabla\mathcal{L}_p(\boldsymbol{w}_q^{(MJ+p)})$$

$$\geq -\eta\sum_{p=m+1}^{M-1}\sum_{q=0}^{K-1}\sum_{\substack{(i,j)\in I_m\times I_p \\ \boldsymbol{x}_i^\top\boldsymbol{x}_j>0}}\ell'(\boldsymbol{x}_i^\top\boldsymbol{w}_k^{(MJ+m)})\ell'(\boldsymbol{x}_j^\top\boldsymbol{w}_q^{(MJ+p)})\boldsymbol{x}_i^\top\boldsymbol{x}_j$$

$$\geq -\eta\sum_{p=m+1}^{M-1}\sum_{q=0}^{K-1}\left[\ell'\left(\ell^{-1}\left(L(J)\right)\right)\right]^2\sum_{\substack{(i,j)\in I_m\times I_p \\ \boldsymbol{x}_i^\top\boldsymbol{x}_j>0}}\boldsymbol{x}_i^\top\boldsymbol{x}_j$$

$$\geq -\eta K\{L(J)\}^2\sum_{p=m+1}^{M-1}\sum_{\substack{(i,j)\in I_m\times I_p \\ \boldsymbol{x}_i^\top\boldsymbol{x}_j>0}}\boldsymbol{x}_i^\top\boldsymbol{x}_j.$$

Define

$$A_{p,q}^+ := \sum_{\substack{(i,j)\in I_p\times I_q \\ \boldsymbol{x}_i^\top\boldsymbol{x}_j>0}}\boldsymbol{x}_i^\top\boldsymbol{x}_j \geq 0 \quad\text{and}\quad A_{p,q}^- := \sum_{\substack{(i,j)\in I_p\times I_q \\ \boldsymbol{x}_i^\top\boldsymbol{x}_j<0}}\left|\boldsymbol{x}_i^\top\boldsymbol{x}_j\right| \geq 0.$$

Since

$$\sum_{m=0}^{M-1}\sum_{p=m+1}^{M-1}\sum_{\substack{(i,j)\in I_m\times I_p \\ \boldsymbol{x}_i^\top\boldsymbol{x}_j>0}}\boldsymbol{x}_i^\top\boldsymbol{x}_j = \sum_{p\neq q}A_{p,q}^+,$$

$$-\sum_{m=0}^{M-1}\sum_{p=m+1}^{M-1}\sum_{\substack{(i,j)\in I_m\times I_p \\ \boldsymbol{x}_i^\top \boldsymbol{x}_j < 0}}\boldsymbol{x}_i^\top \boldsymbol{x}_j = \sum_{p\neq q}A_{p,q}^-,$$

we can conclude

$$-\eta K\{L(J)\}^2 \cdot \frac{\sum_{p\neq q}A_{p,q}^+}{M} \leq \frac{1}{M}\sum_{m=0}^{M-1}\mathcal{F}^{(MJ+m)}(MJ+M) \leq \eta K\{L(J)\}^2 \cdot \frac{\sum_{p\neq q}A_{p,q}^-}{M}.$$

# F  PROOFS FOR SECTION 4: RANDOM TASK ORDERING, JOINTLY SEPARABLE

## F.1  ASYMPTOTIC LOSS CONVERGENCE ANALYSIS (PROOF OF THEOREM 4.1)

Let us restate the theorem here for the sake of readability.

**Theorem 4.1.** *Let $\{w_k^{(t)}\}_{k\in[0:K-1],t\geq 0}$ be the sequence of GD iterates (2) from any starting point $w_0^{(0)}$, where tasks are given randomly. Under Assumptions 3.1 and 3.3, if the learning rate satisfies $\eta < \frac{2\phi^2}{\beta\sigma_{\max}^4}$, then the following statements hold with probability 1:*

1. *Loss converges to zero:* $\lim\limits_{t\to\infty} \mathcal{L}(w_k^{(t)}) = 0, \forall k \in [0 : K - 1]$.

2. *All data points are eventually classified correctly:* $\lim\limits_{t\to\infty} y_i x_i^\top w_k^{(t)} = 0, \forall k \in [0 : K - 1], i \in I$.

3. *Square sum of the change of weight is finite:* $\sum_{t=0}^{\infty}\sum_{k=0}^{K-1} \|w_{k+1}^{(t)} - w_k^{(t)}\|^2 < \infty$.

Since $\mathcal{L}$ is a $\sigma_{\max}^2\beta$-smooth function, we get

$$
\mathbb{E}\left[\mathcal{L}(w_{k+1}^{(t)})\right] - \mathbb{E}\left[\mathcal{L}(w_k^{(t)})\right]
$$

$$
\leq \mathbb{E}\left[\nabla\mathcal{L}(w_k^{(t)})^\top(w_{k+1}^{(t)} - w_k^{(t)})\right] + \frac{\sigma_{\max}^2\beta}{2}\mathbb{E}\left[\left\|w_{k+1}^{(t)} - w_k^{(t)}\right\|^2\right]
$$

$$
= \mathbb{E}\left[\mathbb{E}\left[\nabla\mathcal{L}(w_k^{(t)})^\top(w_{k+1}^{(t)} - w_k^{(t)}) \mid w_k^{(t)}\right]\right] + \frac{\sigma_{\max}^2\beta}{2}\mathbb{E}\left[\left\|w_{k+1}^{(t)} - w_k^{(t)}\right\|^2\right]
$$

$$
= \mathbb{E}\left[\mathbb{E}\left[\nabla\mathcal{L}(w_k^{(t)})^\top(w_{k+1}^{(t)} - w_k^{(t)}) \mid w_k^{(t)}\right]\right] + \frac{\sigma_{\max}^2\beta}{2}\eta^2\mathbb{E}\left[\left\|\sum_{s\in I} z_s^{(t)}\ell'(x_s^\top w_k^{(t)})x_s\right\|^2\right]
$$

$$
= -\frac{\eta}{M}\mathbb{E}\left[\left\|\nabla\mathcal{L}(w_k^{(t)})\right\|^2\right] + \frac{\sigma_{\max}^2\beta}{2}\eta^2\mathbb{E}\left[\left\|\sum_{s\in I} z_s^{(t)}\ell'(x_s^\top w_k^{(t)})x_s\right\|^2\right]
$$

$$
\leq -\frac{\eta}{M}\mathbb{E}\left[\left\|\nabla\mathcal{L}(w_k^{(t)})\right\|^2\right] + \frac{\sigma_{\max}^4\beta}{2}\eta^2\mathbb{E}\left[\sum_{s\in I}\left[z_s^{(t)}\ell'(x_s^\top w_k^{(t)})\right]^2\right]
$$

$$
= -\frac{\eta}{M}\mathbb{E}\left[\left\|\nabla\mathcal{L}(w_k^{(t)})\right\|^2\right] + \frac{\sigma_{\max}^4\beta}{2}\eta^2\sum_{s\in I}\mathbb{E}\left[(z_s^{(t)})^2\right]\mathbb{E}\left[\ell'(x_s^\top w_k^{(t)})^2\right]
$$

$$
= -\frac{\eta}{M}\mathbb{E}\left[\left\|\nabla\mathcal{L}(w_k^{(t)})\right\|^2\right] + \frac{\sigma_{\max}^4\beta}{2M}\eta^2\mathbb{E}\left[\sum_{s\in I}\ell'(x_s^\top w_k^{(t)})^2\right],
$$

where $z_s^{(t)}$ is a variable which is 1 when $x_s$ is in the task on stage $t$, or 0 otherwise. The second inequality comes from the fact $\forall \lambda_s \in \mathbb{R} : \left\|\sum_{s\in I}\lambda_s x_s\right\|_2 \leq \sigma_{\max}\sqrt{\sum_{s\in I}\lambda_s^2}$.

By applying Lemma E.1, which only depends on the total dataset (but not on the task ordering), we obtain

$$
\mathbb{E}\left[\mathcal{L}(w_{k+1}^{(t)})\right] - \mathbb{E}\left[\mathcal{L}(w_k^{(t)})\right] \leq -\frac{\eta}{M}\left(1 - \eta\frac{\sigma_{\max}^4\beta}{2\phi^2}\right)\mathbb{E}\left[\left\|\nabla\mathcal{L}(w_k^{(t)})\right\|^2\right]
$$

$$
= -\frac{\eta}{M}(1 - \eta\beta'')\mathbb{E}\left[\left\|\nabla\mathcal{L}(w_k^{(t)})\right\|^2\right], \tag{49}
$$

where $\beta'' := \frac{\sigma_{\max}^4\beta}{2\phi^2}$. Given that $\eta < \frac{1}{\beta''}$,

$$
\sum_{t=0}^{\infty}\sum_{k=0}^{K-1}\mathbb{E}\left[\left\|\nabla\mathcal{L}(w_k^{(t)})\right\|^2\right] \leq \frac{\mathcal{L}(w_0^{(0)}) - \lim_{t\to\infty}\mathbb{E}\left[\mathcal{L}(w_0^{(t)})\right]}{\frac{\eta}{M}(1 - \eta\beta'')} \leq \frac{M\mathcal{L}(w_0^{(0)})}{\eta(1 - \eta\beta'')} < \infty.
$$

Thus, by Fubini-Tonelli's theorem, we have $\mathbb{E}\left[\sum_{t=0}^{\infty}\sum_{k=0}^{K-1}\left\|\nabla\mathcal{L}(\boldsymbol{w}_k^{(t)})\right\|^2\right] < \infty$. Since a non-negative random variable with a finite mean is finite almost surely (because of Markov inequality), we have that $\sum_{t=0}^{\infty}\sum_{k=0}^{K-1}\left\|\nabla\mathcal{L}(\boldsymbol{w}_k^{(t)})\right\|^2$ is bounded with probability 1. The boundedness of infinite sum of nonzero elements implies $\forall k \in [0:K-1] : \lim_{t\to 0}\left\|\nabla\mathcal{L}(\boldsymbol{w}_k^{(t)})\right\|^2 = 0$. Combined with Lemma E.1, we obtain $\lim_{t\to 0}\ell'(x_i^\top \boldsymbol{w}_k^{(t)}) = 0, \forall i \in I, k \in [0:K-1]$. Since $\ell'(u) \to 0$ only when $u \to \infty$, $x_i^\top \boldsymbol{w}_k^{(t)} \to \infty, \forall i \in I, k \in [0:K-1]$. And $\lim_{t\to\infty}\mathcal{L}(\boldsymbol{w}_k^{(t)}) = 0, \forall k \in [0:K-1]$. Finally, followed by

$$\left\|\nabla\mathcal{L}(\boldsymbol{w}_k^{(t)})\right\| \geq \phi\sqrt{\sum_{i\in I}\left[\ell'(\boldsymbol{x}_i^\top \boldsymbol{w}_k^{(t)})\right]^2} \geq \phi\sqrt{\sum_{i\in I(t)}\left[\ell'(\boldsymbol{x}_i^\top \boldsymbol{w}_k^{(t)})\right]^2}$$

$$\geq \frac{\phi}{\sigma_{\max}}\left\|\sum_{i\in I(t)}\ell'(\boldsymbol{x}_i^\top \boldsymbol{w}_k^{(t)})x_i\right\| = \frac{\phi}{\sigma_{\max}}\eta^{-1}\left\|\boldsymbol{w}_{k+1}^{(t)} - \boldsymbol{w}_k^{(t)}\right\|.$$

We obtain that $\sum_{t=0}^{\infty}\sum_{k=0}^{K-1}\left\|\boldsymbol{w}_{k+1}^{(t)} - \boldsymbol{w}_k^{(t)}\right\|^2 < \infty$ with probability 1.

## F.2 DIRECTIONAL CONVERGENCE ANALYSIS (PROOF OF THEOREM 4.2)

In this section, we prove Theorem 4.2 and further discuss the convergence of $\boldsymbol{\rho}_k^{(t)}$ beyond boundedness.

**Theorem 4.2.** *Let $\{\boldsymbol{w}_k^{(t)}\}_{k\in[0:K-1],t\geq 0}$ be the sequence of GD iterates (2) from any starting point $\boldsymbol{w}_0^{(0)}$, where tasks are given randomly. Suppose that Assumptions 3.1, 3.2, 3.3, and 3.4 hold. Then, under the same learning rate condition as in Theorem 4.1, $\boldsymbol{w}_k^{(t)}$ will behave as:*

$$\boldsymbol{w}_k^{(t)} = \ln(t)\,\hat{\boldsymbol{w}} + \boldsymbol{\rho}_k^{(t)},$$

*where $\|\boldsymbol{\rho}_k^{(t)}\|$ stays bounded as $t$ grows.*

We only need to prove that the two following lemmas still hold in random order.

**Lemma F.1.** *When tasks are given randomly, there exists $\check{\boldsymbol{w}}, \boldsymbol{m}_{t,k} \in \mathbb{R}^d$ the following almost surely holds for all $t \in \mathbb{N}$, $k \in [0:K-1]$:*

$$K\sum_{u=1}^{t-1}\frac{1}{u}\sum_{s\in S^{(u)}}\alpha_s \boldsymbol{x}_s + \frac{k}{t}\sum_{s\in S^{(t)}}\alpha_s \boldsymbol{x}_s = \frac{K}{M}\log(\frac{t}{M})\hat{\boldsymbol{w}} + \frac{K}{M}\check{\boldsymbol{w}} + \boldsymbol{m}_{t,k}, \qquad (50)$$

$$\boldsymbol{m}_{t,K} := \boldsymbol{m}_{t+1,0},$$

*such that $\|\boldsymbol{m}_{t,k}\| = o(t^{-0.5+\epsilon})$, and $\|\boldsymbol{m}_{t,k+1} - \boldsymbol{m}_{t,k}\| = \mathcal{O}(t^{-1})$ for all $k \in [0:K-1], \epsilon > 0$, and $\check{\boldsymbol{w}}$ only depends on the order of tasks and constant with respect to $t$.*

*Proof.* See Appendix F.2.1. $\qquad\square$

Using Lemma F.1, we set $\boldsymbol{m}_{t,k}$ and $\check{\boldsymbol{w}}$ and define $\boldsymbol{\rho}_k^{(t)}$ and $\boldsymbol{r}_k^{(t)}$ as we did in cyclic order. That is,

$$\boldsymbol{w}_k^{(t)} = \ln(t)\,\hat{\boldsymbol{w}} + \boldsymbol{\rho}_k^{(t)}$$

$$= \ln(t)\,\hat{\boldsymbol{w}} + \ln\left(\frac{K}{M}\right)\hat{\boldsymbol{w}} + \tilde{\boldsymbol{w}} + \frac{M}{K}\boldsymbol{m}_{t,k} + \boldsymbol{r}_k^{(t)},$$

and

$$\boldsymbol{\rho}_K^{(t)} = \boldsymbol{\rho}_0^{(t+1)}, \quad \boldsymbol{r}_K^{(t)} = \boldsymbol{r}_0^{(t+1)},$$

where $\tilde{w}$ is the solution of

$$\forall i \in S : \eta \exp\left(-\boldsymbol{x}_i^\top \tilde{\boldsymbol{w}}\right) = \alpha_i, \quad \bar{P}(\tilde{\boldsymbol{w}} - \boldsymbol{w}_0^{(0)}) = 0.$$

which is unique under Assumption 3.2. Then by the definition,

$$\boldsymbol{r}_k^{(t)} = \boldsymbol{w}_k^{(t)} - \frac{M}{K}\left(\frac{K}{M}\log\left(\frac{K}{M}t\right)\hat{\boldsymbol{w}} + \boldsymbol{m}_{t,k}\right) - \tilde{\boldsymbol{w}}$$

$$= \boldsymbol{w}_k^{(t)} - \frac{M}{K}\left(K\sum_{u=1}^{t-1}\frac{1}{u}\sum_{s\in S^{(u)}}\alpha_s\boldsymbol{x}_s + \frac{k}{t}\sum_{s\in S^{(t)}}\alpha_s\boldsymbol{x}_s\right) - \log K\hat{\boldsymbol{w}} - \tilde{\boldsymbol{w}} + \breve{\boldsymbol{w}}.$$

Then we can get the second primary lemma of $\boldsymbol{r}_k^{(t)}$.

**Lemma F.2.** *Under Assumption 3.1, 3.2, 3.3, and 3.4, if learning rate is $\eta < \frac{2\phi^2}{\beta\sigma_{\max}^4}$, then*

1. $\exists \tilde{t}, C_1, C_2 > 0$ *such that* $\forall t > \tilde{t}$,

$$(\boldsymbol{r}_{k+1}^{(t)} - \boldsymbol{r}_k^{(t)})^\top \boldsymbol{r}_k^{(t)} \le C_1 t^{-\theta} + C_2 t^{-1-0.5\tilde{\mu}}, \forall k \in [0:K-1].$$

2. *Moreover, for all $\epsilon_1 > 0$, $\exists \tilde{t}^*, C_3 > 0$ such that if $\left\|P\boldsymbol{r}_k^{(t)}\right\| \ge \epsilon_1$ and $S^{(t)} \ne \emptyset$,*

$$(\boldsymbol{r}_{k+1}^{(t)} - \boldsymbol{r}_k^{(t)})^\top \boldsymbol{r}_k^{(t)} \le -C_3 t^{-1}, \forall t > \tilde{t}^*, k \in [0:K-1].$$

*Proof.* Only the learning rate is different from the cyclic case. Therefore see Appendix E.2.2. □

The remaining step is the same as the proof of Theorem 3.2. To sum up, we can set $\boldsymbol{a}_k^{(t)}$ as $\left\|\boldsymbol{r}_{k+1}^{(t)} - \boldsymbol{r}_k^{(t)}\right\|^2 = \left\|\boldsymbol{w}_{k+1}^{(t)} - \boldsymbol{w}_k^{(t)} - \boldsymbol{a}_k^{(t)}\right\|^2$. Then by Lemma F.1, $\exists t_1$ such that $\forall t \ge t_1, \forall k \in [0:K-1] : \left\|\boldsymbol{a}_k^{(t)}\right\| \le t^{-1}$.

For all $T \ge t_1$.

$$\sum_{t=t_1}^{T}\sum_{k=0}^{K-1}\left\|\boldsymbol{r}_{k+1}^{(t)} - \boldsymbol{r}_k^{(t)}\right\|^2 = \sum_{t=t_1}^{T}\sum_{k=0}^{K-1}\left\|\boldsymbol{w}_{k+1}^{(t)} - \boldsymbol{w}_k^{(t)} - \boldsymbol{a}_k^{(t)}\right\|^2$$

$$= \sum_{t=t_1}^{T}\sum_{k=0}^{K-1}\left\|\boldsymbol{w}_{k+1}^{(t)} - \boldsymbol{w}_k^{(t)}\right\|^2 + \sum_{t=t_1}^{T}\sum_{k=0}^{K-1}2(\boldsymbol{w}_k^{(t)} - \boldsymbol{w}_{k+1}^{(t)})^\top\boldsymbol{a}_k^{(t)} + \sum_{t=t_1}^{T}\sum_{k=0}^{K-1}\left\|\boldsymbol{a}_k^{(t)}\right\|^2$$

$$\le \sum_{t=t_1}^{T}\sum_{k=0}^{K-1}\left\|\boldsymbol{w}_{k+1}^{(t)} - \boldsymbol{w}_k^{(t)}\right\|^2 + 2\sqrt{\sum_{t=t_1}^{T}\sum_{k=0}^{K-1}\left\|\boldsymbol{w}_k^{(t)} - \boldsymbol{w}_{k+1}^{(t)}\right\|^2\sum_{t=t_1}^{T}\sum_{k=0}^{K-1}\left\|\boldsymbol{a}_k^{(t)}\right\|^2} + \sum_{t=t_1}^{T}\sum_{k=0}^{K-1}\left\|\boldsymbol{a}_k^{(t)}\right\|^2$$

$$\le \sum_{t=t_1}^{T}\sum_{k=0}^{K-1}\left\|\boldsymbol{w}_{k+1}^{(t)} - \boldsymbol{w}_k^{(t)}\right\|^2 + 2\sqrt{\sum_{t=t_1}^{T}\sum_{k=0}^{K-1}\left\|\boldsymbol{w}_k^{(t)} - \boldsymbol{w}_{k+1}^{(t)}\right\|^2\sum_{t=t_1}^{T}\sum_{k=0}^{K-1}t^{-2}} + \sum_{t=t_1}^{T}\sum_{k=0}^{K-1}t^{-2}$$

$$< \infty. \tag{51}$$

We use Cauchy-Schwarz inequality for the first inequality and the fact that $\sum_{t=t_1}^{T}t^{-2} < \infty$ and $\sum_{t=t_1}^{T}\sum_{k=0}^{K-1}\left\|\boldsymbol{w}_k^{(t)} - \boldsymbol{w}_{k+1}^{(t)}\right\|^2 < \infty$ by Theorem 4.1.

Combined with Lemma F.2 and the fact that $\forall c > 1 : \sum_{t=1}^{\infty}t^{-c} < \infty$, we almost surely get

$$\left\|\boldsymbol{r}_0^{(t)}\right\|^2 - \left\|\boldsymbol{r}_0^{(t_1)}\right\|^2 = \sum_{u=t_1}^{t-1}\sum_{k=0}^{K-1}\left(\left\|\boldsymbol{r}_{k+1}^{(u)}\right\|^2 - \left\|\boldsymbol{r}_k^{(u)}\right\|^2\right)$$

$$= \sum_{u=t_1}^{t-1}\sum_{k=0}^{K-1}\left(2(\boldsymbol{r}_{k+1}^{(u)} - \boldsymbol{r}_k^{(u)})^\top\boldsymbol{r}_k^{(u)} + \left\|\boldsymbol{r}_{k+1}^{(u)} - \boldsymbol{r}_k^{(u)}\right\|^2\right) < \infty.$$

### F.2.1 PROOF OF LEMMA F.1

Here we restate the lemma for the sake of readability.

**Lemma F.1.** *When tasks are given randomly, there exists $\check{w}, m_{t,k} \in \mathbb{R}^d$ the following almost surely holds for all $t \in \mathbb{N}$, $k \in [0 : K - 1]$:*

$$K \sum_{u=1}^{t-1} \frac{1}{u} \sum_{s \in S^{(u)}} \alpha_s x_s + \frac{k}{t} \sum_{s \in S^{(t)}} \alpha_s x_s = \frac{K}{M} \log(\frac{t}{M}) \hat{w} + \frac{K}{M} \check{w} + m_{t,k}, \tag{50}$$

$$m_{t,K} := m_{t+1,0},$$

*such that $\|m_{t,k}\| = o(t^{-0.5+\epsilon})$, and $\|m_{t,k+1} - m_{t,k}\| = \mathcal{O}(t^{-1})$ for all $k \in [0 : K - 1], \epsilon > 0$, and $\check{w}$ only depends on the order of tasks and constant with respect to $t$.*

We define an (i.i.d.) random variable(s) $z_i^{(t)} := \mathbb{1}\{x_i \in I^{(t)}\}$. Note that $\mathbb{E}[z_i^{(t)}] = \frac{1}{M}$ and $\mathrm{Var}(z_i^{(t)}) = \frac{M-1}{M^2}$ due to uniform sampling of the task index in $[0 : M - 1]$. Then, we can write a sum on the right-hand side of Equation (50) as follows:

$$K \sum_{u=1}^{t-1} \frac{1}{u} \sum_{s \in S^{(u)}} \alpha_s x_s = K \sum_{s \in S} \left( \sum_{u=1}^{t-1} \frac{z_s^{(u)}}{u} \right) \alpha_s x_s$$

$$= K \sum_{s \in S} \left( \sum_{u=1}^{t-1} \frac{\mathbb{E}[z_s^{(u)}]}{u} + \sum_{u=1}^{t-1} \frac{z_s^{(u)} - \mathbb{E}[z_s^{(u)}]}{u} \right) \alpha_s x_s$$

$$= K \sum_{s \in S} \left( \frac{1}{M} \sum_{u=1}^{t-1} \frac{1}{u} + . \sum_{u=1}^{t-1} \frac{z_s^{(u)} - \mathbb{E}[z_s^{(u)}]}{u} \right) \alpha_s x_s.$$

Since

$$\sum_{u=1}^{t-1} \frac{1}{u} = \log t + \gamma + q(t)$$

where $\gamma$ is the Euler-Mascheroni constant and $q(t) = \mathcal{O}(t^{-1})$, we have

$$K \sum_{s \in S} \left( \frac{1}{M} \sum_{u=1}^{t-1} \frac{1}{u} \right) \alpha_s x_s = \frac{K}{M} \left( \log t + \gamma + q(t) \right) \hat{w}.$$

Now we are going to deal with the sum

$$\sum_{u=1}^{t-1} \frac{z_s^{(u)} - \mathbb{E}[z_s^{(u)}]}{u}$$

in two aspects: (1) it converges with probability 1 as $t \to \infty$ and (2) the almost-sure vanishing rate of the "residual" (a sum from $u = t$ to $\infty$) is $o(t^{-0.5+\epsilon})$ for any $\epsilon > 0$. Let us look at its almost-sure convergence. To this end, we utilize the following useful proposition.

**Proposition F.3** (Theorem 5.2.6 of Durrett (2019)). *Suppose $X_1, X_2, \dots$ are zero-mean independent random variables. If $\sum_{n=1}^{\infty} \mathrm{Var}(X_n) < \infty$, then $\sum_{n=1}^{\infty} X_n$ converges almost surely (i.e., with probability 1).*

Observe that $X_u := \frac{z_s^{(u)} - \mathbb{E}[z_s^{(u)}]}{u}$ is a zero-mean random variables. Not only are they independent for all $u$, but also the sum of their variances is convergent:

$$\sum_{u=1}^{\infty} \mathrm{Var}(X_u) = \frac{M-1}{M^2} \sum_{u=1}^{\infty} \frac{1}{u^2} < \infty.$$

Thus, by Proposition F.3, the sum $\sum_{u=1}^{\infty} X_u$ converges with probability 1. Next, we want to show the vanishing rate

$$\sum_{u=t}^{\infty} X_u = o(t^{-0.5+\epsilon})$$

with probability 1, where we choose any $\epsilon > 0$. Observe that it is equivalent to show, for any $\delta > 0$,

$$\mathbb{P}\left(t^{0.5-\epsilon} \cdot \left|\sum_{u=t}^{\infty} X_u\right| > \delta \text{ for infinitely many } t\right) = 0.$$

Here we bring a renowned Borel-Cantelli Lemma.

**Proposition F.4** (Borel-Cantelli lemma; Theorem 2.3.1 of Durrett (2019)). *Consider a sequence of events* $A_1, A_2, \cdots$. *If* $\sum_{n=1}^{\infty} \mathbb{P}(A_n) < \infty$, *then*

$$\mathbb{P}(\limsup_{n\to\infty} A_n) := \mathbb{P}(A_n \text{ happens for infinitely many } n) = 0.$$

By Proposition F.4, it suffices to show

$$\forall \delta > 0, \quad \sum_{t=1}^{\infty} \mathbb{P}\left(t^{0.5-\epsilon} \cdot \left|\sum_{u=t}^{\infty} X_u\right| > \delta\right) < \infty.$$

Let us recall Hoeffding inequality here:

**Proposition F.5** (Hoeffding inequality). *Consider a collection of independent random variables* $X_1, \cdots, X_n$ *satisfying* $a_i \le X_i \le b_i$ *for each* $i = 1, \cdots, n$ $(a_i < b_i)$. *Then,*

$$\mathbb{P}\left(\left|\sum_{i=1}^{n} X_i\right| \ge r\right) \le 2\exp\left(-\frac{2r^2}{\sum_{i=1}^{n}(b_i - a_i)^2}\right).$$

Since the sum $\sum_{u=t}^{\infty} X_u$ converges almost surely, it is a well-defined random variable with probability 1, and

$$\mathbb{P}\left(\left|\sum_{u=t}^{\infty} X_u\right| > \delta \cdot t^{-0.5+\epsilon}\right) = \mathbb{P}\left(\left|\sum_{u=t}^{T} X_u\right| > \delta \cdot t^{-0.5+\epsilon} \text{ for all but finitely many } T\right)$$

$$=: \mathbb{P}\left(\liminf_{T\to\infty}\left\{\left|\sum_{u=t}^{T} X_u\right| > \delta \cdot t^{-0.5+\epsilon}\right\}\right)$$

$$\le \liminf_{T\to\infty} \mathbb{P}\left(\left|\sum_{u=t}^{T} X_u\right| > \delta \cdot t^{-0.5+\epsilon}\right) \tag{52}$$

$$\le \liminf_{T\to\infty} 2\exp\left(-\frac{2\delta^2 t^{-1+2\epsilon}}{\sum_{u=t}^{T} \frac{1}{u^2}}\right) \tag{53}$$

$$= 2\exp\left(-\frac{2\delta^2 t^{-1+2\epsilon}}{\sum_{u=t}^{\infty} \frac{1}{u^2}}\right) \tag{54}$$

$$\le 2\exp\left(-\delta^2 t^{2\epsilon}\right). \tag{55}$$

We use the fact "$\mathbb{P}(\liminf_n A_n) \le \liminf_n \mathbb{P}(A_n)$" in Equation (52); we apply Hoeffding inequality (Proposition F.5) and the fact $-\frac{1}{Mu} \le X_u \le \frac{M-1}{Mu}$ in Equation (53); and we utilize the fact $\sum_{u=t}^{\infty} \frac{1}{u^2} \le \frac{2}{t}$ in Equation (55). Since $\exp(-\delta^2 t^{2\epsilon}) = o(t^{-2})$ for any $\epsilon > 0$ and large enough $t$, the sum $\sum_t \exp(-\delta^2 t^{2\epsilon})$ converges. Therefore, we have desired almost-sure convergence guarantees.

From now on, let us proceed with the proof. Using the almost-sure convergence results, let

$$\check{\boldsymbol{w}} := (\log M + \gamma)\hat{\boldsymbol{w}} + M\sum_{s\in S}\left(\sum_{u=1}^{\infty} X_u\right)\alpha_s \boldsymbol{x}_s,$$

$$\boldsymbol{m}_{t,k} := \frac{K}{M} q(t)\hat{\boldsymbol{w}} + K \sum_{s \in S} \left( \sum_{u=t}^{\infty} X_u \right) \alpha_s \boldsymbol{x}_s + \frac{k}{t} \sum_{s \in S^{(t)}} \alpha_s \boldsymbol{x}_s.$$

Then with probability 1, the statement of the lemma holds: Equation (50) holds, where $\check{\boldsymbol{w}}$ is a constant vector in terms of $t$, $\|\boldsymbol{m}_{t,k}\| \leq o(t^{-0.5+\epsilon})$ for any $\epsilon > 0$, and

$$\|\boldsymbol{m}_{t,k+1} - \boldsymbol{m}_{t,k}\| = \mathcal{O}(t^{-1}), \quad (k = 0, ..., K-2)$$
$$\|\boldsymbol{m}_{t+1,0} - \boldsymbol{m}_{t,K-1}\| = \mathcal{O}(t^{-1}).$$

This concludes the proof of the lemma.

### F.2.2 Convergence of $\rho_k^{(t)}$

We can also prove a characterization of the limit of $\boldsymbol{\rho}_k^{(t)}$, as done in Appendix E.2.3. However, when tasks are given randomly, we need an additional assumption to guarantee the convergence of $\boldsymbol{\rho}_k^{(t)}$ to the particular point.

**Assumption F.1.** Every task has at least one support vector. That is, $\forall m \in [0 : M-1] : S_m \neq \emptyset$.

**Proposition F.6.** *Under the same setting of Theorem 4.2 with additional Assumptions E.1 and F.1, the "residual" converges to $\lim_{t\to\infty} \boldsymbol{\rho}_k^{(t)} = \tilde{\boldsymbol{w}}, \forall k \in [0 : K-1]$. Here, $\tilde{\boldsymbol{w}}$ is the vector defined in Proposition E.6.*

*Proof.* First, $\bar{P}\boldsymbol{r}_k^{(t)} = \bar{P}\boldsymbol{w}_0^{(0)} - \bar{P}\tilde{\boldsymbol{w}} = 0$ holds as in cyclic case. See Appendix E.2.3.

Second, we get to show $P\boldsymbol{r}_k^{(t)} \to 0$. By Equation (51), $\lim_{T\to\infty} \sum_{t=t_1}^{T} \sum_{k=0}^{K-1} \left\| \boldsymbol{r}_{k+1}^{(t)} - \boldsymbol{r}_k^{(t)} \right\|^2 = C_4$. That means $\forall k \in [0 : K-1] : \lim_{T\to\infty} \left\| \boldsymbol{r}_{k+1}^{(T)} - \boldsymbol{r}_k^{(T)} \right\| = 0$. Therefore, for any $\epsilon_0$, there exists $t_2 > 0$ such that $\left\| \boldsymbol{r}_{k+1}^{(t)} - \boldsymbol{r}_k^{(t)} \right\| < \frac{\epsilon_0}{K}$ for all $t \geq t_2, k \in [0 : K-1]$. As a result,

$$\left\| P\boldsymbol{r}_0^{(t)} \right\| + \frac{k}{K}\epsilon_0 \geq \left\| P\boldsymbol{r}_k^{(t)} \right\| \geq \left\| P\boldsymbol{r}_0^{(t)} \right\| - \frac{k}{K}\epsilon_0.$$

For $t \geq \max\{t_1, t_2, \tilde{t}^*\}$, if $\left\| P\boldsymbol{r}_0^{(t)} \right\| \geq \epsilon_1 + \epsilon_0$ and $S^{(t)} \neq \emptyset$, then $\forall k \in [0 : K-1] : \left\| P\boldsymbol{r}_k^{(t)} \right\| \geq \epsilon_1$. By Lemma F.2 (2),

$$\sum_{u=t-1}^{t} \sum_{v=0}^{K-1} (\boldsymbol{r}_{v+1}^{(u)} - \boldsymbol{r}_v^{(u)})^\top \boldsymbol{r}_v^{(u)} \leq -KC_3 t^{-1} + K\left( C_1 t^{-\theta} + C_2 t^{-1-0.5\tilde{\mu}} \right),$$

Since $t^{-1}$ decrease to zero slower than $t^{-\theta}$ and $t^{-1-0.5\tilde{\mu}}$, there exists $t_3 > \max\{t_1, t_2, \tilde{t}^*\}, C_4 > 0$ such that $-KC_3 t^{-1} + K\left( C_1 t^{-\theta} + C_2 t^{-1-0.5\tilde{\mu}} \right) \leq -C_5 t^{-1}$. Also $S^{(t)} \neq \emptyset$ is given by Assumption F.1. To sum up, for any $\epsilon_0, \epsilon_2 > 0$, there exists $t_3 > \max\{t_1, t_2, \tilde{t}^*\}$ such that if $\left\| P\boldsymbol{r}_0^{(t)} \right\| \geq \epsilon_0 + \epsilon_1$, then

$$\sum_{u=t-1}^{t} \sum_{v=0}^{K-1} (\boldsymbol{r}_{v+1}^{(u)} - \boldsymbol{r}_v^{(u)})^\top \boldsymbol{r}_v^{(u)} \leq -C_5 t^{-1},$$

Now, define two sets for each $k \in [0 : K-1]$

$$\mathcal{T}_k := \{t > t_3 : \left\| P\boldsymbol{r}_k^{(t)} \right\| < \epsilon_0 + \epsilon_1\}$$
$$\bar{\mathcal{T}}_k := \{t > t_3 : \left\| P\boldsymbol{r}_k^{(t)} \right\| \geq \epsilon_0 + \epsilon_1\}$$

We will finish our proof by showing that $\bar{\mathcal{T}}_k$ is finite. Here, we use the fact that every $\mathcal{T}_k$ is infinite. The proof is the same as in the cyclic case. Since $\lim_{T\to\infty} \sum_{t=t_1}^{T} \sum_{k=0}^{K-1} \left\| r_{k+1}^{(t)} - r_k^{(t)} \right\|^2 = C_4$, we get

$$\sum_{u=t_1}^{t} \sum_{k=0}^{K-1} \left\| r_{k+1}^{(u)} - r_k^{(u)} \right\|^2 = C_4 - h(t),$$

where $h(t)$ is a positive function monotonic decreasing to zero.

Now, assume that there exists some $k'$ that $\bar{\mathcal{T}}_k$ is infinite. WLOG, we set $k' = 0$. Since $\mathcal{T}_0$ is infinite, for any $t \in \bar{\mathcal{T}}_0$ there exists $t', t'' \in \mathcal{T}_0$ such that $t \in [t'+1, t''-1] \subset \bar{\mathcal{T}}_0$. We divide it into two cases: For all $t \in [t'+1, t''-1]$,

1. if $t = t'+1$, then $\left\| Pr_0^{(t)} \right\|^2 \le \left\| Pr_0^{(t')} \right\|^2 + \epsilon_0 \le 2\epsilon_0 + \epsilon_1$.

2. if $t \ge t'+1$, then

$$\left\| Pr_0^{(t)} \right\|^2 = \left\| Pr_0^{(t')} \right\|^2 + \sum_{u=t'}^{t-1} \sum_{k=0}^{K-1} \left[ \left\| r_{k+1}^{(u)} \right\|^2 - \left\| r_k^{(u)} \right\|^2 \right]$$

$$= \left\| Pr_0^{(t')} \right\|^2 + \sum_{u=t'}^{t-1} \sum_{k=0}^{K-1} \left[ \left\| r_{k+1}^{(u)} - r_k^{(u)} \right\|^2 + 2(r_{k+1}^{(u)} - r_k^{(u)})^\top r_k^{(u)} \right]$$

$$= \left\| Pr_0^{(t')} \right\|^2 + h(t) - h(t') + 2 \sum_{u=t'}^{t-1} \sum_{k=0}^{K-1} \left[ (r_{k+1}^{(u)} - r_k^{(u)})^\top r_k^{(u)} \right]$$

$$\le (\epsilon_0 + \epsilon_1)^2 + h(t) - 2C_5 \frac{1}{t'+1} - 2C_3 \sum_{u=t'+2}^{t-1} \frac{1}{u}$$

$$\le (\epsilon_0 + \epsilon_1)^2 + h(t).$$

Since $h(t)$ is monotonic decreasing function, for any $\epsilon_2 > 0$, there exists $t_4$ such that $\forall t \ge t_4 : h(t) < \epsilon_2$.

Therefore, $\forall t \ge \max\{t_3, t_4\} : \left\| Pr_0^{(t)} \right\|^2 \le (\epsilon_0 + \epsilon_1)^2 + \epsilon_2$. Since it holds for any $\epsilon_0, \epsilon_1, \epsilon_2$, it contradicts with the assumption that $\bar{\mathcal{T}}_0$ is infinite. $\qquad \square$

## G  PROOFS FOR SECTION 5: CYCLIC TASK ORDERING, JOINTLY NON-SEPARABLE

**Review on Bregman Divergence.**  Before we start the proofs, we briefly overview some basic properties of *Bregman divergence*.

Given a differentiable convex function $f : \mathcal{S} \to \mathbb{R}$ defined on a closed convex set $\mathcal{S} \subset \mathbb{R}^d$, the Bregman divergence between two points $(\boldsymbol{x}, \boldsymbol{y}) \in \mathcal{S} \times \operatorname{int} \mathcal{S}$ with respect to $f$ is defined as

$$D_f(\boldsymbol{x}, \boldsymbol{y}) := f(\boldsymbol{x}) - f(\boldsymbol{y}) - \langle \nabla f(\boldsymbol{y}), \boldsymbol{x} - \boldsymbol{y} \rangle.$$

Note that $D_f(\boldsymbol{x}, \boldsymbol{y}) \geq 0$ for any $\boldsymbol{x}, \boldsymbol{y} \in \mathcal{S}$ because of the definition of convexity; when $f$ is strictly convex, $D_f(\boldsymbol{x}, \boldsymbol{y}) = 0$ if and only if $\boldsymbol{x} = \boldsymbol{y}$. Also, if $f$ is $\beta$-smooth, $D_f(\boldsymbol{x}, \boldsymbol{y}) \leq \frac{\beta}{2} \|\boldsymbol{x} - \boldsymbol{y}\|^2$ holds. We often use the following useful identity that links three different points $\boldsymbol{x}, \boldsymbol{y} \in \mathcal{S}$ and $\boldsymbol{z} \in \operatorname{int} \mathcal{S}$:

$$\langle \nabla f(\boldsymbol{z}), \boldsymbol{x} - \boldsymbol{y} \rangle = [f(\boldsymbol{x}) - f(\boldsymbol{y})] - [D_f(\boldsymbol{x}, \boldsymbol{z}) - D_f(\boldsymbol{y}, \boldsymbol{z})]. \tag{56}$$

Here is another useful fact: for a convex $\beta$-smooth function $f$ on $\mathbb{R}^d$, the Bregman divergence is bound below by the squared distance between gradients.

**Proposition G.1.** *Let $f : \mathbb{R}^d \to \mathbb{R}$ be a convex, $\beta$-smooth function. For any $\boldsymbol{x}, \boldsymbol{y} \in \mathbb{R}^d$,*

$$\|\nabla f(\boldsymbol{x}) - \nabla f(\boldsymbol{y})\|^2 \leq 2\beta D_f(\boldsymbol{x}, \boldsymbol{y}).$$

*Proof.* Observe that $D_f(\cdot, \boldsymbol{y})$ is also a $\beta$-smooth function for any $\boldsymbol{y}$. Let $\boldsymbol{z} = \boldsymbol{x} - \frac{1}{\beta} \nabla_{\boldsymbol{x}} D_f(\boldsymbol{x}, \boldsymbol{y}) = \boldsymbol{x} - \frac{1}{\beta}[\nabla f(\boldsymbol{x}) - \nabla(\boldsymbol{y})] \in \mathbb{R}^d$. Then by $\beta$-smoothness and the non-negativity of $D_f(\cdot, \boldsymbol{y})$, we have

$$0 \leq D_f(\boldsymbol{z}, \boldsymbol{y})$$
$$\leq D_f(\boldsymbol{x}, \boldsymbol{y}) + \langle \nabla_{\boldsymbol{x}} D_f(\boldsymbol{x}, \boldsymbol{y}), \boldsymbol{z} - \boldsymbol{x} \rangle + \frac{\beta}{2} \|\boldsymbol{z} - \boldsymbol{x}\|^2$$
$$= D_f(\boldsymbol{x}, \boldsymbol{y}) - \frac{1}{\beta} \langle \nabla f(\boldsymbol{x}) - \nabla(\boldsymbol{y}), \nabla f(\boldsymbol{x}) - \nabla(\boldsymbol{y}) \rangle + \frac{1}{2\beta} \|\nabla f(\boldsymbol{x}) - \nabla(\boldsymbol{y})\|^2$$
$$= D_f(\boldsymbol{x}, \boldsymbol{y}) - \frac{1}{2\beta} \|\nabla f(\boldsymbol{x}) - \nabla f(\boldsymbol{y})\|^2.$$

This proves the proposition. $\qquad\square$

**Useful Inequalities.**  There are two other crucial inequalities for the proofs in this appendix. One is a variant of Jensen's inequality applied to a squared norm.

**Proposition G.2.** *For any positive numbers $\lambda_1, \cdots, \lambda_n > 0$, any vectors $\boldsymbol{u}_1, \cdots, \boldsymbol{u}_n \in \mathbb{R}^d$, and an integer $m \in [0 : n]$,*

$$\left\| \sum_{i=1}^{m} \boldsymbol{u}_i \right\|^2 \leq \left( \sum_{i=1}^{n} \lambda_i \right) \left( \sum_{i=1}^{n} \frac{1}{\lambda_i} \|\boldsymbol{u}_i\|^2 \right).$$

*Proof.* Let $\Lambda_m = \sum_{i=1}^{m} \lambda_i$. Then by convexity of the squared norm,

$$\left\| \sum_{i=1}^{m} \boldsymbol{u}_i \right\|^2 = \left\| \sum_{i=1}^{m} \frac{\lambda_i}{\Lambda_m} \left( \frac{\Lambda_m}{\lambda_i} \boldsymbol{u}_i \right) \right\|^2$$
$$\leq \sum_{i=1}^{m} \frac{\lambda_i}{\Lambda_m} \left\| \frac{\Lambda_m}{\lambda_i} \boldsymbol{u}_i \right\|^2$$
$$= \Lambda_m \sum_{i=1}^{m} \frac{1}{\lambda_i} \|\boldsymbol{u}_i\|^2$$
$$\leq \Lambda_n \sum_{i=1}^{n} \frac{1}{\lambda_i} \|\boldsymbol{u}_i\|^2.$$

$\qquad\square$

Another is about solving a recurrent inequality.

**Proposition G.3.** *Consider $0 < \mu \leq \beta$, $V > 0$, $T > 1$, $0 < c = \Theta(1)$, $0 < m = \Theta(1)$, and $\Delta_0 \geq 0$. Suppose the following inequality holds for any positive $\alpha \leq \frac{c}{\beta}$ and $t \in [0 : T - 1]$:*

$$\Delta_{t+1} \leq \frac{1}{1 + \alpha\mu}\Delta_t + \alpha^{m+1}V.$$

*If we take*

$$\alpha = \min\left\{\frac{c}{\beta},\ \frac{c+1}{\mu T}\ln\left(T^m \cdot \max\left\{1, \frac{\Delta_0\mu^{m+1}}{V}\right\}\right)\right\},$$

*we have*

$$\Delta_T = \mathcal{O}\left(\exp\left(-\frac{c\mu}{(c+1)\beta}T\right)\Delta_0 + \frac{V \cdot \ln^m(T)}{\mu^{m+1}T^m}\right).$$

*Proof.* Since $\alpha\mu \leq \frac{c\mu}{\beta} \leq c$, we have $\frac{1}{1+\alpha\mu} \leq 1 - \frac{\alpha\mu}{c+1}$. By unrolling the recurrent inequality, we have

$$\Delta_T \leq \left(1 - \frac{\alpha\mu}{c+1}\right)^T \Delta_0 + \alpha^{m+1}V \sum_{t=0}^{T-1}\left(1 - \frac{\alpha\mu}{c+1}\right)^t$$

$$\leq \exp\left(-\frac{\alpha\mu}{c+1}T\right)\Delta_0 + \frac{2\alpha^m V}{\mu}.$$

With the choice of $\alpha$, the first exponential term is bounded as

$$\exp\left(-\frac{\alpha\mu}{c+1}T\right)\Delta_0 \leq \max\left\{\exp\left(-\frac{c\mu}{(c+1)\beta}T\right)\Delta_0,\ \frac{V}{\mu^{m+1}T^m}\right\}$$

$$\leq \exp\left(-\frac{c\mu}{(c+1)\beta}T\right)\Delta_0 + \frac{V}{\mu^{m+1}T^m}.$$

Also, the second term is bounded as

$$\frac{2\alpha^m V}{\mu} \leq \frac{2(c+1)^m V}{\mu^{m+1}T^m}\ln^m\left(T^m \cdot \max\left\{1, \frac{\Delta_0\mu^{m+1}}{V}\right\}\right).$$

Combining these two and ignoring some constant factors and polylogarithmic terms, we have the desired bound. □

## G.1 LOCAL STRONG CONVEXITY ANALYSIS (PROOF OF LEMMA 5.1)

Recall that we consider cyclic continual learning on $M$ jointly strictly non-separable classification tasks. Let us restate the lemma here for the sake of readability.

**Lemma 5.1.** *Consider learning $M$ linear classification tasks cyclically. Suppose that Assumptions 5.1 and 5.2 hold. Let $B := \sum_{m=0}^{M-1}\beta_m$ and $V_\star := \sum_{m=0}^{M-1}\frac{1}{\beta_m}\|\nabla\mathcal{L}_m(\boldsymbol{w}_\star)\|^2$. Take a step size $\eta \leq \frac{1}{2\sqrt{2}KB}$. Then, there exists a compact set $\mathcal{W} \subset \mathbb{R}^d$ containing $\boldsymbol{w}_\star$ and every $\boldsymbol{w}_0^{(jM)}$ $(j = 0, 1, 2, \ldots)$, whose radius is independent of $J$ (the number of cycles) but depends on other parameters like $b$, $G$, $B$, and $V_\star$. Also, the offline training loss $\mathcal{L}$ is $\mu$-strongly convex on $\mathcal{W}$, where*

$$\mu := \left(\min_{i\in[0:N-1], \boldsymbol{w}\in\mathcal{W}}\ell''\left(y_i\boldsymbol{x}_i^\top\boldsymbol{w}\right)\right) \cdot \lambda_{\min}\left(\boldsymbol{X}\boldsymbol{X}^\top\right) > 0. \tag{6}$$

To prove the boundedness of end-of-cycle iterates and the local strong convexity, we first establish a per-cycle recurrent inequality in terms of squared distance to an arbitrary comparator $\boldsymbol{u} \in \mathbb{R}^d$ and the risk values. We put $\boldsymbol{u} = \boldsymbol{w}_\star$ later.

**Lemma G.4** (Backward recurrent inequality). *Consider learning $M$ linear classification tasks cyclically. Suppose that Assumption 5.2 holds. Let $B = \sum_{m=0}^{M-1}\beta_m$. If we take any step size $\eta \leq \frac{1}{2\sqrt{2}KB}$, the iterates of sequential GD satisfies*

$$\left\|\boldsymbol{w}_0^{((j+1)M)} - \boldsymbol{u}\right\|^2$$

$$\leq \left\| \boldsymbol{w}_0^{(jM)} - \boldsymbol{u} \right\|^2 - 2\eta K \left[ \mathcal{L}\left(\boldsymbol{w}_0^{(jM)}\right) - \mathcal{L}(\boldsymbol{u}) \right] + 2\sqrt{2}\eta^2 K^2 B \left( \sum_{m=0}^{M-1} \frac{1}{\beta_m} \|\nabla \mathcal{L}_m(\boldsymbol{u})\|^2 \right),$$

*for any vector $\boldsymbol{u} \in \mathbb{R}^d$ and for all $j = 0, 1, \cdots$.*

*Proof.* We defer the proof to Appendix G.1.1. We remark that this lemma holds even without assuming the non-separability. □

Observe that the following holds as a special case:

$$\left\| \boldsymbol{w}_0^{((j+1)M)} - \boldsymbol{w}_\star \right\|^2 \leq \left\| \boldsymbol{w}_0^{(jM)} - \boldsymbol{w}_\star \right\|^2 - 2\eta K \left[ \mathcal{L}\left(\boldsymbol{w}_0^{(jM)}\right) - \mathcal{L}(\boldsymbol{w}_\star) \right] + 2\sqrt{2}\eta^2 K^2 B V_\star,$$
(57)

where $V_\star = \sum_{m=0}^{M-1} \frac{1}{\beta_m} \|\nabla \mathcal{L}_m(\boldsymbol{w}_\star)\|^2$.

The next step is to construct a compact ball $\mathcal{W}$ centered at $\boldsymbol{w}_\star$, containing every end-of-cycle iterate of sequential GD. The crucial step is to apply the non-separability coefficient $b > 0$ (Assumption 5.1).

**Lemma G.5** (Boundedness of the end-of-cycle iterates). *Consider learning $M$ linear classification tasks cyclically. Suppose that Assumptions 5.1 and 5.2 holds. Let $B = \sum_{m=0}^{M-1} \beta_m$ and $V_\star = \sum_{m=0}^{M-1} \frac{1}{\beta_m} \|\nabla \mathcal{L}_m(\boldsymbol{w}_\star)\|^2$. If we take any step size satisfying $\eta \leq \frac{1}{2\sqrt{2}KB}$, the end-of-cycle iterates of sequential GD are contained in a compact set which is fixed in terms of the number of the cycle: for all $j = 0, 1, \cdots$,*

$$\boldsymbol{w}_0^{(jM)} \in \mathcal{W} := \left\{ \boldsymbol{w} \in \mathbb{R}^d : \|\boldsymbol{w} - \boldsymbol{w}_\star\|^2 \leq \left[ \frac{1}{Gb}\left(\mathcal{L}(\boldsymbol{w}_\star) + \sqrt{2}\eta KBV_\star\right) + \|\boldsymbol{w}_\star\| \right]^2 + 2\sqrt{2}\eta^2 K^2 BV_\star \right\}$$

$$\subseteq \left\{ \boldsymbol{w} \in \mathbb{R}^d : \|\boldsymbol{w} - \boldsymbol{w}_\star\|^2 \leq \left[ \frac{1}{Gb}\left(\mathcal{L}(\boldsymbol{w}_\star) + \frac{V_\star}{2}\right) + \|\boldsymbol{w}_\star\| \right]^2 + \frac{V_\star}{2\sqrt{2}B} \right\}.$$

*Proof.* The proof is done by induction based on Equation (57). We defer the proof to Appendix G.1.2. □

The last part of the proof is to compute the strong convexity coefficient of $\mathcal{L}$ on $\mathcal{W}$. Since $\mathcal{L}$ is twice differentiable, it can be directly done by computing a lower bound of the minimum Hessian eigenvalue on $\mathcal{W}$: for any $\boldsymbol{w} \in \mathcal{W}$,

$$\nabla^2 \mathcal{L}(\boldsymbol{w}) = \sum_{i=0}^{N-1} \ell''(\boldsymbol{x}_i^\top \boldsymbol{w}) \boldsymbol{x}_i \boldsymbol{x}_i^\top \succeq \left( \min_{\substack{i \in [0:N-1] \\ \boldsymbol{w} \in \mathcal{W}}} \ell''(\boldsymbol{x}_i^\top \boldsymbol{w}) \right) \boldsymbol{X} \boldsymbol{X}^\top \succeq \mu \boldsymbol{I}.$$

This concludes the proof of Lemma 5.1.

### G.1.1 PROOF OF LEMMA G.4

For the sake of readability, we restate the lemma.

**Lemma G.4** (Backward recurrent inequality). *Consider learning $M$ linear classification tasks cyclically. Suppose that Assumption 5.2 holds. Let $B = \sum_{m=0}^{M-1} \beta_m$. If we take any step size satisfying $\eta \leq \frac{1}{2\sqrt{2}KB}$, the iterates of sequential GD satisfies*

$$\left\| \boldsymbol{w}_0^{((j+1)M)} - \boldsymbol{u} \right\|^2$$

$$\leq \left\| \boldsymbol{w}_0^{(jM)} - \boldsymbol{u} \right\|^2 - 2\eta K \left[ \mathcal{L}\left(\boldsymbol{w}_0^{(jM)}\right) - \mathcal{L}(\boldsymbol{u}) \right] + 2\sqrt{2}\eta^2 K^2 B \left( \sum_{m=0}^{M-1} \frac{1}{\beta_m} \|\nabla \mathcal{L}_m(\boldsymbol{u})\|^2 \right),$$

*for any vector $\boldsymbol{u} \in \mathbb{R}^d$ and for all $j = 0, 1, \cdots$.*

We start the proof by bounding the squared distance between two iterates in the same cycle of continual learning. For $k \in [0 : K]$ and $m \in [0 : M - 1]$,

$$\left\| \boldsymbol{w}_0^{(jM)} - \boldsymbol{w}_k^{(jM+m)} \right\|^2$$

$$= \eta^2 \left\| \sum_{l=0}^{m-1} \sum_{h=0}^{K-1} \nabla \mathcal{L}_l(\boldsymbol{w}_h^{(jM+l)}) + \sum_{h=0}^{k-1} \nabla \mathcal{L}_m(\boldsymbol{w}_h^{(jM+m)}) \right\|^2$$

$$\leq \eta^2 \left( \sum_{l=0}^{M-1} \sum_{h=0}^{K-1} \beta_l \right) \left( \sum_{l=0}^{M-1} \sum_{h=0}^{K-1} \frac{1}{\beta_l} \left\| \nabla \mathcal{L}_l(\boldsymbol{w}_h^{(jM+l)}) \right\|^2 \right)$$

$$\leq 2\eta^2 K B \left( \sum_{l=0}^{M-1} \sum_{h=0}^{K-1} \frac{1}{\beta_l} \left[ \left\| \nabla \mathcal{L}_l(\boldsymbol{w}_h^{(jM+l)}) - \nabla \mathcal{L}_l(\boldsymbol{u}) \right\|^2 + \left\| \nabla \mathcal{L}_l(\boldsymbol{u}) \right\|^2 \right] \right)$$

$$\leq 4\eta^2 K B \sum_{l=0}^{M-1} \sum_{h=0}^{K-1} D_{\mathcal{L}_l}(\boldsymbol{u}, \boldsymbol{w}_h^{(jM+l)}) + 2\eta^2 K^2 B \sum_{l=0}^{M-1} \frac{1}{\beta_l} \left\| \nabla \mathcal{L}_l(\boldsymbol{u}) \right\|^2. \tag{58}$$

We use (modified) Jensen's inequality (e.g., Proposition G.2) in the first two inequalities above; the last inequality is due to Proposition G.1.

Next, we decompose the $(j + 1)$-th squared distance into $j$-th squared distance and more:

$$\left\| \boldsymbol{w}_0^{((j+1)M)} - \boldsymbol{u} \right\|^2$$

$$= \left\| \boldsymbol{w}_0^{(jM)} - \boldsymbol{u} \right\|^2 - 2\eta \sum_{m=0}^{M-1} \sum_{k=0}^{K-1} \left\langle \nabla \mathcal{L}_l(\boldsymbol{w}_k^{(jM+m)}), \boldsymbol{w}_0^{(jM)} - \boldsymbol{u} \right\rangle + \left\| \boldsymbol{w}_0^{(jM)} - \boldsymbol{w}_0^{((j+1)M)} \right\|^2.$$

Using Equation (56) and $\beta_m$-smoothnesses,

$$\left\| \boldsymbol{w}_0^{((j+1)M)} - \boldsymbol{u} \right\|^2 - \left\| \boldsymbol{w}_0^{(jM)} - \boldsymbol{u} \right\|^2$$

$$= -2\eta \sum_{m=0}^{M-1} \sum_{k=0}^{K-1} \left[ \mathcal{L}_m(\boldsymbol{w}_0^{(jM)}) - \mathcal{L}_m(\boldsymbol{u}) - D_{L_m}(\boldsymbol{w}_0^{(jM)}, \boldsymbol{w}_k^{(jM+m)}) + D_{L_m}(\boldsymbol{u}, \boldsymbol{w}_k^{(jM+m)}) \right]$$

$$+ \left\| \boldsymbol{w}_0^{(jM)} - \boldsymbol{w}_0^{((j+1)M)} \right\|^2$$

$$\leq -2\eta K \left[ \mathcal{L}\left( \boldsymbol{w}_0^{(jM)} \right) - \mathcal{L}(\boldsymbol{u}) \right] + \eta \sum_{m=0}^{M-1} \sum_{k=0}^{K-1} \beta_m \left\| \boldsymbol{w}_0^{(jM)} - \boldsymbol{w}_k^{(jM+m)} \right\|^2$$

$$- 2\eta \sum_{m=0}^{M-1} \sum_{k=0}^{K-1} D_{L_m}(\boldsymbol{u}, \boldsymbol{w}_k^{(jM+m)}) + \left\| \boldsymbol{w}_0^{(jM)} - \boldsymbol{w}_0^{((j+1)M)} \right\|^2$$

$$\leq -2\eta K \left[ \mathcal{L}\left( \boldsymbol{w}_0^{(jM)} \right) - \mathcal{L}(\boldsymbol{u}) \right] + 2\eta^2 K^2 B(1 + \eta K B) \sum_{m=0}^{M-1} \frac{1}{\beta_m} \left\| \nabla \mathcal{L}_m(\boldsymbol{u}) \right\|^2 \tag{59}$$

$$- 2\eta(1 - 2\eta K B - 2\eta^2 K^2 B^2) \sum_{m=0}^{M-1} \sum_{k=0}^{K-1} D_{L_m}(\boldsymbol{u}, \boldsymbol{w}_k^{(jM+m)})$$

$$\leq -2\eta K \left[ \mathcal{L}\left( \boldsymbol{w}_0^{(jM)} \right) - \mathcal{L}(\boldsymbol{u}) \right] + 2\sqrt{2}\eta^2 K^2 B \sum_{m=0}^{M-1} \frac{1}{\beta_m} \left\| \nabla \mathcal{L}_m(\boldsymbol{u}) \right\|^2.$$

In Equation (59), we use the result from Equation (58) multiple times. The last inequality is due to our choice of step size: $\eta K B \leq \frac{1}{2\sqrt{2}} < \frac{\sqrt{3}-1}{2} < \sqrt{2} - 1$ ($\because 1 - 2q - 2q^2 \geq 0$ if $q \in \left[ 0, \frac{\sqrt{3}-1}{2} \right]$). This is the end of the proof.

### G.1.2 PROOF OF LEMMA G.5

For the sake of readability, we restate the lemma.

**Lemma G.5** (Boundedness of the end-of-cycle iterates). *Consider learning $M$ linear classification tasks cyclically. Suppose that Assumptions 5.1 and 5.2 holds. Let $B = \sum_{m=0}^{M-1} \beta_m$ and $V_\star = \sum_{m=0}^{M-1} \frac{1}{\beta_m} \|\nabla \mathcal{L}_m(w_\star)\|^2$. If we take any step size satisfying $\eta \leq \frac{1}{2\sqrt{2}KB}$, the end-of-cycle iterates of sequential GD are contained in a compact set which is fixed in terms of the number of the cycle: for all $j = 0, 1, \cdots,$*

$$w_0^{(jM)} \in \mathcal{W} := \left\{ w \in \mathbb{R}^d : \|w - w_\star\|^2 \leq \left[ \frac{1}{Gb} \left( \mathcal{L}(w_\star) + \sqrt{2}\eta K B V_\star \right) + \|w_\star\| \right]^2 + 2\sqrt{2}\eta^2 K^2 B V_\star \right\}$$

$$\subseteq \left\{ w \in \mathbb{R}^d : \|w - w_\star\|^2 \leq \left[ \frac{1}{Gb} \left( \mathcal{L}(w_\star) + \frac{V_\star}{2} \right) + \|w_\star\| \right]^2 + \frac{V_\star}{2\sqrt{2}B} \right\}.$$

Also, recall the backward recurrent inequality, which we write here again:

$$\left\| w_0^{((j+1)M)} - w_\star \right\|^2 \leq \left\| w_0^{(jM)} - w_\star \right\|^2 - 2\eta K \left[ \mathcal{L}\left( w_0^{(jM)} \right) - \mathcal{L}(w_\star) \right] + 2\sqrt{2}\eta^2 K^2 B V_\star. \tag{60}$$

We choose $w_0^{(0)}$ as we want: if $w_0^{(0)} = 0$, since $\left\| w_0^{(0)} - w_\star \right\|^2 = \|w_\star\|^2$, it is clear that $w_0^{(0)} \in \mathcal{W}$.

Now assume $w_0^{(jM)} \in \mathcal{W}$ and proceed with induction on $j$: we aim to show $w_0^{((j+1)M)} \in \mathcal{W}$.

The proof is divided into two parts:

1. If the current total risk is too high, then we can show that the squared distance to $w_\star$ will decrease.
2. The other case means that the current iterate is close enough to $w_\star$ (due to the strict non-separability of the full dataset). Thus, the squared distance to $w_\star$ at the next cycle will not grow that much.

**Part 1: High-Risk Case.** Suppose $\mathcal{L}\left( w_0^{(jM)} \right) - \mathcal{L}(w_\star) \geq \sqrt{2}\eta K B V_\star$. Then Equation (60) implies $\left\| w_0^{((j+1)M)} - w_\star \right\|^2 \leq \left\| w_0^{(jM)} - w_\star \right\|^2$. Thus, $w_0^{((j+1)M)} \in \mathcal{W}$.

**Part 2: Low-Risk Case.** We first show that $\mathcal{L}\left( w_0^{(jM)} \right) - \mathcal{L}(w_\star) \leq \sqrt{2}\eta K B V_\star$ implies an upper bound on the current squared distance to $w_\star$. Because of Assumptions 5.1 and 5.2, for any $w \in \mathbb{R}^d$,

$$\mathcal{L}(w) = \sum_{i=0}^{N-1} \ell\left( x_i^\top w \right)$$

$$\geq \sum_{i=0}^{N-1} G \left[ x_i^\top w \right]^-$$

$$= G \|w\| \cdot \sum_{i=0}^{N-1} \left[ x_i^\top \frac{w}{\|w\|} \right]^-$$

$$\geq G \|w\| \, b,$$

by the definition of the non-separability $b > 0$. Thus, we have

$$\left\| w_0^{(jM)} - w_\star \right\| \leq \left\| w_0^{(jM)} \right\| + \|w_\star\|$$

$$\leq \frac{1}{Gb} \mathcal{L}\left( w_0^{(jM)} \right) + \|w_\star\|$$

$$\leq \frac{1}{Gb} \left[ \mathcal{L}(u) + \sqrt{2}\eta K B V_\star \right] + \|w_\star\|.$$

Thus, $w_0^{(jM)}$ lies in a strict subset of $\mathcal{W}$. Also, Equation (60) implies

$$\left\| w_0^{((j+1)M)} - w_\star \right\|^2 \leq \left\| w_0^{(jM)} - w_\star \right\|^2 + 2\sqrt{2}\eta^2 K^2 B V_\star$$

$$\leq \left[ \frac{1}{Gb} \left[ \mathcal{L}(\boldsymbol{u}) + \sqrt{2}\eta KBV_\star \right] + \|\boldsymbol{w}_\star\| \right]^2 + 2\sqrt{2}\eta^2 K^2 BV_\star.$$

Thus, by the definition of $\mathcal{W}$, $\boldsymbol{w}_0^{((j+1)M)} \in \mathcal{W}$. This concludes the proof of the lemma.

## G.2 Non-Asymptotic Loss Convergence Analysis (Proof of Theorem 5.2)

Recall that we write $B = \sum_{m=0}^{M-1} \beta_m$ and $V_\star = \sum_{m=0}^{M-1} \frac{1}{\beta_m} \|\nabla \mathcal{L}_m(\boldsymbol{w}_\star)\|^2$. Also, in the previous sub-section, we discovered a local strong convexity (with coefficient $\mu > 0$) of the total risk function satisfied on a compact ball $\mathcal{W}$ containing $\boldsymbol{w}_\star$ and every end-of-cycle iterate of the sequential GD.

Let us restate the theorem for the sake of readability.

**Theorem 5.2.** *Suppose we learn $M$ tasks cyclically for $J > 1$ cycles. We adopt the notation from Lemma 5.1. Then, with a step size $\eta = \mathcal{O}\left( \frac{1}{K} \cdot \min\left\{ \frac{1}{B}, \frac{\ln(J)}{J} \right\} \right)$, sequential GD satisfies*

$$\left\| \boldsymbol{w}_0^{(MJ)} - \boldsymbol{w}_\star \right\|^2 \leq \mathcal{O}\left( \exp\left( -\frac{\mu J}{(1+2\sqrt{2})B} \right) \cdot \left\| \boldsymbol{w}_0^{(0)} - \boldsymbol{w}_\star \right\|^2 + \frac{B^2 V_\star \ln^2 J}{\mu^3 J^2} \right). \tag{7}$$

**Remark G.1.** In fact, the full expression of our step size choice is:

$$\eta = \min\left\{ \frac{1}{2\sqrt{2}KB}, \frac{1+2\sqrt{2}}{2\sqrt{2}KJ} \ln\left( J^2 \cdot \max\left\{ 1, \frac{\|\boldsymbol{w}_0^{(0)} - \boldsymbol{w}_\star\|^2 \mu^3}{B^2 V_\star} \right\} \right) \right\},$$

which is omitted in the theorem statement above.

The theorem states a fast $\tilde{\mathcal{O}}(J^{-2})$ rate of convergence. One could try to prove it with the backward recurrent inequality (Equation (57)), but it is difficult due to the $\eta^2$ dependency of the so-called "noise" term. We only succeeded in proving a slower $\tilde{\mathcal{O}}(J^{-1})$ rate with the backward recurrent inequality, whose proof is pretty much similar to that in this sub-section. To take a step further towards a faster rate, we should use a different way of writing the recurrent inequality, with a higher exponent for $\eta$ in the "noise" term. Here is how it goes:

**Lemma G.6** (Forward recurrent inequality). *Consider learning $M$ linear classification tasks cyclically. Suppose that Assumption 5.2 holds. If we take any step size satisfying $\eta \leq \frac{1}{\sqrt{2}KB}$, the iterates of sequential GD satisfies*

$$\left\| \boldsymbol{w}_0^{((j+1)M)} - \boldsymbol{u} \right\|^2$$
$$\leq \left\| \boldsymbol{w}_0^{(jM)} - \boldsymbol{u} \right\|^2 - 2\eta K \left[ \mathcal{L}\left( \boldsymbol{w}_0^{((j+1)M)} \right) - \mathcal{L}(\boldsymbol{u}) \right] + 2\eta^3 K^3 B^2 \left( \sum_{m=0}^{M-1} \frac{1}{\beta_m} \|\nabla \mathcal{L}_m(\boldsymbol{u})\|^2 \right),$$

*for any vector $\boldsymbol{u} \in \mathbb{R}^d$ and for all $j = 0, 1, \cdots$.*

*Proof.* We defer the proof to Appendix G.2.1. We remark that this lemma holds even without assuming the non-separability. $\square$

In particular, we have

$$\left\| \boldsymbol{w}_0^{((j+1)M)} - \boldsymbol{w}_\star \right\|^2 \leq \left\| \boldsymbol{w}_0^{(jM)} - \boldsymbol{w}_\star \right\|^2 - 2\eta K \left[ \mathcal{L}\left( \boldsymbol{w}_0^{((j+1)M)} \right) - \mathcal{L}(\boldsymbol{w}_\star) \right] + 2\eta^3 K^3 B^2 V_\star. \tag{61}$$

Applying $\mu$-strong convexity, i.e.,

$$\mathcal{L}\left( \boldsymbol{w}_0^{((j+1)M)} \right) - \mathcal{L}(\boldsymbol{w}_\star) \geq \frac{\mu}{2} \left\| \boldsymbol{w}_0^{((j+1)M)} - \boldsymbol{w}_\star \right\|^2,$$

we eventually have a recurrent inequality purely on the squared distance to $\boldsymbol{w}_\star$:

$$\left\| \boldsymbol{w}_0^{((j+1)M)} - \boldsymbol{w}_\star \right\|^2 \leq \frac{1}{1+\eta K\mu} \left\| \boldsymbol{w}_0^{(jM)} - \boldsymbol{w}_\star \right\|^2 + 2\eta^3 K^3 B^2 V_\star. \tag{62}$$

We now conclude the proof by applying Proposition G.3: plugging $\alpha \leftarrow \eta K$, $\beta \leftarrow B$, $T \leftarrow J$, $c \leftarrow \frac{1}{2\sqrt{2}}$, $V \leftarrow 2B^2 V_\star$, $m = 2$, and $\Delta_j \leftarrow \left\| \boldsymbol{w}_0^{(jM)} \right\|$ to the proposition, we have a desired result.

### G.2.1 Proof of Lemma G.6

For the sake of readability, we restate the lemma, whose proof is very similar to that of Lemma G.4.

**Lemma G.6** (Forward recurrent inequality). *Consider learning $M$ linear classification tasks cyclically. Suppose that Assumption 5.2 holds. If we take any step size satisfying $\eta \leq \frac{1}{\sqrt{2}KB}$, the iterates of sequential GD satisfies*

$$\left\| \boldsymbol{w}_0^{((j+1)M)} - \boldsymbol{u} \right\|^2$$

$$\leq \left\| \boldsymbol{w}_0^{(jM)} - \boldsymbol{u} \right\|^2 - 2\eta K \left[ \mathcal{L}\left( \boldsymbol{w}_0^{((j+1)M)} \right) - \mathcal{L}(\boldsymbol{u}) \right] + 2\eta^3 K^3 B^2 \left( \sum_{m=0}^{M-1} \frac{1}{\beta_m} \left\| \nabla \mathcal{L}_m(\boldsymbol{u}) \right\|^2 \right),$$

*for any vector $\boldsymbol{u} \in \mathbb{R}^d$ and for all $j = 0, 1, \cdots$.*

We start the proof by bounding the squared distance between two iterates in the same cycle of continual learning. For $k \in [0 : K-1]$ and $m \in [0 : M-1]$,

$$\left\| \boldsymbol{w}_0^{((j+1)M)} - \boldsymbol{w}_k^{(jM+m)} \right\|^2$$

$$= \eta^2 \left\| \sum_{l=m+1}^{M-1} \sum_{h=0}^{K-1} \nabla \mathcal{L}_l(\boldsymbol{w}_h^{(jM+l)}) + \sum_{h=k}^{K-1} \nabla \mathcal{L}_m(\boldsymbol{w}_h^{(jM+m)}) \right\|^2$$

$$\leq \eta^2 \left( \sum_{l=0}^{M-1} \sum_{h=0}^{K-1} \beta_l \right) \left( \sum_{l=0}^{M-1} \sum_{h=0}^{K-1} \frac{1}{\beta_l} \left\| \nabla \mathcal{L}_l(\boldsymbol{w}_h^{(jM+l)}) \right\|^2 \right)$$

$$\leq 2\eta^2 KB \left( \sum_{l=0}^{M-1} \sum_{h=0}^{K-1} \frac{1}{\beta_l} \left[ \left\| \nabla \mathcal{L}_l(\boldsymbol{w}_h^{(jM+l)}) - \nabla \mathcal{L}_l(\boldsymbol{u}) \right\|^2 + \left\| \nabla \mathcal{L}_l(\boldsymbol{u}) \right\|^2 \right] \right)$$

$$\leq 4\eta^2 KB \sum_{l=0}^{M-1} \sum_{h=0}^{K-1} D_{\mathcal{L}_l}(\boldsymbol{u}, \boldsymbol{w}_h^{(jM+l)}) + 2\eta^2 K^2 B \sum_{l=0}^{M-1} \frac{1}{\beta_l} \left\| \nabla \mathcal{L}_l(\boldsymbol{u}) \right\|^2 \tag{63}$$

We use (modified) Jensen's inequality (e.g., Proposition G.2) in the first two inequalities above; the last inequality is due to Proposition G.1.

Next, we decompose the $j$-th squared distance into $(j+1)$-th squared distance and more:

$$\left\| \boldsymbol{w}_0^{(jM)} - \boldsymbol{u} \right\|^2 \geq \left\| \boldsymbol{w}_0^{((j+1)M)} - \boldsymbol{u} \right\|^2 + 2\eta \sum_{m=0}^{M-1} \sum_{k=0}^{K-1} \left\langle \nabla \mathcal{L}_l(\boldsymbol{w}_k^{(jM+m)}), \boldsymbol{w}_0^{((j+1)M)} - \boldsymbol{u} \right\rangle.$$

Using Equation (56) and $\beta_m$-smoothnesses,

$$\left\| \boldsymbol{w}_0^{((j+1)M)} - \boldsymbol{u} \right\|^2 - \left\| \boldsymbol{w}_0^{(jM)} - \boldsymbol{u} \right\|^2$$

$$\leq -2\eta \sum_{m=0}^{M-1} \sum_{k=0}^{K-1} \left[ \mathcal{L}_m(\boldsymbol{w}_0^{((j+1)M)}) - \mathcal{L}_m(\boldsymbol{u}) - D_{\mathcal{L}_m}(\boldsymbol{w}_0^{((j+1)M)}, \boldsymbol{w}_k^{(jM+m)}) + D_{\mathcal{L}_m}(\boldsymbol{u}, \boldsymbol{w}_k^{(jM+m)}) \right]$$

$$\leq -2\eta K \left[ \mathcal{L}\left( \boldsymbol{w}_0^{((j+1)M)} \right) - \mathcal{L}(\boldsymbol{u}) \right] + \eta \sum_{m=0}^{M-1} \sum_{k=0}^{K-1} \beta_m \left\| \boldsymbol{w}_0^{((j+1)M)} - \boldsymbol{w}_k^{(jM+m)} \right\|^2$$

$$- 2\eta \sum_{m=0}^{M-1} \sum_{k=0}^{K-1} D_{\mathcal{L}_m}(\boldsymbol{u}, \boldsymbol{w}_k^{(jM+m)})$$

$$\leq -2\eta K \left[ \mathcal{L}\left( \boldsymbol{w}_0^{((j+1)M)} \right) - \mathcal{L}(\boldsymbol{u}) \right] + 2\eta^3 K^3 B^2 \sum_{m=0}^{M-1} \frac{1}{\beta_m} \left\| \nabla \mathcal{L}_m(\boldsymbol{u}) \right\|^2$$

$$- 2\eta(1 - 2\eta^2 K^2 B^2) \sum_{m=0}^{M-1} \sum_{k=0}^{K-1} D_{\mathcal{L}_m}(\boldsymbol{u}, \boldsymbol{w}_k^{(jM+m)}) \tag{64}$$

$$\leq -2\eta K \left[ \mathcal{L}\left(\boldsymbol{w}_0^{((j+1)M)}\right) - \mathcal{L}(\boldsymbol{u})\right] + 2\eta^3 K^3 B^2 \sum_{m=0}^{M-1} \frac{1}{\beta_m} \left\| \nabla \mathcal{L}_m(\boldsymbol{u})\right\|^2.$$

In Equation (64), we use the result from Equation (63) multiple times. The last inequality is due to our choice of step size: $\eta K B \leq \frac{1}{\sqrt{2}}$. This is the end of the proof.

