# OpenReview forum: "Convergence and Implicit Bias of Gradient Descent on Continual Linear Classification"
_ICLR.cc/2025/Conference — ICLR 2025 Poster_

### Official Review · Reviewer_a4PQ · 2024-11-03

**Soundness:** 3
**Presentation:** 4
**Contribution:** 3
**Rating:** 8
**Confidence:** 4

**Summary:**

The paper analyses continual learning in a linear classification setting.
The analysis focuses on a simple unregularized setting with a fixed number of gradient steps per task (in contrast to a previous paper that studied the regularized setting when learnt to convergence).
Both asymptotic and non-asymptotic results are derived, showing that when task recur cyclically or randomly, the iterates converge to the global offline max-margin solution across all tasks.
This paper advances the theoretical CL field by clearly showing a different behavior than the one of the weakly regularized case studied in a previous paper.

**Strengths:**

- **The paper is well written**.
- **The contributions are clear and significant to the theoretical CL field**.
The main results show the implicit bias of continual linear classification when learned in a practical setup with a fixed number of gradient steps, given that tasks recur in the sequence. Consequently, they show a separation between that setup and a previously-studied weakly-regularized setup which can be considered as less practical.

**Weaknesses:**

- **Results are not especially insightful**.
The previous paper by Evron et al. (ICML 2023) proved a clear projection-like behavior ("SMM") under the weakly-regularized continual linear classification setup. Their results showed an iterative behavior that is somewhat insightful even for "arbitrary" task sequences.
However, the paper reviewed here proves a behavior that is only informative in the presence of task repetitions.
(This is not strictly a weakness since it might be the only behavior that one could possibly prove; but still it is only insightful given task recurrence.)

- **The appendices are hard to follow and verify.** Many steps are insufficiently explained (e.g., it's not very clear how Eq. (20) stems from (19); the steps in Eq. (26) are also unclear).
This prevented me from quickly validating a random subset of the proofs.

**Questions:**

I have no questions but rather some minor remarks that I suggest that the authors address in their revisioned version. I do not expect an author response on these bullets.

- Line 62: I wouldn't say SMM *"cannot be shown to converge to the offline max-margin solution"*, but rather that it *does not always* converge to that solution (as shown in Figure 1c of Evron et al. 2023). Also, I'm not sure that it should be considered a *"limitation"* of SMM, since this is merely the behavior of the weakly-regularized setup. I wouldn't even say that it's an advantage of the unregularized setup on the weakly-regularized setup, but rather that we see here two different behaviors that can describe continual classification to some extent.
- Figure 1 could be clearer. Maybe a 2d example would be more suitable for a paper?
- Line 105: the sentence negates the forgetting and the loss but at this point it's still unclear what the difference is.
- Line 112: is "symptotic" a word? That part of the sentence might need to be rephrased.
- Line 143: "be a set of" => "be the set of"?
- Line 212: not sure "generic" is a good choice here.
- Theorems 3.1 & 4.1:
    - In the second result, *"Every data point is classified correctly"*; perhaps say ".. is **eventually** classified correctly"?
    - The third result seems like an auxiliary result (proposition?) that is not of any practical interest (unlike the first two). I would reconsider its current location in the main results.
- Figure 2: not very clear. Maybe add a subfigure with the actual 2d parameter space?
- Line 432: do you really need to span the *whole* space? In linear regression we can usually ignore unused directions due to the rotation invariance. Maybe I am missing something, but please reverify this.
- Line 434: strictly or strongly?
- Line 436: *"faster rate"*, faster than what?
- Theorem 5.2: authors should explain the differences from the separable case. And also explicitly explain how the bound on the iterate gap yields a bound on the loss and forgetting.
- Line 506: "implicit bias on" => "of"?
- Line 523 say *"are no solution"* instead of "is".

- Many formulas in the Appendices exceed the page margins (e.g., Pages 19-20).
- Some citations are incomplete or inaccurate. For instance, *"Stochastic gradient descent on separable data: Exact convergence with a fixed learning rate"* was published at AISTATS 2019; and *"Continual Learning with Tiny Episodic Memories"* was published at ICML 2019.

- Line 806-809: Evron et al. (2023) already showed it (see their aforementioned Figure 1c).
- Line 828: might need rephrasing.
- Line 831 say *"contrained"*.

- Line 918: *"proposed by indicates"*, rephrase?
- Line 1152: Missing whitespace.
- Equations (74,75): It's not common to use the $O(\cdot)$ notation for negative quantities. I think the authors need instead $\log(t)-\log(t-1)=O(t^{-1})$.

---

> ### Author Response · Authors · 2024-11-22
> **Author's Response (1)**
>
> We sincerely appreciate the reviewer for taking the time and efforts to read and examine our paper closely. Your dedication helped us improve the paper significantly. Below, let us address your comments and questions:
>
> **W1. The results are only informative under certain task repetitions; cannot handle arbitrary task sequences.**
>
> - Thanks to the connection between weakly-regularized continual learning and SMM, Evron et al. (2023) obtain the exact trajectory of iterates over different stages obtained by sequential projections. On the other hand, in our sequential GD setting, it is very difficult to keep track of the exact location of the iterates, since the iterates are updated multiple times but training stops before convergence. To overcome this issue, we use different proof techniques motivated from (Soudry et al. 2018; Nacson et al. 2019) to analyze the direction in which sequential GD eventually converges to. We are not sure if any conclusions we draw from our analysis can extend to arbitrary sequences.
> - Nonetheless, we believe that our theoretical results provide some useful insights to more general continual classification scenarios. Specifically, we believe that our theoretical findings can be extended to a more general **online** continual learning setup, beyond the current setup that uses the same data points over and over again. Consider a finite number of tasks’ data distributions (with bounded supports) which are jointly separable, and let them be cyclically/randomly chosen one by one at each stage. Suppose a finite-size training dataset from each data distribution is sampled every time we encounter a task. Even in this case, we expect that the story would be the same: the continually trained linear classifier will eventually converge to the max-margin direction that jointly classify every task’s data distribution support. In our revised paper, we empirically demonstrate this in Appendix C.2.4.
>
> **W2. Readability issues of appendices**
>
> - We are sorry for any confusion caused by our proofs. We have supplemented our explanations with more explanations for the involved steps. We aim to supplement appendices with proof sketches carefully delivering key technical challenges and insights, but due to the length/complexity of proofs and lack of time we were not able to finish this in time. We plan to do so after the discussion period.
> - From Eq. (19) to Eq. (20): We found there was a missing factor inside Eq. (19). We corrected it by adding the gradient norm factor to (19).
> - Eq. (26): we add a supplementary explanation. The third inequality comes from the property of the largest singular value, and the last equality holds by the definition of the gradient update rule.
>
> **Q. A collection of suggestions for improving the writing**
>
> - We greatly appreciate for the reviewer’s careful reading and insightful comments on improving our manuscript. Although the reviewer do not expect responses about these comments, we provide answers to some of them.
> - Line 62: Thank you for an insightful comment. As per the reviewer’s suggestion, we toned down our claims about the existing work Evron et al. (2023); we realize we were overly critical as we try to motivate our setting. Please check out our updated introduction (Section 1).
> - Figure 1: The reason why we opt for 3D visualization in Figure 1 is to represent the irrelevance between the directions of individual tasks’ max-margin and the joint max-margin classifiers. We also conducted 2D experiments and made a cleaner visualization of directional convergence to joint max-margin direction (e.g., Figure 2). For more details about experiments, please take a look at our **General Response** and the updated manuscript.
> - Line 432: we assumed the data spanning whole space without loss of generality. This is because, as we mentioned in the same paragraph, every gradient update happens in the span of data points. In other words, the dynamics orthogonal to the data span is constant, so we simply neglect these components. We made it more clear in our updated manuscript by adding “without loss of generality”.
> - Line 434: the original “strictly” is correct. The empirical risk function based on logistic function cannot be *globally* strongly convex (i.e., bounded below by positive-definite quadratic functions everywhere). In particular in the strictly non-separable case, however, the training loss becomes *locally* strongly convex near the optimum, which we prove to eventually obtain our convergence result for non-separable case.
> - Theorem 5.2: since the total loss $\mathcal{L}(w)$ is $B$-smooth (where $B=\sum_{m=0}^{M-1} \beta_m$), it satisfies that $\mathcal{L}(w) - \mathcal{L}(w_\star) \le \langle \nabla \mathcal{L}(w_\star), w-w_\star \rangle + \tfrac{B}{2}\\|w-w_\star\\|^2 = \tfrac{B}{2}\\|w-w_\star\\|^2$.  Thus, the iterate convergence naturally implies the loss convergence with an exactly the same rate. We made this clearer in the updated manuscript.

---

> > ### Author Response · Authors · 2024-11-22
> > **Author's Response (2)**
> >
> > **References**
> >
> > - Evron et al., Continual learning in linear classification on separable data. ICML 2023.
> > - Soudry et al., The implicit bias of gradient descent on separable data. JMLR 2018.
> > - Nacson et al., Stochastic gradient descent on separable data: Exact convergence with a fixed learning rate. AISTATS 2019.
> > - Chaudhry et al., On tiny episodic memories in continual learning. arXiv preprint. 2019.

---

> > > ### Comment · Reviewer_a4PQ · 2024-11-25
> > > **Raising my score**
> > >
> > > I have read the other reviews and (most of) the author responses.
> > > I also skimmed over the revised paper.
> > >
> > > I believe the reviewing process has improved the paper, which is a good paper and, in my humble opinion, is now ready for publication.
> > > Therefore, I raise my score to 8.
> > >
> > > ----
> > >
> > > Final remarks:
> > > 1. I encourage the authors to include some general proof/appendix sketches like they said they will add in the next version.
> > > 2. I believe that the revised part in Line 204 should be $\phi=1/\Vert \hat{w} \Vert$.

---

> > > > ### Author Response · Authors · 2024-11-25
> > > > **Thank you very much**
> > > >
> > > > Thank you very much for reading our rebuttal and raising the score! We are currently working on the proof sketches and will definitely put them in our final manuscript. Also, thank you for correcting the typo. We contained this correction in the currently revised version of our paper.

---

### Official Review · Reviewer_uaBW · 2024-11-03

**Soundness:** 3
**Presentation:** 3
**Contribution:** 3
**Rating:** 6
**Confidence:** 3

**Summary:**

This paper studies continual linear classification tasks by sequentially running gradient descent (GD) on the unregularized (logistic) loss for a fixed budget of iterations per each given task. When all tasks are jointly linearly separable and are presented in a cyclic/random order, the full training loss asymptotically converges to zero and the trained linear classifier converges to the joint (offline) max-margin solution (i.e., directional convergence). Additionally, when tasks are given in a cyclic order, non-asymptotic analysis on cycle-averaged forgetting and loss convergence at cycle $J$ are provided, which are $O(\frac{\ln^4J}{J^2})$ and $O(\frac{\ln^2J}{J})$ respectively. Lastly, when the tasks are no longer jointly separable and presented cyclically, a fast non-asymptotic convergence rate of $\tilde{O}(J^{-2})$ towards the global minimum of the joint training loss is established.

**Strengths:**

1. The paper is easy to follow and clearly demonstrates the contribution of the theoretical results.

2. When all tasks are jointly linearly separable, standard GD without any regularization not only learns every task but also converges to the joint max-margin direction eventually. The implicit bias happens on both cyclic/random task ordering, which is new to my knowledge.

3. When tasks are presented in a cyclic order, non-asymptotic convergence rates are established: $O\left(\frac{\ln^4 J}{J^2}\right)$ if the tasks are jointly linearly separable, and $\tilde{O}(J^{-2})$ if they are not.

**Weaknesses:**

1. The theoretical framework for continual learning in this paper does not fully align with practical settings in deep learning; for example, tasks are assumed to follow a cyclic order, although this is a common setup in theoretical studies on continual learning.

2. The experimental setup of the paper is quite limited. While the toy example in Figure 1 effectively illustrates the motivation, additional empirical results under more practical continual learning scenarios—or at least beyond the simplest toy example—are necessary to validate the theoretical findings, e.g., the implicit bias phenomenon for both cyclic/random order, cycled average forgetting converges faster than the joint training loss.

3. The proofs in the Appendix are involved. Although the remark paragraphs following each theorem provide brief explanations, proof sketches or high-level technical overviews for all of the theoretical results are lacking. Specifically, it is unclear what technical novelties are introduced from a high-level perspective and how the current analysis improves upon [Evron et al. (2023)](https://arxiv.org/pdf/2306.03534).

**Questions:**

1. Could you provide a geometric interpretation or intuition of the theoretical results, just as what [Goldfarb et al. (2024)](https://arxiv.org/pdf/2401.12617), [Evron et al. (2023)](https://arxiv.org/pdf/2306.03534) and [Evron et al. (2022)](https://arxiv.org/pdf/2205.09588) did?

2. Could you provide proof sketches and a technical overview of the theoretical analysis? In particular, it would be helpful to understand more about the connections and distinctions from prior works, such as [Evron et al. (2023)](https://arxiv.org/pdf/2306.03534) and [Evron et al. (2022)](https://arxiv.org/pdf/2205.09588), as well as what new technical contributions this work presents. Appendix A.1 provides a brief overview and comparisons with [Evron et al. (2023)](https://arxiv.org/pdf/2306.03534), but it lacks an in-depth discussion of the technical and theoretical challenges and contributions.

**Details Of Ethics Concerns:**

No ethics concerns.

---

> ### Author Response · Authors · 2024-11-22
> **Author's Response (1)**
>
> We highly appreciate the reviewer for their constructive feedback. Below, let us provide our response to the comments and questions raised by the reviewer:
>
> **W1. Misalignment with practical settings, e.g., cyclic ordering of tasks**
>
> Thank you for the insightful question. Indeed, our setting does not fully align with the practice of continual learning. However, we believe that our theoretical analyses are useful because 1) the insights can carry over to more general setups, 2) they offer a theoretical basis for a simple baseline algorithm, and 3) they can serve as a stepping stone for more sophisticated analyses on continual learning techniques.
>
> 1. Indeed, if there are all training data for all tasks given a priori, one can combine them and run SGD. However, we note that there can be scenarios where utilizing the combined training dataset becomes **impossible**, e.g., when the datasets for different tasks are stored in distributed data silos and communication of data points is expensive or prohibited. Moreover, we believe that our theoretical findings can be extended to a more general **online** continual learning setup, beyond the current setup that uses the same data points over and over again. Consider a finite number of tasks’ data distributions (with bounded supports) which are jointly separable, and let them be cyclically/randomly chosen one by one at each stage. Suppose a finite-size training dataset from each data distribution is sampled every time we encounter a task. Even in this case, we expect that the story would be the same: the continually trained linear classifier will eventually converge to the max-margin direction that jointly classify every task’s data distribution support. In our revised paper, we empirically demonstrate this in Appendix C.2.4.
> 2. Our theoretical analyses may not immediately offer any guidelines or suggestions for practitioners interested in general continual learning problems. However, we believe that our work provides a theoretical basis for a concrete **baseline**: “you should do better than this.” Our theory reveals that when the same (or similar) tasks appear repeatedly, a naive algorithm without any techniques (e.g., regularization or replay) specialized for continual learning will eventually “work,” in terms of convergence to a desirable offline solution and vanishing forgetting. Thus, we expect that such a naive algorithm does not always fail in more practical scenarios, and any new approach for continual learning should be able to outperform this simple baseline.
> 3. Furthermore, we believe our analysis can serve as a stepping stone to more advanced theoretical analyses on more practical approaches such as regularization- and replay-based methods.
>
> **W2. Empirical results to validate the theoretical findings**
>
> - Thank you for pointing this out. As all reviewers raised similar concerns, we posted a **General Response** specifically about experiments; please have a look. To be a bit more specific, we performed several experiments on synthetic 2D datasets in order to validate our theoretical results under cyclic/random task ordering. Also, we provide an experiments on a more realistic dataset (CIFAR10). Besides, we also plan to conduct deep learning experiments and will put the results in our paper.

---

> > ### Author Response · Authors · 2024-11-22
> > **Author's Response (2)**
> >
> > **W3+Q2. Proof sketches and High-level technical overviews.**
> >
> > - Thank you for your comment, and we agree with your point. Unfortunately, since our long  proofs operate with a bunch of highly technical lemmas and calculations, it is not an easy task to provide a high-level idea of the proof structure and to highlight the technical novelties without any backgrounds: it will take too much time. We plan to do so after the discussion period.
> > - Improvement upon Evron et al. (2023)
> >     - The major distinction from Evron et al. (2023) in terms of proof techniques comes from the difference in the analyzed setting. Evron et al. (2023) use regularized continual learning. Linearly separable classification with a regularized objective function yields a unique minimum in a finite position. Hence, within finite iterations, regularized continual learning can converge to its minimum. They prove that the iterates at the end of every stage behave like a projection method as the regularizer converges to zero. On the other hand, we use sequential continual learning without a regularizer. This means that there is no minimum at any finite location. Thus we can’t learn one task until it converges to its minimum as Evron et al. (2023) did.
> >     - This yields technical challenges to analyze implicit bias. First of all, while one task is being trained, the model tends to diverge in the task’s own max-margin direction. Thus each task fights other tasks to train the model in their own max-margin direction. Second, Evron et al. (2023) use the projection method and obtain the exact trajectory of every stage. in our sequential GD setting, it is very difficult to keep track of the exact location of the iterate after one task is trained. The uncertainty accumulates and grows larger as training goes on. To handle this issue, we focus on “where the iterate eventually converges to as the cycle increases”, not “where the iterate is after each stage”. Based on our experimental observation, we set the proxy $\log t\cdot  \hat{w}$ and show the distance between the proxy and iterate is bounded by some constant value. Since the proxy diverges to infinity, the boundedness implies implicit bias in the direction of iteration.
> >
> > **Q1. Geometric Interpretation or intuition of the theoretical results.**
> >
> > - Unlike the results of [Goldfarb et al. (2024)](https://arxiv.org/pdf/2401.12617), [Evron et al. (2023)](https://arxiv.org/pdf/2306.03534), and [Evron et al. (2022)](https://arxiv.org/pdf/2205.09588), which use projection methods to learn a new task from the current task, our method uses gradient descent for fixed iterations. This technical difference makes it challenging to pinpoint where the model arrives after training on a certain task. Even when we know at which point we start learning, it is hard to pinpoint where we will end up at after one cycle. However, our result shows that after many cycles, the direction of the model will close to the joint max-margin direction. We provide this geometric interpretation in Figure 1 as 3D plot. We also include 2D visualization of our result in the updated manuscript (see Figure 2).

---

> > > ### Comment · Reviewer_uaBW · 2024-11-26
> > >
> > > I thank the authors for their rebuttal. It addresses my concerns, and I suggest including the proof sketches in the final version. I will keep my score.

---

### Official Review · Reviewer_1DCj · 2024-11-04

**Soundness:** 3
**Presentation:** 3
**Contribution:** 3
**Rating:** 6
**Confidence:** 3

**Summary:**

This paper investigates how gradient descent (GD) converges to the joint max-margin solution for multiple tasks that are jointly linearly separable. The authors present loss and forgetting bounds under both cyclic and random task presentation orders. Theoretical results are provided for both asymptotic and non-asymptotic settings. Overall, this is a highly novel and interesting paper that offers promising insights for continual learning.

**Strengths:**

1. Novel analysis of continual learning: The paper contributes to the understanding of continual learning by offering new insights into the convergence behavior of gradient descent on multiple tasks.

2. The presentation is clear and easy to follow.

**Weaknesses:**

This is a very novel and interesting paper, but I have the following concerns,

1. The logistic loss function and linear model are somehow limited. We usually use large foundation model and cross entropy loss nowadays. So can this theorem be generalized to more practical cases.

2. No convincing experiments and compared baselines are provided, such as deep learning experiments for continual learning.

**Questions:**

I have the following questions:
1. Can you provide the intuition about Assumption 3.4 Tight Exponential Tail?

2. Figure 1 present a toy example to help us to understand the motivation, but all the tasks with labels 1. It seems not realistic, so I am curious whether this setting is necessary, can you make it more general.

3. What is the definition of task similarity in continual learning? Do $N_{p, q}$ and $\bar{N}_{p, q}$ in Theorem 3.4 measure the task similarity? I hope the author can give the clear definition of task similarity. Intuitively, task similarity will affect the forgetting as well as training loss. For example, there are two tasks with significant different optimized directions and a model is trained to learn two tasks, I think it's usually harder than learning two similar tasks, so it probably has larger training loss. So it is reasonable that task similarity would affect the training loss either, do author agree with me?

4. This paper gives a new definition about cycle-averaged forgetting $\mathcal{CF}(J)$, and also show the upper and lower bound of $\mathcal{CF}(J)$ in Theorem 3.4. I think this definition somehow restricts the forgetting measure within the current cycle. The problem is that can you guarantee the model performs better on task $m$ in the current cycle $J$ than before? If not, even if $\mathcal{CF}(J) \rightarrow 0$, we can not say the forgetting performance is good. Is it possible to change the baseline as $\min_j\mathcal{L}_{m}(w_K^{Mj+m})$, which considers the best performance of model on task $m$. Then the cycle-averaged forgetting $\mathcal{CF}(J) = \frac{1}{M}\sum\_{m=1}^{M-1}\mathcal{L}\_{m}(w_0^{MJ+M}) - \mathcal{L}\_{m}(w_K^{Mj+m})$
which is consistent with conventional definition of forgetting.

5. Line 412: $\lim_{t\rightarrow \infty} x_i^T w_k^{(t)}=\infty$?

---

> ### Author Response · Authors · 2024-11-22
> **Author's Response (1)**
>
> We appreciate the reviewer for their insightful and positive review. Below, we address the reviewer’s concern and answer the raised questions.
>
> **W1. Limited setting: logistic loss and linear model.**
>
> - We remind the reviewer that logistic loss is a special case of cross-entropy loss for binary classification. Since cross-entropy loss shares the same shape as logistic loss, we should be able to extend our analysis to general multi-class classification as Soudry et al. (2018) did. We only considered the binary classification case just for simplicity of presentation.
> - Due to its non-convexity, non-linearity, and the existence of local minima, theoretical understanding of deeper models is much more involved. In our work, we consider linear classifiers as a starting point of our theoretical analysis since they are more amenable to study. However, we believe some insights from our results can be extended to a larger model; for instance, when tasks are given repeatedly in a neural network setup, we expect that the total loss would eventually decrease even with the naive and problem-agnostic sequential GD algorithm. We plan to add an empirical verification of our claim later by the end of the discussion period.
>
> **W2. Lack of experiments, e.g., deep learning.**
>
> - Thank you for pointing this out. We plan to add some deep learning experiments later in our revisions. However, due to time constraint and computational burden, we will attach some of the results before the end of the discussion period, and will attach some other results later. Please check out our **General Response** handling all reviewer’s concerns about lack of experiments.
>
> **Q1. Intuition behind “Tight Exponential Tail” Assumption**
>
> - The tight exponential tail assumption means the loss gradient exponentially decreases as the prediction becomes more confident in the accurate direction. Loss functions such as logistic loss and cross-entropy loss satisfy this condition. Since those functions are practically used in classification, it is a standard assumption.
> - Also, we would like to mention that the assumption is crucial in proving the desired directional convergence to the max-margin direction. If the loss function does not meet the tight exponential tail assumption, then it can induce a totally different implicit bias of the optimization algorithm (Ji et al., 2020).
>
> **Q2. Figure 1: Why all data have labels +1?**
>
> - In binary linear classification, a dataset $\\\{(x_i, y_i)\\\}$ which has labels $y_i\in \\\{-1,1\\\}$ is equivalent to a dataset $\\\{(y_i x_i, 1)\\\}$. That is, the two datasets result in exactly the same loss function, optimization trajectory, max-margin direction, etc. Thus, we choose all data to have labels +1 without loss of generality.

---

> ### Author Response · Authors · 2024-11-22
> **Author's Response (2)**
>
> **Q3. Regarding the definition of task similarity**
>
> - To the best of our knowledge, there is no single widely accepted definition of task similarity. In the well-studied continual linear **regression** setting, various notions of task similarity have been proposed by several previous works (Doan et al., 2021; Evron et al., 2022; Lin et al., 2023). The task similarity is of great interest in this line of work because it is often closely connected to the continual learning performance (e.g., catastrophic forgetting or knowledge transfer). In continual **classification** setup like ours, we do not know much of works studying the relevance between task similarity and forgetting.
> - In our paper, we theoretically characterize the connection between the amount of forgetting per cycle and a form of “alignment” between different tasks. The term $N_{p,q}$ ($\bar {N}\_{p,q}$, resp.) captures positive (negative, resp.) alignment of the data pairs between tasks $p$ and $q$. We view these terms as positive/negative alignments, because they are sums of positive/negative inner products between data points in two tasks. We originally explained $N_{p,q}$ and $\bar{N}\_{p,q}$ as “task similarities” for simplicity because, similar to the previous works revealing connection between task similarity and forgetting, we figure out they are closely related to the amount of forgetting: high $N_{p,q}$ and low $\bar{N}\_{p,q}$ yield low forgetting (and thus high knowledge transfer), while low $N_{p,q}$ and high $\bar{N}\_{p,q}$ yield high forgetting. However, we realized that the terminology “task similarity” may be slightly misleading, because even two identical tasks (say $p$) may have negative $\bar{N}\_{p,p}$. We revised the writing in our manuscript.
> - Lastly, we comment on the reviewer’s intuition on the relationship between task similarity and loss convergence. We totally agree with the reviewer’s intuition on the connection between the task similarity (or our specific notion of task alignments) and loss convergence. However, we remark that our loss convergence analysis in Theorem 3.3 does not quite capture that intuition. The non-asymptotic convergence upper bound we presented only captures the **worst-case** scenario (which is due to the nature of upper bounds), where the model experiences the largest possible forgetting during each cycle. This would correspond to the case of significantly dissimilar but jointly separable tasks, as we illustrate in Appendix C.3 through a toy example.
>
> **Q4. Regarding the definition of cycle-averaged forgetting**
>
> - As you correctly pointed out, the convergence of $\mathcal{C}\mathcal{F}(J)$ to zero may not imply the convergence of the “conventional forgetting” to zero. There can be cases where $\mathcal{C}\mathcal{F}(J)$ decays to zero but the overall loss increases over cycles, hence resulting in larger and larger conventional forgetting. For the question “can you guarantee the model performs better on task $m$ in the current cycle $J$ than before?”, the answer is “not always.” We presented an example in Appendix C.3 where the loss on a certain task increases over some early stages of training. In Figure 9(a), one can see that the loss for Task 1 (green line) initially increases during early cycles.
> - However, we can see from the Figure 9(b) that the loss eventually decreases, as predicted by our Theorem 3.3. Indeed, Theorem 3.3 guarantees that the total loss will eventually converge to zero, which also implies that the individual task losses will also converge to zero. From this, we expected that the model will eventually enter a stage where the individual task losses will monotonically decrease over cycles, in which case the cycle-averaged forgetting coincides with the conventional forgetting. Also, convergence of individual losses to zero already implies that the conventional forgetting will also eventually converge to zero. For these reasons, we chose to focus on the analysis of cycle-averaged forgetting.
> - It is possible to extend our analysis of cycle-averaged forgetting to conventional forgetting. However, due to limitations of our proof techniques, the upper and lower bounds become looser as we have to accumulate the task alignment terms over multiple cycles.
>
> **Q5. Typo in Line 412**
>
> - Thank you for your careful reading. We reflected your suggestion to our updated manuscript.
>
>
> ---
>
> **References**
>
> - Doan et al., A Theoretical Analysis of Catastrophic Forgetting through the NTK Overlap Matrix, AISTATS 2021.
> - Evron et al., How catastrophic can catastrophic forgetting be in linear regression?, COLT 2022.
> - Ji et al., Gradient descent follows the regularization path for general losses, COLT 2020.
> - Lin et al., Theory on Forgetting and Generalization of Continual Learning, ICML 2023.

---

> > ### Comment · Reviewer_1DCj · 2024-11-26
> >
> > Thank you for the responses. It has addressed most of my concerns. Based on my overall assessment of the paper, I prefer to maintain my current rating.

---

### Official Review · Reviewer_sGag · 2024-11-04

**Soundness:** 4
**Presentation:** 4
**Contribution:** 2
**Rating:** 5
**Confidence:** 3

**Summary:**

The paper considers continual learning setting in which a linear classifier is trained for a fixed number of iterations on tasks (either cyclic or random). In the separable case, the papers shows (directional) convergence to the offline (i.e., combined) max-margin classifier for cyclic and randomised ordering of tasks. In the separable, the paper also characterises (i) forgetting and (ii) backward transfer (for aligned tasks). The paper also shows convergence to overall minimum in the non-separable case.

**Strengths:**

Clarity : The paper is clearly written.
Soundness : The theorems are stated clearly with proofs.
Originality : Shows results under fixed budget of iterations at every stage. This is an improvement over the Sequential Max-Margin.

**Weaknesses:**

The main weakness is in the novelty and significance.

1) No experiment with real world dataset.

2) No experiments with realistic synthetic dataset : The paper presents only one experiment on a synthetic dataset with 6 points. Experiments should be detailed in such a way to bring out all the important results.

3) Setting may not be novel: Given that there is an overall linear classifier, the protocols of cycling training or randomised training seem superfluous. To elaborate, we are not in the online learning setting, so we have all the training data of all the tasks apriori. We also know that there is a single linear classifier. So what is wrong if we combine the datasets of all the tasks into one and train using SGD with batches of data (batches either cycled or random) as it is done always?

4) Are batches Tasks? : It is a standard practice to use SGD with batches of data (either cyclic or random). And it is not surprising/novel to know that SGD with batch converges. Is the paper just fancily calling batches as tasks?

5) No Connection to realistic/standard settings : The paper mentions that "cyclic task ordering covers search engines influenced by periodic events. Random task ordering bears resemblance to Autonomous driving in randomly recurring environments". These instances do convey meaning of cyclic, and randomised tasks for a reader who is looking at these the first time. However, the paper has to be specific in terms of specific set of standard datasets and standard real world scenarios.

6) No recommendations for practice : Theoretical results are very important and sometimes can be obtained under even restrictive assumptions. However, the results should also give rise to some recommendation in realistic scenarios. This paper fails in this aspect.

**Questions:**

Please see the weaknesses.

---

> ### Author Response · Authors · 2024-11-22
> **Author's Response (1)**
>
> We thank the reviewer for their detailed and insightful comments on our work. Below, we address the concerns raised by the reviewer.
>
> **W1+W2. Experiments with real-world datasets & realistic synthetic datasets.**
>
> - Thank you for pointing these out. Please check out our **General Response** handling all questions and concerns about experiments raised by all reviewers. For your specific concerns, please refer to #1 and #2.
>
> **W3. We know all training data in advance; why don’t we combine them and use SGD?**
>
> - Thank you for the insightful question. Indeed, if there are all training data for all tasks given a priori, one can combine them and run SGD. However, there can be scenarios where utilizing the combined training dataset becomes **impossible**, e.g., when the datasets for different tasks are stored in distributed data silos and communication of data points is expensive or prohibited.
> - Moreover, we believe that our theoretical findings can be extended to a more general **online** continual learning setup, beyond the current setup that uses the same data points over and over again. Consider a finite number of tasks’ data distributions (with bounded supports) which are jointly separable, and let them be cyclically/randomly chosen one by one at each stage. Suppose a finite-size training dataset from each data distribution is sampled every time we encounter a task. Even in this case, we expect that the story would be the same: the continually trained linear classifier will eventually converge to the max-margin direction that jointly classify every task’s data distribution support. In our revised paper, we empirically demonstrate this in Appendix C.2.4.
> - This example demonstrates that the intuitions gained from our analysis can extend to more realistic and complex scenarios. In our paper, however, we focus only on our data repetition setting so that precise theoretical guarantees are attainable; rigorous extensions to more general settings would be much more challenging.
>
> **W4. Batch = Task? It’s not surprising that SGD converges.**
>
> - As the reviewer pointed out, the algorithm we analyzed (sequential GD) and the usual online SGD share several properties. At every time step, they use some part of the dataset (called "batch") to update the model iteratively. Hence, both can be used when we cannot access the full dataset at every update.
> - However, the critical difference between these two is in **repetition**. When we theoretically analyze (online) SGD, the most common assumption is that every stochastic gradient is independently sampled. Thus, if we collect many gradients, the overall update can eventually approximate the full-batch gradient update. When we run sequential GD in the continual learning setup, however, there must be certain dependencies between updates since we repeatedly use the same batch of data points for $K$ consecutive updates during each stage. Thus, if we do many updates for a single task (i.e., large $K$), the total update over each stage is significantly biased toward a specific task, not the joint/offline solution. These dependencies made it difficult to apply the usual convergence analysis of optimization algorithms, which was the main obstacle we had to overcome. Thus, we claim that our theoretical result is not a direct result of SGD convergence analysis.
> - We also note that our theoretical analysis covers not only the convergence (in loss and the iterate direction) but also a non-asymptotic characterization of forgetting, which is one of the most important quantities in the literature on continual learning.
>
> **W5. It has to be specific in terms of standard datasets and real-world scenarios.**
>
> - Thank you for a great suggestion. Here we provide more precise references and citations about the cyclic/random task ordering. We also attached the corresponding references/citations in revised paper.
> - Cyclic task ordering:
>     - Number of hits for certain keywords (e.g., ‘dinner’)  in search engine ([trends.google.com/trends/](https://trends.google.com/trends/explore?date=now%207-d&q=dinner))
>     - Hourly revenue of a recommender system (Yang et al., 2022)
> - Random task ordering:
>     - Autonomous driving (Verwimp et al., 2023)

---

> > ### Author Response · Authors · 2024-11-22
> > **Author's Response (2)**
> >
> > **W6. Practical insights**
> >
> > - Our theoretical analyses may not immediately offer any guidelines or suggestions for practitioners interested in general continual learning problems. However, we believe that our work provides a theoretical basis for a concrete **baseline**: “you should do better than this.” (A paper with similar high-level idea: Prabhu et al. (2020)) Our theory reveals that when the same (or similar) tasks appear repeatedly, a naive algorithm without any techniques (e.g., regularization or replay) specialized for continual learning will eventually “work,” in terms of convergence to a desirable offline solution and vanishing forgetting. Thus, we expect that such a naive algorithm does not always fail in more practical scenarios, and any new approach for continual learning should be able to outperform this simple baseline.
> > - Furthermore, we believe our analysis can serve as a stepping stone to more advanced theoretical analyses on more practical approaches such as regularization- and replay-based methods.
> >
> > ---
> >
> > **References**
> >
> > - Prabhu et al., GDumb: A Simple Approach that Questions Our Progress in Continual Learning. ECCV 2020.
> > - Verwimp et al., CLAD: A realistic Continual Learning benchmark for Autonomous Driving, Neural Networks, Vol. 161, 2023.
> > - Yang et al., Fourier Learning with Cyclical Data, ICML 2022.

---

> ### Author Response · Authors · 2024-11-28
> **Kind reminder regarding review process**
>
> Dear reviewer sGag
>
> Thank you for your important questions and insightful comments in the first review. We uploaded our developed manuscript which provides clearer real-world scenarios and supplementary experiments. As the discussion period is almost over, please check out our revised paper and our response if you haven’t done so. We hope that our revision and the rebuttal resolve your initial concerns. If you feel the same, we would appreciate if this could be reflected through an updated score. Also, please do not hesitate to let us know if you have any remaining questions. We appreciate your contribution to the reviewing process.
>
> Sincerely,
> Authors

---

### Author Response · Authors · 2024-11-22
**General Response about Experiments (1/3)**

Dear all reviewers,

We genuinely appreciate your insightful reviews and valuable comments. We are happy about the reviews being mostly positive, providing constructive and insightful feedback.

We find that one of the most common comments on our work is the lack of experimental results. Being a theoretical work based on mathematical equations and proofs, we acknowledge that we did not provide sufficient empirical demonstrations. Thus, here, we present new experiments we conducted and finalized during the discussion period and our plans about additional experiments.

## **1. Synthetic (but more realistic) 2D datasets**

- Reviewers sGag & uaBW raised a question that the toy experimental setting in our original submission (involving less than 10 data points) is too artificial. To address their concern, we conduct experiments on a set of more realistic synthetic 2D datasets and put the results in our revised manuscript.
- We carefully constructed a data generation process (Appendix C.2.1) and randomly generated three jointly separable classification tasks. Given that the tasks are revealed in a cyclic order, we visually observe the directional convergence of sequential GD’s iterates towards the joint max-margin classifier of all tasks (Figure 2) and the loss convergence (Figure 5). We also observe a similar behavior of sequential GD under random task ordering (Appendix C.2.3).
- We hope that the visualization of our data points and the trajectory of sequential GD iterates shown in Figure 2 will be able to handle the concerns about clearer visualization in 2D (raised by reviewer a4BQ) and about geometric interpretation of our theoretical findings (raised by reviewer uaBW) as well.

## **2. Beyond batch repetition: synthetic 2D data “distributions”**

- Reviewers sGag, uaBW, and a4BQ raised a question about the generality of problem setting for our theoretical analysis, especially about repeatedly used identical batches of data.
- To demonstrate that our theoretical insight still holds beyond the batch repetition setup we theoretically analyzed, we further extend our experiments on synthetic 2D data to non-repeating dataset cases. Instead, we consider three data “distributions” (with jointly separable and bounded support) each corresponding to a task, but we draw a totally new dataset from a given task’s distribution at the beginning of every stage when we encounter it. Again, in both cyclic ordering and random ordering of task’s distributions, we still observe that a similar implicit bias and convergence behavior appear in dataset repetition cases (Appendix C.2.4).

## **3. Realistic data: CIFAR-10 experiment with linear models**

- Reviewer sGag raised a concern that there are no experiments on more realistic data. To address this, we conduct additional experiments by switching the dataset to CIFAR-10 (Krizhevsky, 2009).
- We conducted an experiment on CIFAR-10 and discussed the result in Appendix C.5 of the updated manuscript. Our theoretical result on linearly non-separable data such as CIFAR-10 shows that sequential GD iterates should not diverge and instead converge to the global minimum under the properly chosen learning rate. To corroborate our theoretical finding, we train a linear model using joint task data to get a proxy $\hat w$ of the global minimum solution of the joint training loss, and then measure the distance between the jointly trained model parameter $\hat w$ and iterates of sequential GD at the end of every stage. As a result, we observe that the distance diminishes to near zero, even when we do not adopt a small learning rate as our theorem requires.

(continued in the next official comment)

---

> ### Author Response · Authors · 2024-11-22
> **General Response about Experiments (2/3)**
>
> ## **4. Deeper model: synthetic 2D data with ReLU nets**
>
> - Some reviewers including 1DCj pointed out that there are no deep learning experiments. To address the concern, we plan to train ReLU networks on our synthetic 2D datasets with sequential GD and see what will happen.
> - For simple (e.g., 2D) datasets, in particular, there is a method for visualization of decision boundary of a neural network (Pezeshki et al., 2021). Observe that monitoring the trajectory of the parameter vector of a linear classifier is semantically equivalent to monitoring the evolution of decision boundary the model. Interpreting our implicit bias result as the convergence between decision boundaries of the continually learned model and a jointly trained model, we can also verify our theoretical insight through ReLU net experiments on 2D data by monitoring the decision boundary.
> - Due to the time constraint, we are still running the experiments. We will update our manuscript before the discussion period finishes.
> - **(Update: 2024.11.25)** Our new revision contains the experimental results with ReLU nets! Please check out our paper and an additional response below.
>
> ## **5. Large-scale experiments with even deeper models and more realistic datasets**
>
> - Continuing from the previous response (#4), we initially planned to conduct more experiments on various deep network architectures and more (high-dimensional) datasets other than synthetic ones. However, due to our constraints in time and computational resources, there was not enough time to build up a code base for these experiments.
> - More importantly, as far as we know, it is not very straightforward which statement we should verify by these experiments. Considering a comparison between continually learned vs. jointly trained models (in terms of implicit bias), it is not really straightforward to judge whether two neural networks are close or not; for instance, simple L2 distance between parameter values is not enough for this because there can be many different model parameter configurations representing exactly the same mapping from input to output.
> - Still, we believe we might be able to assess the (cycle-averaged) forgetting and loss convergence speeds for these realistic settings. We plan to include these results in a later revision.
>
> We hope this general response will handle most reviewer’s concerns about lack of experiments. Thank you again for your time and effort in reviewing our work.
>
> Best,
>
> Authors
>
> ---
>
> **References**
>
> - Krizhevsky. “Learning Multiple Layers of Features from Tiny Images.” 2009.
> - Pezeshki et al. “Gradient Starvation: A Learning Proclivity in Neural Networks.” NeurIPS 2021.

---

> ### Author Response · Authors · 2024-11-25
> **General Response about Experiments (3/3) (New!)**
>
> Dear all reviewers,
>
> **We just made an additional revision to our experiments and the paper, as we promised before. The updates in the paper are marked in green. Before describing the updates, please recall that the end of the discussion period is only a few days away. Since we have responded to all reviewers' concerns and incorporated corresponding changes in our revised paper, please look into our responses and the new manuscript. If these address your concerns, it would be appreciated if this could be reflected via updated assessments. If you have any remaining concerns or questions, we would be grateful to hear them and will do our best to address them as soon as possible.**
>
> For the rest of this official comment, we briefly summarize the key updates in our paper. Note that we also updated the codes provided in the supplementary material.
>
> ### **Overall changes throughout the paper**
>
> * We modified a bit misleading term "task similarity" into "task alignment" to capture the meaning of the terms $N\_{p,q}$ and $\bar{N}\_{p,q}$ in essence. For a detailed context on it, please refer to our response to the reviewer 1DCj's Q3.
>
> ### **Section 1: Introduction**
>
> * When describing a previous work (Evron et al., 2023), we toned down our previous criticism on it in the fourth paragraph, as per the reviewer a4PQ's suggestion.
> * We slightly changed the visualization in Figure 1, which originally contained too many points and thus slowed down opening the paper. Instead of tracking every gradient update, we drew only the end-of-stage iterates in order to capture the stage-wise evolutions in each trajectory, which is enough to showcase the implicit bias of sequential GD in our toy example. The code for generating Figure 1 is modified as well, which can be found in our supplementary material.
>
> ### **Section 2: Problem setup**
>
> * As per the reviewer sGag's suggestion (Weakness 5), we cited explicit examples of cyclic and random task orderings.
>
> ### **Section 3: Cyclic Learning of Jointly Separable Tasks**
>
> * **Figure 2:** We conduct an experiment on a more sophisticated two-dimensional dataset and provide a clear visualization of the parameter trajectories, clearly exhibiting the expected implicit bias toward the joint max-margin in direction.
> * Below Figure 2, we left comments about:
>   * The experimental results on a more general continual learning setup where the total dataset is no longer fixed over time.
>   * The experimental results with shallow (two-layer) ReLU networks.
> * **Figure 3:** We provide a better visual description of the cases where the cycle-averaged forgetting is small ("aligned") and large ("contradict").
>
> ### **Section 5: Beyond Jointly Separable Tasks**
>
> * The original version of Lemma 5.1 misses the dependency on a parameter $G$ (introduced in Assumption 5.2): we corrected it.
> * Below Theorem 5.2, We added a discussion on the loss convergence rate, naturally derived from the iterate convergence rate in the theorem.
> * After that, we left a comment on a real-world dataset experiment (with CIFAR-10).
>
> ### **Appendix A: Other Related Works**
>
> * Due to the space limit, we moved this section to the very first appendix.
>
> ### **Appendix B: Brief Overview of Evron et al. (2023) and Comparisons**
>
> * We now provide a better description & discussion on Sequential Max-Margin (SMM) algorithm, especially in the last two paragraphs.
> * We moved the plot about SMM (not converging to the joint max-margin in direction) to the next appendix (Appendix C).
>
> ### **Appendix C: Experiment Details & Omitted Experimental Results**
>
> * ***We put almost all experimental results produced during the discussion period in this appendix!***
> * All codes for generating every figure are now available in our supplementary material.
> * Here is a list of experiments added during the discussion period:
>   * More realistic synthetic 2D datasets + linear classifier, under cyclic and random task ordering. (Appendices C.2.1 - C.2.3, see #1 in our general response above.)
>   * Constantly changing 2D datasets + linear classifier, under cyclic and random task ordering. (Appendix C.2.4, see #2 in our general response above.)
>   * 2D dataset + two-layer **ReLU** nets. (Appendix C.4, see #4 in our general response above.) **(NEW!)**
>     * Although we only present the cyclic ordering case in our paper, we had almost identical observations for random ordering and even under constantly changing datasets. You can find the plots in our supplementary material.
>   * **CIFAR-10** + linear classifier. (Appendix C.5, see #3 in our general response above.)
>
> Again, we thank all the reviewers for their time and effort in reviewing our work. We hope these revisions address your concerns. Please do not hesitate to leave more comments and questions if you have any.
>
> Warm regards,
>
> Authors

---

### Meta-Review · Area_Chair_6dAy · 2024-12-23

**Metareview:**

This work investigates continual learning for multiple linear classification tasks using gradient descent (GD) with a fixed iteration budget per task. For tasks that are jointly linearly separable and presented in a cyclic or random order, the trained linear classifier is shown to converge directionally to the joint offline max-margin solution. This is a clean and novel theoretical result, and the paper is well-written. While the empirical component is not extensive, the authors have made a good effort to include experiments that support their findings. Overall, the strength of the theoretical results justifies acceptance of this work.

**Additional Comments On Reviewer Discussion:**

3 out of the 4 reviewers leaned toward acceptance. Reviewer sGag gave a score of 5 and had questions about the technical details in the setting as well as the practical relevance of the work. The AC thinks that these questions were addressed satisfactorily in the author rebuttal.

---

### Decision · Program_Chairs · 2025-01-22

Accept (Poster)